# Two-timescale Extragradient for Finding Local Minimax Points

**Jiseok Chae, Kyuwon Kim & Donghwan Kim**
Department of Mathematical Sciences, KAIST
{jsch,kkw4053,donghwankim}@kaist.ac.kr

## ABSTRACT

Minimax problems are notoriously challenging to optimize. However, we present that the two-timescale extragradient method can be a viable solution. By utilizing dynamical systems theory, we show that it converges to points that satisfy the second-order necessary condition of local minimax points, under mild conditions that the two-timescale gradient descent ascent fails to work. This work provably improves upon all previous results on finding local minimax points, by eliminating a crucial assumption that the Hessian with respect to the maximization variable is nondegenerate.

## 1 INTRODUCTION

Many noteworthy modern machine learning problems, such as generative adversarial networks (GANs) (Goodfellow et al., 2014), adversarial training (Mądry et al., 2018), and sharpness-aware minimization (SAM) (Foret et al., 2021), are instances of minimax problems, formulated as $\min_{\boldsymbol{x}} \max_{\boldsymbol{y}} f(\boldsymbol{x}, \boldsymbol{y})$. First-order methods, such as *gradient descent ascent* (GDA) (Arrow et al., 1958) and *extragradient* (EG) (Korpelevich, 1976), are workhorses of minimax optimization in modern machine learning, but they still remain remarkably unreliable. This is in contrast to the remarkable success of *gradient descent* for minimization problems in machine learning, which is supported by theoretical results; under mild conditions, gradient descent converges to a local minimum, and almost surely avoids strict saddle points (Lee et al., 2016; 2019). Minimax optimization, however, lacks such comparable theory, and this paper is a step towards establishing it.

If the problem is convex-concave, then by Sion's minimax theorem (Sion, 1958), the order of $\min$ and $\max$ is insignificant. However, modern machine learning problems are highly nonconvex-nonconcave, and thus their order does matter. Indeed, one of the complications in training GANs, called the *mode collapse* phenomenon, is considered to be due to finding a solution of a max-min problem, rather than that of the min-max problem (Goodfellow, 2016). The widely used notion of saddle points, also called Nash equilibria, fails to capture the ordered structure of minimax problems (Jin et al., 2020). Accordingly, Jin et al. (2020) introduced a new appropriate notion for local optimum in minimax problems, called *local minimax points*, built upon Stackelberg equilibrium from sequential game theory (von Stackelberg, 2011), which encompasses the Nash equilibrium.

Yet, how one can find such local minimax points (possibly via first-order methods) is still left as an open question. A partial answer was provided also by Jin et al. (2020); they showed that the *two-timescale* GDA (Heusel et al., 2017), *i.e.,* GDA with different step sizes for $\boldsymbol{x}$ and $\boldsymbol{y}$, converges to certain (but not any) local minimax points. In particular, they only covered the case where the Hessian with respect to the maximization variable $\nabla^2_{\boldsymbol{yy}} f$ is nondegenerate, which disregards possibly meaningful "degenerate" local optimal points, *e.g.,* in over-parameterized training (Liu et al., 2022). In this paper, we show that the *two-timescale* EG can actually find local minimax points beyond the assumption that $\nabla^2_{\boldsymbol{yy}} f$ is nondegenerate. So, our main contribution is providing a more complete answer to the aforementioned open question. Our specific contributions can be listed as follows.

- In Section 3, we derive a second order characterization of local minimax points, without assuming that $\nabla^2_{\boldsymbol{yy}} f$ is nondegenerate. In doing so, we introduce the notion of a *restricted* Schur complement. This leads to a natural way of defining *strict non-minimax points* that we would like to avoid, analogous to the strict saddle points in minimization.

- In Section 4, we develop new tools to achieve the main results in Section 5. These tools include a new spectral analysis that does not rely on the nondegeneracy assumption on $\nabla^2_{\boldsymbol{yy}} f$, as well as the concept of hemicurvature to better understand the behavior of eigenvalues.

- In Section 5, from a dynamical system perspective, we show that the limit points of the two-timescale EG in the continuous time limit are the local minimax points under mild conditions, by establishing a second order property of those points. This continuous-time analysis is then used to derive a similar conclusion in the discrete-time case. In the discrete-time case, Section 5.4 further shows that two-timescale EG almost surely avoids (undesirable) *strict non-minimax points*, as desired, while Section 4.3 shows that two-timescale GDA may avoid (desirable) local minimax points where $\nabla^2_{\boldsymbol{yy}} f$ is *degenerate*.

- In Section 6, we extend the local result of Section 5 to a global statement: under the Minty variational inequality (MVI) condition (Minty, 1967) and additional mild conditions, the two-timescale EG *globally* converges to local minimax points.

## 2 PRELIMINARIES

### 2.1 NOTATIONS AND PROBLEM SETTING

The *spectrum* of a square matrix $\boldsymbol{A}$, denoted by $\mathrm{spec}(\boldsymbol{A})$, is the set of all eigenvalues of $\boldsymbol{A}$. The *range* of a matrix $\boldsymbol{A}$, denoted by $\mathcal{R}(\boldsymbol{A})$, is the span of its column vectors. The *open left* (resp. *right*) *half plane*, denoted by $\mathbb{C}^\circ_-$ (resp. $\mathbb{C}^\circ_+$), is the set of all complex numbers $z$ such that $\mathrm{Re}\, z < 0$ (resp. $\mathrm{Re}\, z > 0$). The *imaginary axis* is denoted by $i\mathbb{R}$. To denote the minimization variable $\boldsymbol{x} \in \mathbb{R}^{d_1}$ and the maximization variable $\boldsymbol{y} \in \mathbb{R}^{d_2}$ at once, we use the notation $\boldsymbol{z} := (\boldsymbol{x}, \boldsymbol{y})$. Let $C^2$ be the set of twice continuously differentiable functions. The saddle-gradient operator of the objective function $f$ will be denoted by $\boldsymbol{F} := (\nabla_{\boldsymbol{x}} f, -\nabla_{\boldsymbol{y}} f)$, and the derivative of $\boldsymbol{F}$ will be denoted by $D\boldsymbol{F}$. When necessary, we will impose the following standard assumption on $f$.

**Assumption 1.** *Let $f \in C^2$, and there exists $L > 0$ such that $\|D\boldsymbol{F}(\boldsymbol{z})\| \leq L$ for all $\boldsymbol{z}$.*

### 2.2 LOCAL MINIMAX POINTS

Jin et al. (2020) introduced the following new notion of local optimality for minimax problems.

**Definition 1** (Jin et al. (2020))**.** *A point $(\boldsymbol{x}^*, \boldsymbol{y}^*)$ is said to be a **local minimax point** if there exists $\delta_0 > 0$ and a function $h$ satisfying $h(\delta) \to 0$ as $\delta \to 0$ such that, for any $\delta \in (0, \delta_0]$ and any $(\boldsymbol{x}, \boldsymbol{y})$ satisfying $\|\boldsymbol{x} - \boldsymbol{x}^*\| \leq \delta$ and $\|\boldsymbol{y} - \boldsymbol{y}^*\| \leq \delta$, we have*

$$f(\boldsymbol{x}^*, \boldsymbol{y}) \leq f(\boldsymbol{x}^*, \boldsymbol{y}^*) \leq \max_{\boldsymbol{y}' \,:\, \|\boldsymbol{y}' - \boldsymbol{y}^*\| \leq h(\delta)} f(\boldsymbol{x}, \boldsymbol{y}'). \tag{1}$$

Local minimax points can be characterized by the following conditions (Jin et al., 2020).

- (First-order necessary) For $f \in C^1$, any local minimax point $\boldsymbol{z}^*$ satisfies $\nabla f(\boldsymbol{x}^*, \boldsymbol{y}^*) = \boldsymbol{0}$.
- (Second-order necessary) For $f \in C^2$, any local minimax $\boldsymbol{z}^*$ satisfies $\nabla^2_{\boldsymbol{yy}} f(\boldsymbol{x}^*, \boldsymbol{y}^*) \preceq \boldsymbol{0}$. In addition, if $\nabla^2_{\boldsymbol{yy}} f(\boldsymbol{x}^*, \boldsymbol{y}^*) \prec \boldsymbol{0}$, then $[\nabla^2_{\boldsymbol{xx}} f - \nabla^2_{\boldsymbol{xy}} f(\nabla^2_{\boldsymbol{yy}} f)^{-1} \nabla^2_{\boldsymbol{yx}} f](\boldsymbol{x}^*, \boldsymbol{y}^*) \succeq \boldsymbol{0}$.
- (Second-order sufficient) For $f \in C^2$, any stationary point $\boldsymbol{z}^*$ satisfying

$$[\nabla^2_{\boldsymbol{xx}} f - \nabla^2_{\boldsymbol{xy}} f(\nabla^2_{\boldsymbol{yy}} f)^{-1} \nabla^2_{\boldsymbol{yx}} f](\boldsymbol{x}^*, \boldsymbol{y}^*) \succ \boldsymbol{0} \quad \text{and} \quad \nabla^2_{\boldsymbol{yy}} f(\boldsymbol{x}^*, \boldsymbol{y}^*) \prec \boldsymbol{0} \tag{2}$$

   is a local minimax point.

Here, the second-order necessary condition is loose when $\nabla^2_{\boldsymbol{yy}} f$ is degenerate. However, the existing works mostly overlook this discrepancy between that and the second-order sufficient condition (2), and focus only on the *strict*[1] local minimax points (Jin et al., 2020), also known as differential Stackelberg equilibria (Fiez et al., 2020), which are the stationary points that satisfy (2). We close this gap in Section 3, and consider a broader set of local minimax points in later sections.

---

[1]The term *strict* used here is slightly more restrictive than that in the usual strict notion of local optimum in minimization; see *e.g.,* (Wright & Recht, 2022, p.15).

### 2.3 GRADIENT DESCENT ASCENT AND EXTRAGRADIENT

This paper considers the following two standard gradient methods (with time separation introduced later). Here, $\eta$ denotes the step size.

- Gradient descent ascent (GDA) (Arrow et al., 1958): $\quad z_{k+1} = z_k - \eta F(z_k)$
- Extragradient (EG) (Korpelevich, 1976): $\quad z_{k+1} = z_k - \eta F(z_k - \eta F(z_k))$

We also consider their continuous time limits, as the analyses in continuous time is more accessible than their discrete counterparts. The analyses in discrete time, which is more of our interest, can be easily translated from the continuous time limit analyses. Taking the limit as $\eta \to 0$, both GDA and EG share the same continuous time limit $\dot{z}(t) = -F(z(t))$. However, it is also known that even when $f$ is convex-concave, EG converges to an optimum, while GDA and $\dot{z}(t) = -F(z(t))$ may not (Mescheder et al., 2018). This distinction suggested a need for ODEs with a higher order of approximation. In particular, Lu (2022) derived the ODE $\dot{z}(t) = -F(z(t)) + (\eta/2)DF(z(t))F(z(t))$ as the $O(\eta)$-approximation of EG, in the sense that it captures the dynamics of EG up to order $O(\eta)$ as $\eta \to 0$. In our dynamical system analyses, we found the following to be particularly more useful.

**Lemma 2.1.** *Under Assumption 1, let $s := \eta/2$ and $0 < s < 1/L$. Then, the ordinary differential equation $\dot{z}(t) = -(I + sDF(z(t)))^{-1}F(z(t))$ is a $O(s)$-approximation of EG.*

### 2.4 DYNAMICAL SYSTEMS

This paper analyzes (two-timescale) GDA and EG methods through dynamical systems, in both continuous-time and discrete-time. In particular, we determine whether an equilibrium point $z^*$ is *asymptotically stable*, which means that a trajectory starting at a point that is sufficiently close to $z^*$ will remain close enough and eventually converge to $z^*$. We adopt the following widely used criteria, which we refer to as *strict linear stability*, as a sufficient condition for an equilibrium to be asymptotically stable; see, *e.g.,* (Khalil, 2002, Theorem 4.7) and (Galor, 2007, Theorem 4.8).

**Definition 2** (Linear stability of dynamical systems)**.**

(i) *(Continuous system) For a $C^1$ mapping $\phi$, an equilibrium point $z^*$ of $\dot{z}(t) = \phi(z(t))$, such that $\phi(z^*) = 0$, is a **strict linearly stable point** if its Jacobian matrix at $z^*$ has all eigenvalues with negative real parts, i.e., $\mathrm{spec}(D\phi(z^*)) \subset \mathbb{C}_-^\circ$.*

(ii) *(Discrete system) For a $C^1$ mapping $w$, an equilibrium point $z^*$ of $z_{k+1} = w(z_k)$, such that $z^* = w(z^*)$, is a **strict linearly stable point** if its Jacobian matrix at $z^*$ has spectral radius smaller than 1, i.e., $\rho(Dw(z^*)) < 1$.*

The set of *unstable* equilibrium points in a discrete dynamical system is similarly defined.

**Definition 3.** *Given a $C^1$ mapping $w$, the set $\mathcal{A}^*(w) := \{z^* : z^* = w(z^*), \ \rho(Dw(z^*)) > 1\}$ is the set of **strict linearly unstable** equilibrium points.*

We also study whether the methods avoid strictly non-optimal points in minimax problems, using the following theorem, built upon the stable manifold theorem (Shub, 1987). This was used by Lee et al. (2019) to show that gradient descent in minimization almost surely escapes strict saddle points.

**Theorem 2.2** (Lee et al. (2019, Theorem 2))**.** *Let $w$ be a $C^1$ mapping such that $\det(Dw(z)) \neq 0$ for all $z$. Then the set of initial points that converge to a strict linearly unstable equilibrium point has (Lebesgue) measure zero, i.e., $\mu(\{z_0 : \lim_{k\to\infty} w^k(z_0) \in \mathcal{A}^*(w)\}) = 0$.*

## 3 ON THE NECESSARY CONDITION OF LOCAL MINIMAX POINTS

In minimization, both second-order necessary and sufficient conditions of local minimum are simply characterized by the Hessian of the objective function; see *e.g.,* (Wright & Recht, 2022, Theorems 2.4 and 2.5). However, not a similar simple correspondence between conditions for a local minimax point was previously known, making it difficult to further develop a theory for minimax optimization comparable to that in minimization. In this section, we show that such correspondence in fact exists.

### 3.1 RESTRICTED SCHUR COMPLEMENT

For the clarity of presentation, we first define a variant of Schur complement, named *restricted Schur complement*, which we will soon relate to second-order conditions of a local minimax point.

From now on, we assume that $f$ is twice continuously differentiable. Let us denote the second derivatives of $f$ by $\boldsymbol{A} = \nabla^2_{\boldsymbol{xx}} f \in \mathbb{R}^{d_1 \times d_1}$, $\boldsymbol{B} = \nabla^2_{\boldsymbol{yy}} f \in \mathbb{R}^{d_2 \times d_2}$, and $\boldsymbol{C} = \nabla^2_{\boldsymbol{xy}} f \in \mathbb{R}^{d_1 \times d_2}$. In particular we can express the Jacobian matrix of the saddle-gradient $\boldsymbol{F}$ as

$$\boldsymbol{H} := D\boldsymbol{F} = \begin{bmatrix} \boldsymbol{A} & \boldsymbol{C} \\ -\boldsymbol{C}^\top & -\boldsymbol{B} \end{bmatrix} \in \mathbb{R}^{(d_1+d_2) \times (d_1+d_2)}. \tag{3}$$

Although $\boldsymbol{A}$, $\boldsymbol{B}$, and $\boldsymbol{C}$ are, strictly speaking, matrix valued *functions* of $(d_1 + d_2)$-dimensional vector inputs, the point where those functions are evaluated will be clear from context, so we simply write $\boldsymbol{A}$, $\boldsymbol{B}$, and $\boldsymbol{C}$ to denote the function values.

Let $r := \text{rank}(\boldsymbol{B})$. As $\boldsymbol{B}$ is symmetric, it can be orthogonally diagonalized as $\boldsymbol{B} = \boldsymbol{P}\boldsymbol{\Delta}\boldsymbol{P}^\top$, where $\boldsymbol{\Delta} = \text{diag}\{\delta_1, \ldots, \delta_r, 0, \ldots, 0\}$ and $\boldsymbol{P}$ is orthogonal. Let $\boldsymbol{\Gamma}$ be a submatrix of $\boldsymbol{CP}$, which consists of the $d_2 - r$ rightmost columns of $\boldsymbol{CP}$, let $q := \text{rank}(\boldsymbol{\Gamma})$, and let $\boldsymbol{U} \in \mathbb{R}^{d_1 \times (d_1-q)}$ be a matrix whose columns form an orthonormal basis of $\mathcal{R}(\boldsymbol{\Gamma})^\perp$. Under this setting, we define the following.

**Definition 4.** *For a matrix $\boldsymbol{H}$ in the block form of* (3)*, using the definition*[2] *of $\boldsymbol{U}$ given above, the **restricted Schur complement**[3] is defined as $\boldsymbol{S}_{\text{res}}(\boldsymbol{H}) := \boldsymbol{U}^\top(\boldsymbol{A} - \boldsymbol{C}\boldsymbol{B}^\dagger\boldsymbol{C}^\top)\boldsymbol{U} \in \mathbb{R}^{(d_1-q) \times (d_1-q)}$.*

The positive semidefiniteness of the restricted Schur complement $\boldsymbol{S}_{\text{res}}$ is related to the (generalized) Schur complement $\boldsymbol{S} := \boldsymbol{A} - \boldsymbol{C}\boldsymbol{B}^\dagger\boldsymbol{C}^\top$ being positive semidefinite on a certain subspace, as follows. We remark that a $0 \times 0$ matrix is considered to be "vacuously" positive semidefinite.

**Proposition 3.1.** *The restricted Schur complement $\boldsymbol{S}_{\text{res}}(\boldsymbol{H})$ is positive semidefinite if and only if $\boldsymbol{v}^\top\boldsymbol{S}\boldsymbol{v} \geq 0$ for any $\boldsymbol{v} \in \mathbb{R}^{d_1}$ satisfying $\boldsymbol{C}^\top\boldsymbol{v} \in \mathcal{R}(\boldsymbol{B})$.*

### 3.2 REFINING THE SECOND-ORDER NECESSARY CONDITION

We then have the following second-order necessary condition in terms of the restricted Schur complement, which is our first main contribution. Notice that when $\boldsymbol{B}$ is invertible, the restricted Schur complement becomes the usual Schur complement $\boldsymbol{A} - \boldsymbol{C}\boldsymbol{B}^{-1}\boldsymbol{C}^\top$, which appears in the second-order sufficient condition (2). Thus, the *restricted Schur complement* provides a unified point of view in considering both the necessary and sufficient conditions.

**Proposition 3.2** (Refined second-order necessary condition)**.** *Let $f \in C^2$, then any local minimax point $(\boldsymbol{x}^*, \boldsymbol{y}^*)$ satisfies $\nabla^2_{\boldsymbol{yy}} f(\boldsymbol{x}^*, \boldsymbol{y}^*) \preceq \boldsymbol{0}$. In addition, if the function $h(\delta)$ in Definition 1 satisfies $\limsup_{\delta \to 0+} h(\delta)/\delta < \infty$, then $\boldsymbol{S}_{\text{res}}(D\boldsymbol{F}(\boldsymbol{x}^*, \boldsymbol{y}^*)) \succeq \boldsymbol{0}$.*

**Remark 3.3.** *The additional condition on $h$ in Proposition 3.2 slightly limits the choice of $h$, which determines the size of the set we take the maximum over in Definition 1. Hence, this can be interpreted as restricting the local minimax points we are interested in, or alternatively, refining the definition of local minimax points itself; c.f., (Ma et al., 2023). Note, however, that such refined definition is yet much broader than the definition of strict local minimax points.*

Based on the refined second-order necessary condition, we define *strict non-minimax* points as below. These are the points we hope to avoid, and the definition is analogous to that of strict saddle points in minimization problems (Lee et al., 2016). In Section 5.4, we show that two-timescale EG almost surely avoids strict non-minimax points.

**Definition 5** (Strict non-minimax point; $\mathcal{T}^*$)**.** *A stationary point $\boldsymbol{z}^*$ is said to be a **strict non-minimax point** of $f$ if $\lambda_{\min}(\boldsymbol{S}_{\text{res}}(D\boldsymbol{F}(\boldsymbol{z}^*))) < 0$ or $\lambda_{\min}(-\nabla^2_{\boldsymbol{yy}} f(\boldsymbol{z}^*)) < 0$, where $\lambda_{\min}(\boldsymbol{A})$ denotes the smallest eigenvalue of $\boldsymbol{A}$. We denote the set of strict non-minimax points by $\mathcal{T}^*$.*

When necessary, we will impose the following assumption at stationary points $\boldsymbol{z}^*$ that is weaker than the nondegeneracy condition on $\nabla^2_{\boldsymbol{yy}} f(\boldsymbol{z}^*)$, which was crucial in all existing literatures.

---

[2]A matrix $\boldsymbol{U}$ is not unique in general, but there are ways to fix the choice of $\boldsymbol{U}$, *e.g.*, applying a predetermined QR factorization algorithm on $\boldsymbol{\Gamma}$. However, because only the spectrum of $\boldsymbol{S}_{\text{res}}(\boldsymbol{H})$ will be important in the subsequent analyses, we do not specify the choice of $\boldsymbol{U}$.

[3]A similar matrix has also been considered by Zhang et al. (2022, Theorem 4.4), but as a part of a local optimality condition for *quadratic* problems only.

**Assumption 2.** *For a stationary point $z^*$ in consideration, at least one of the matrices $S_{\mathrm{res}}(DF(z^*))$ and $\nabla_{yy}^2 f(z^*)$ is nondegenerate.*

**Example 1.** *We provide two simple representative examples of "non-strict" local minimax points satisfying Assumption 2, especially with nondegenerate $S_{\mathrm{res}}(DF)$ and degenerate $\nabla_{yy}^2 f$.*

- *$f_{\mathsf{b}}(x, y) = xy$ has a unique local minimax point $\mathbf{0} = (0, 0)$;*
- *$f_{\mathsf{q}}(x_1, x_2, y_1, y_2, y_3) = \frac{1}{2}x_1^2 - \frac{1}{2}y_1^2 - \frac{1}{2}y_3^2 + x_2 y_2 + y_1 y_3$ has non-unique and non-isolated local minimax points $(0, 0, t, 0, t)$ for all $t \in \mathbb{R}$.*

*See Appendix D.3 for the details and proofs. Later in Examples 2, 3 and 4, we show that our proposed two-timescale EG finds these optima, while GDA (and its timescaled variant) avoids them.*

Only when characterizing *strict* linear stability of equilibrium points in Sections 5.2 and 5.3, we will use the following slightly stronger version of Assumption 2.

**Assumption 2'.** *For $z^*$ in consideration, in addition to Assumption 2, $DF(z^*)$ is nondegenerate.*

# 4 CHARACTERIZING TIMESCALE SEPARATION WITHOUT NONDEGENERACY CONDITION ON $\nabla_{yy}^2 f$

In this section we begin with reviewing the close relationship between two-timescale GDA and strict local minimax points, under a nondegeneracy condition on $\nabla_{yy}^2 f(z^*)$ (Jin et al., 2020). Then, we extend this to a more general setting without such nondegeneracy condition, and observe a negative result stating that two-timescale GDA avoids some degenerate local minimax points.

## 4.1 TIMESCALE SEPARATION IN GDA AND ITS RELATION TO STABILITY

Let us recall the *two-timescale GDA* method, which is GDA with a timescale separation with timescale parameter $\tau \geq 1$:

$$z_{k+1} = \tilde{w}_\tau(z_k) \coloneqq z_k - \eta \Lambda_\tau F(z_k) \quad \text{where} \quad \Lambda_\tau \coloneqq \mathrm{diag}\{(1/\tau)I, I\}, \qquad (4)$$

and its continuous time limit $\dot{z}(t) = -\Lambda_\tau F(z(t))$. The stability of (4) and its continuous limit depends on the spectrum of a timescaled matrix

$$H_\tau \coloneqq \Lambda_\tau H = \begin{bmatrix} \frac{1}{\tau}A & \frac{1}{\tau}C \\ -C^\top & -B \end{bmatrix}.$$

Jin et al. (2020) studied the following asymptotic behavior of its spectrum as $\epsilon = 1/\tau \to 0$, under a nondegeneracy condition on $\nabla_{yy}^2 f(z^*)$.

**Lemma 4.1** (Jin et al. (2020, Lemma 40)). *Suppose that $B = \nabla_{yy}^2 f(z^*)$ is nondegenerate. Then, the $d_1 + d_2$ complex eigenvalues $\lambda_j$ of $H_\tau$ have the following asymptotics as $\epsilon = 1/\tau \to 0+$:*

$$|\lambda_j - \epsilon\mu_j| = o(\epsilon), \quad j = 1, \dots, d_1, \qquad |\lambda_{j+d_1} - \nu_j| = o(1), \quad j = 1, \dots, d_2,$$

*where $\mu_j$ and $\nu_j$ are the eigenvalues of $A - CB^{-1}C^\top$ and $-B$, respectively.*

The eigenvalues of $H_\tau$ become related to the definition of the strict local minimax point (2) as $\epsilon \to 0$, and this was used to show that GDA with sufficiently large timescale separation converges to a strict local minimax point in Fiez & Ratliff (2021, Theorem 1), and also in Jin et al. (2020, Theorem 28).

**Theorem 4.2** (Fiez & Ratliff (2021, Theorem 1)). *For $f \in C^2$ and its stationary point $z^*$ with $\det(\nabla_{yy}^2 f(z^*)) \neq 0$, the point $z^*$ is a strict local minimax point if and only if there exists a constant $\tau^* > 0$ such that the point $z^*$ is strict linearly stable for two-timescale GDA with any $\tau > \tau^*$.*

This seems to be complete, but this tells us nothing about when the nondegeneracy condition on $\nabla_{yy}^2 f(z^*)$ is removed. So, as our next step, we generalize Lemma 4.1, the spectral analysis of the timescaled matrix $H_\tau$, and investigate the stability of the two-timescale GDA without such nondegeneracy condition.

## 4.2 TIMESCALE SEPARATION WITHOUT THE NONDEGENERACY CONDITION ON $\nabla_{yy}^2 f$

Mimicking how we constructed the restricted Schur complement (in Definition 4 and above), we can reduce $H_\tau$ into a simpler form without changing the spectrum. More precisely, for a block diagonal matrix $Q = \mathrm{diag}\{I, P^\top\}$, the matrix $QH_\tau Q^\top = \begin{bmatrix} \frac{1}{\tau}A & \frac{1}{\tau}CP \\ -(CP)^\top & -\Delta \end{bmatrix}$ is similar to $H_\tau$, hence has the same spectrum with $H_\tau$. Thus, by replacing $C$ with $CP$ if necessary, we may assume without loss of generality that $B$ is a diagonal matrix. Moreover, with $r = \mathrm{rank}(B)$, we may further assume that there exists a diagonal $D \in \mathbb{R}^{r \times r}$ with nonzero diagonal entries where $B$ is of the form $B = \begin{bmatrix} -D & 0 \\ 0 & 0 \end{bmatrix}$. Then a subdivision $C = [C_1 \ C_2]$ arises naturally, where $C_1$ is constructed from $C$ by taking the $r$ leftmost columns, and $C_2$ takes the rest. Note that $q = \mathrm{rank}(\Gamma) = \mathrm{rank}(C_2)$.

Using these notations, we can characterize the asymptotic behaviors of the eigenvalues of $H_\tau$ when $\tau \to \infty$, as in the following theorem. This reduces to Lemma 4.1 when $r = d_2$ (and thus $q = 0$).

**Theorem 4.3.** *Let $A \in \mathbb{R}^{d_1 \times d_1}$ be a square matrix, and let $B \in \mathbb{R}^{d_2 \times d_2}$ and $C \in \mathbb{R}^{d_1 \times d_2}$ be matrices in the form that is discussed above. Assume that $f \in C^2$. Then, under Assumption 2, for $\tau \geq 1$ and $\epsilon := {}^1/_\tau$, it is possible to construct continuous functions $\lambda_j(\epsilon)$, $j = 1, \ldots, d_1 + d_2$ so that they are the $d_1 + d_2$ complex eigenvalues of $H_\tau$, with the following asymptotics as $\epsilon \to 0+$;*

*(i)* $|\lambda_j - i\sigma_j\sqrt{\epsilon}| = o(\sqrt{\epsilon})$, $\quad |\lambda_{j+d_1} + i\sigma_j\sqrt{\epsilon}| = o(\sqrt{\epsilon})$, $\qquad j = 1, \ldots, q$,

*(ii)* $|\lambda_{j+q} - \epsilon\mu_j| = o(\epsilon)$, $\qquad\qquad\qquad\qquad\qquad\qquad j = 1, \ldots, d_1 - q$,

*(iii)* $|\lambda_{j+d_1+q} - \nu_j| = o(1)$, $\qquad\qquad\qquad\qquad\qquad\qquad j = 1, \ldots, r$,

*with being nonzero whenever $\epsilon > 0$ while the remaining $d_2 - q - r$ are constantly $0$. Here, $\mu_j$ are the eigenvalues of the restricted Schur complement $S_{\mathrm{res}}(H)$, $\nu_j$ are the nonzero eigenvalues of $-B$, and $\sigma_j$ are the singular values of $C_2$.*

As we remove the nondegeneracy condition on $B$, the $d_1 - q$ eigenvalues of type (ii) are now related to the eigenvalues of the restricted Schur complement $S_{\mathrm{res}}$, rather than the Schur complement as in Lemma 4.1. This illustrates a close connection to the refined second-order necessary condition. Nevertheless, the overall form of the asymptotic behavior of the eigenvalues of types (ii) and (iii) remains unchanged from that of Lemma 4.1. What differentiates Theorem 4.3 from Lemma 4.1 the most is the existence of the unprecedented $2q$ eigenvalues of type (i).

**Example 2.** *Consider the bilinear function $f_{\mathrm{b}}(x, y) = xy$ in Example 1. A straightforward computation gives $\mathrm{spec}(H_\tau) = \{\pm i\sqrt{\epsilon}\}$. Notice that this spectrum cannot be explained by Lemma 4.1, and Theorem 4.2 does not apply. In fact, $\mathrm{spec}(-H_\tau) \not\subset \mathbb{C}_-^\circ$ for any $\tau > 0$, so from Definition 2(i) one can see that a unique (non-strict) local minimax point $0$ is not strict linearly stable for continuous-time GDA, even with timescale separation. Moreover, Definition 3 implies that the point $0$ is in fact rather a strict linearly unstable point for discrete-time two-timescale GDA.*

*On the contrary, Theorem 4.3 exactly explains why $\mathrm{spec}(H_\tau)$ is $\{\pm i\sqrt{\epsilon}\}$; this is an example where $\mathrm{spec}(H_\tau)$ contains only the eigenvalues of type (i). As we will see in Proposition 5.1 and 5.5, for an equilibrium point to be strict linearly stable for EG with timescale separation, the spectrum $\mathrm{spec}(H_\tau)$ must lie on a certain target set, which contains a deleted neighborhood of the origin in the imaginary axis. Thus, $0$ becomes strict linearly stable for two-timescale EG, as desired.*

However, the general situation in terms of the type (i) eigenvalues is of course not always as simple as those in Example 2. This necessitated us to introduce the term *hemicurvature* of the eigenvalue function $\lambda_j(\epsilon)$, as in next subsection, to help identify whether $\mathrm{spec}(H_\tau)$ lie in a certain *target* set.

### 4.2.1 HEMICURVATURE OF THE EIGENVALUE FUNCTION $\lambda_j(\epsilon)$

Let us focus on the eigenvalues of type (i), *i.e.*, $\lambda_j$ with $j \in \mathcal{I} := \{1, \ldots, q\} \cup \{d_1 + 1, \ldots, d_1 + q\}$. Identifying the complex plane with $\mathbb{R}^2$, we may consider $\lambda_j(\epsilon)$ as a plane curve $(\mathrm{Re}\,\lambda_j(\epsilon), \mathrm{Im}\,\lambda_j(\epsilon))$ parametrized by $\epsilon$. All that Theorem 4.3 tells us about $\lambda_j(\epsilon)$ is that its leading term is of order $\sqrt{\epsilon}$, which determines the tangential direction of $\lambda_j$. More precisely, $\lambda_j(\epsilon)$ converges to $0$ as $\epsilon \to 0+$ asymptotically in the direction along the imaginary axis. The complication here is that the target sets, as mentioned in Example 2, for both GDA and EG happen to have the imaginary axis also as their tangential line at the origin. That is, the tangential information (Theorem 4.3) alone is not sufficient for our stability analysis. For further details, with figures, on this issue, see Appendix E.3.

The local canonical form of curves suggests investigating the curvature information of $\lambda_j$. To this end, let us identify the coordinate functions $\mathrm{Re}(\lambda_j)$ and $\mathrm{Im}(\lambda_j)$, when $j \in \mathcal{I}$. By reindexing the singular values of $\boldsymbol{C}_2$ for notational convenience, we already know that $\mathrm{Im}(\lambda_j) = \pm i\sigma_j\sqrt{\epsilon} + o(\sqrt{\epsilon})$. To characterize $\mathrm{Re}(\lambda_j)$, by identifying $\boldsymbol{H}_\tau = \begin{bmatrix} \boldsymbol{0} & \boldsymbol{0} \\ -\boldsymbol{C}^\top & -\boldsymbol{B} \end{bmatrix} + \epsilon\begin{bmatrix} \boldsymbol{A} & \boldsymbol{C} \\ \boldsymbol{0} & \boldsymbol{0} \end{bmatrix}$ as a perturbed linear operator, one can use a classical result from perturbation theory (see, *e.g.*, Section II.1.2 of (Kato, 1995)) to expand $\lambda_j(\epsilon)$ into a convergent Puiseux series $\lambda_j(\epsilon) = \sum_{k=1}^\infty \alpha_j^{(k)} \epsilon^{k/p_j}$ for some positive integer $p_j$. Let $k_0$ be the smallest $k$ such that $\mathrm{Re}\,\alpha_j^{(k)} \neq 0$, and define $\varrho_j := k_0/p_j$, $\zeta_j := \alpha_j^{(k_0)}$, and $\xi_j := \mathrm{Re}\,\zeta_j$. Then, we may write $\mathrm{Re}(\lambda_j) = \xi_j\epsilon^{\varrho_j} + o(\epsilon^{\varrho_j})$. Notice that $\varrho_j > 1/2$. If such $k_0$ does not exist, then $\mathrm{Re}(\lambda_j) = 0$ holds, so we may simply put $\varrho_j = 1$ and $\xi_j = 0$. Now, instead of the curvature itself, we define and consider the following value, named a *hemicurvature*:

$$\iota_j := \lim_{\epsilon \to 0+} (\xi_j\epsilon^{\varrho_j - 1})/\sigma_j^2. \tag{5}$$

In Proposition E.4 we show that the hemicurvature $\iota_j$ is equal to $-1/2$ times the curvature of the trajectory of $\lambda_j$ when $\epsilon \to 0+$. The reason we consider this quantity instead of the curvature is because not only it captures the curvature information, but also it is the limit of $\mathrm{Re}(1/\lambda_j)$ as $\epsilon \to 0$; see Proposition E.5. This simplifies the subsequent analyses, as we see in Section 5.

### 4.3 TWO-TIMESCALE GDA AVOIDS SOME NON-STRICT LOCAL MINIMAX POINTS

It is obviously desirable to have a method that converges to an optimal point, regardless of whether the point is strict or non-strict. This unfortunately fails for two-timescale GDA, which almost surely avoids some non-strict local minimax points. We already saw such an instance in Example 2. More generally, consider the set $\mathcal{X}_{\mathrm{ns}}^*$ of non-strict local minimax points satisfying $\iota_j < \eta/2$ for some $j \in \mathcal{I}$. We have a negative result below that timescale separation does not help GDA to converge to such non-strict points, despite being useful in converging to *strict* local minimax points.

**Theorem 4.4.** *Let $\boldsymbol{z}^* \in \mathcal{X}_{\mathrm{ns}}^*$. Under Assumptions 1, 2 and $0 < \eta < 1/L$, there exists a constant $\tau^* > 0$ such that for any $\tau > \tau^*$, the set of initial points converging to $\boldsymbol{z}^*$ by the discrete-time two-timescale GDA $\tilde{\boldsymbol{w}}_\tau$ has Lebesgue measure zero, i.e., $\mu(\{\boldsymbol{z}_0 : \lim_{k \to \infty} \tilde{\boldsymbol{w}}_\tau^k(\boldsymbol{z}_0) = \boldsymbol{z}^*\}) = 0$.*

## 5 LOCAL MINIMAX POINTS AND THE LIMIT POINTS OF TWO-TIMESCALE EG

In this section we show that the two-timescale EG converges to a stationary point that satisfies the refined second-order necessary condition mentioned in Proposition 3.1. We first show that such points with an invertible $D\boldsymbol{F}$ are the strict linearly stable points of two-timescale EG. Then, we show that two-timescale EG almost surely avoids a strict non-minimax point, even without the invertibility assumption on $D\boldsymbol{F}$. These results are comparable to the convergence guarantees known for gradient descent on minimization problems—namely that gradient descent locally converges to local minima, and almost surely avoids strict *saddle* points (Lee et al., 2016).

Before we begin, we would like to remark that a point not being strict linearly *stable*—for example because of $D\boldsymbol{F}$ being not invertible—does not imply that the point is strict linearly *unstable*. In other words, the strict linear stability is sufficient but not necessary for a method to find that point, as will soon be detailed in Section 6.

### 5.1 TWO-TIMESCALE EG

We now consider the EG with timescale separation, to resolve the aforementioned avoidance issue of GDA. For simplicity in the analysis, we first work with the continuous-time limit of EG, especially the high-order approximation version in Lemma 2.1, so that we have a clear distinction from GDA. Applying the timescale separation on the ODE in Lemma 2.1 using $\boldsymbol{\Lambda}_\tau$ as in (4), we get

$$\dot{\boldsymbol{z}}(t) = \boldsymbol{\phi}_\tau(\boldsymbol{z}(t)) := -(\boldsymbol{I} + s\boldsymbol{\Lambda}_\tau D\boldsymbol{F}(\boldsymbol{z}(t)))^{-1}\boldsymbol{\Lambda}_\tau \boldsymbol{F}(\boldsymbol{z}(t)). \tag{6}$$

Its discretization with $s = \eta/2$, namely the two-timescale EG, becomes

$$\boldsymbol{z}_{k+1} = \boldsymbol{w}_\tau(\boldsymbol{z}_k) := \boldsymbol{z}_k - \eta\boldsymbol{\Lambda}_\tau \boldsymbol{F}(\boldsymbol{z}_k - \eta\boldsymbol{\Lambda}_\tau \boldsymbol{F}(\boldsymbol{z}_k)). \tag{7}$$

Their correspondence can be easily shown by following the proof of Lemma 2.1 with $\|\boldsymbol{\Lambda}_\tau\| \leq 1$. From now on, we will refer both of them simply as $\tau$-EG, and denote their Jacobian matrix, either

$D\phi_\tau$ or $D\boldsymbol{w}_\tau$, as $\boldsymbol{J}_\tau$. Whether they are in terms of continuous-time or discrete-time will be clear from the context. $\tau$-EG takes the stationary points of $\boldsymbol{F}$ as its equilibria, see Appendix F.

The analyses of the stability properties are based on Definition 2, so the Jacobian matrices, of the systems (6) and (7), play a central role. Hence, in this section, for convenience, the Jacobian matrix at a given equilibrium point $\boldsymbol{z}^*$ will be denoted by $\boldsymbol{J}_\tau^* \coloneqq \boldsymbol{J}_\tau(\boldsymbol{z}^*)$. Meanwhile, similar to Theorem 4.2, the upcoming stability results share a property of being held for all $\tau$ larger than a certain threshold. So for simplicity, we introduce the following notion of $\infty$-EG.

**Definition 6.** *We say that $\boldsymbol{z}^*$ is a strict linearly stable equilibrium point of $\infty$-EG if, there exists some $\tau^\star$ such that, for any $\tau > \tau^\star$ the point $\boldsymbol{z}^*$ is a strict linearly stable equilibrium point of $\tau$-EG.*

## 5.2 RELATION WITH TWO-TIMESCALE EG IN CONTINUOUS-TIME

For the stability analysis of continuous-time two-timescale EG (6) in terms of Definition 2(i), we need to examine $\mathrm{spec}(\boldsymbol{J}_\tau^*)$, but computing $\boldsymbol{J}_\tau$ in general is cumbersome. However, there is a simple characterization of $\mathrm{spec}(\boldsymbol{J}_\tau^*)$ in terms of $\mathrm{spec}(\boldsymbol{H}_\tau^*)$, where $\boldsymbol{H}_\tau^* \coloneqq \boldsymbol{H}_\tau(\boldsymbol{z}^*)$. For $s < 1/L$, define a subset of the complex plane $\overline{\mathcal{D}}_s \coloneqq \{z \in \mathbb{C} : |z + 1/2s| \leq 1/2s\}$, which is a closed disk centered at $-1/2s$ with radius $1/2s$. Then we have the following fact, implying that the stability analysis of the ODE (6) can be done by examining $\mathrm{spec}(\boldsymbol{H}_\tau^*)$. See Appendix G.2 for a visualization of this statement, with a comparison to the continuous-time GDA.

**Proposition 5.1.** *Under Assumption 1, $\mathrm{spec}(\boldsymbol{J}_\tau^*) \subset \mathbb{C}_-^\circ$ if and only if $\mathrm{spec}(\boldsymbol{H}_\tau^*) \cap \overline{\mathcal{D}}_s = \varnothing$.*

We are now ready to state our main stability result of two-timescale EG in terms of the refined second-order necessary condition. This states that under Assumption $2'$, for a step size $s$ taken not too small, a point satisfying the refined second-order necessary condition is strict linearly stable.

**Theorem 5.2.** *Suppose Assumptions 1 and $2'$. Let $s_0 \coloneqq \max_{j \in \mathcal{I}}(-\iota_j)$, for $\iota_j$ defined in (28). Then, an equilibrium point $\boldsymbol{z}^*$ satisfies $\boldsymbol{S}_{\mathrm{res}} \succeq \boldsymbol{0}$, $\boldsymbol{B} \preceq \boldsymbol{0}$, and $s_0 < 1/L$ if and only if there exists a constant $0 < s^\star < 1/L$ such that $\boldsymbol{z}^*$ is a strict linearly stable point of $\infty$-EG for any step size $s \in (s^\star, 1/L)$.*

*Proof.* Let us briefly sketch the proof, whose detail is deferred to Appendix G.4. For any $j \in \mathcal{I}$, using the hemicurvature $\iota_j$ in (5), it is possible to determine whether the eigenvalues $\lambda_j$ of $\boldsymbol{H}_\tau^*$ are in $\overline{\mathcal{D}}_s$ or not, using Lemma G.4. Then, combining this result with the behaviors of $\lambda_j$ where $j \notin \mathcal{I}$, which is established in Theorem 4.3, we can characterize exactly when does $\mathrm{spec}(\boldsymbol{H}_\tau^*) \cap \overline{\mathcal{D}}_s = \varnothing$ hold. The conclusion then follows from Proposition 5.1 and Definition 2(i). $\qquad\square$

Although choosing step size $s$ close enough to $1/L$ would suffice in practice, it is possible that $s$ may not be large enough in view of Theorem 5.2. Furthermore, the condition $s_0 < 1/L$ may not hold, and even checking such condition is challenging as one needs to compute all the hemicurvatures $\iota_j$. As a complement, we provide stability conditions that do not depend on $s_0$ (and consequently on $s^\star$).

**Theorem 5.3.** *Let $\boldsymbol{z}^*$ be an equilibrium point. Suppose Assumptions 1 and $2'$ hold.*

   *(i) (Sufficient condition) Suppose that $\boldsymbol{z}^*$ satisfies $\boldsymbol{S} \succeq \boldsymbol{0}$ and $\boldsymbol{B} \preceq \boldsymbol{0}$. Then for any step size $0 < s < 1/L$, the point $\boldsymbol{z}^*$ is a strict linearly stable point of $\infty$-EG.*

   *(ii) (Necessary condition) Suppose that $\boldsymbol{z}^*$ is a strict linearly stable equilibrium point of $\infty$-EG for any step size $0 < s < 1/L$. Then, it satisfies $\boldsymbol{S}_{\mathrm{res}} \succeq \boldsymbol{0}$ and $\boldsymbol{B} \preceq \boldsymbol{0}$.*

If we introduce a mild additional assumption that all $\sigma_j$ values are distinct, Theorem 5.3 can be refined into a tight necessary and sufficient condition. This, however, excludes local minimax points that satisfy $\boldsymbol{u}_j^\top \boldsymbol{S} \boldsymbol{u}_j < 0$ for some $j$. We leave treating such points for any given $s$ as a future work.

**Theorem 5.4.** *Under Assumptions 1 and $2'$, suppose that all $\sigma_j$ values are distinct. Then, an equilibrium $\boldsymbol{z}^*$ satisfies $\boldsymbol{S}_{\mathrm{res}} \succeq \boldsymbol{0}$, $\boldsymbol{B} \preceq \boldsymbol{0}$, and $\boldsymbol{u}_j^\top \boldsymbol{S} \boldsymbol{u}_j \geq 0$ for every left singular vector $\boldsymbol{u}_j$ of $\boldsymbol{C}_2$ if and only if $\boldsymbol{z}^*$ is a strict linearly stable point of $\infty$-EG for any step size $0 < s < 1/L$.*

## 5.3 RELATION WITH TWO-TIMESCALE EG IN DISCRETE-TIME

For the stability analysis of the discrete-time two-timescale EG (7), this time in terms of Definition 2(ii), we need to examine $\mathrm{spec}(\boldsymbol{J}_\tau^*)$, and this can be similarly characterized in terms of

$\mathrm{spec}(\boldsymbol{H}_\tau^*)$. For $\eta < 1/L$, define a peanut-shaped subset of the complex plane

$$\mathcal{P}_\eta := \big\{ z = x + iy \in \mathbb{C} : (\eta x - 1/2)^2 + \eta^2 y^2 + 3/4 < \sqrt{1 + 3\eta^2 y^2} \big\}, \tag{8}$$

containing a subset of $i\mathbb{R} \setminus \{0\}$, as shown in Appendix H.1. Then we have the following. See Appendix H.2 for a visualization of this statement, with a comparison to the discrete-time GDA.

**Proposition 5.5.** *Under Assumption 1, $\rho(\boldsymbol{J}_\tau^*) < 1$ if and only if $\mathrm{spec}(\boldsymbol{H}_\tau^*) \subset \mathcal{P}_\eta$.*

We then analyze the stability of (7), using arguments similar to the continuous-time counterparts.

**Theorem 5.6.** *Suppose Assumptions 1 and 2' hold. Then, an equilibrium point $\boldsymbol{z}^*$ satisfies $\boldsymbol{S}_{\mathrm{res}} \succeq \boldsymbol{0}$, $\boldsymbol{B} \preceq \boldsymbol{0}$, and $s_0 < 1/2L$ if and only if there exists a constant $0 < \eta^\star < 1/L$ such that $\boldsymbol{z}^*$ is a strict linearly stable point of $\infty$-EG for any step size $\eta$ satisfying $\eta^\star < \eta < 1/L$.*

In Appendix H.5, we also provide stability conditions that do not depend on $s_0$ (and consequently on $\eta^\star$), similar to the continuous-time analyses in Theorems 5.3 and 5.4.

**Example 3.** *Consider the bilinear function $f_{\mathsf{b}}(x,y) = xy$ in Example 1, and recall that $\mathrm{spec}(\boldsymbol{H}_\tau) = \{\pm i\sqrt{\epsilon}\}$. Notice that $\mathrm{spec}(\boldsymbol{H}_\tau) \cap \overline{\mathcal{D}}_s = \varnothing$ for any $s > 0$, and $\mathrm{spec}(\boldsymbol{H}_\tau) \subset \mathcal{P}_\eta$ whenever $\eta$ is sufficiently small. Therefore, as mentioned in Example 2, the local minimiax point $\boldsymbol{0}$ is strict linearly stable for both continuous- and discrete-time two-timescale EG.*

## 5.4 Two-timescale EG almost always avoids strict non-minimax points

Using Theorem 2.2, we show that two-timescale EG almost surely avoids strict non-minimax points.

**Theorem 5.7.** *Let $\boldsymbol{z}^*$ be a strict non-minimax point, i.e., $\boldsymbol{z}^* \in \mathcal{T}^*$. Under Assumptions 1, 2, and $0 < \eta < (\sqrt{5}-1)/2L$, there exists $\tau^\star > 0$ such that for any $\tau > \tau^\star$, the set of initial points that converge to $\boldsymbol{z}^*$ by two-timescale (discrete) EG $\boldsymbol{w}_\tau$ has measure zero. Moreover, if $\mathcal{T}^*$ is finite[4], then there exists $\tau^\star > 0$ such that for any $\tau > \tau^\star$, $\mu(\{\boldsymbol{z}_0 : \lim_{k\to\infty} \boldsymbol{w}_\tau^k(\boldsymbol{z}_0) \in \mathcal{T}^*\}) = 0$.*

This, however, does not mean that strict non-minimax points are always the only points that two-timescale EG almost surely escapes, as one would desire. Considering Theorems 5.6 and 5.7, this is possible if $s_0 < 1/2L$ holds and we can choose the step size so that $\eta^\star < \eta < (\sqrt{5}-1)/2L$. Nevertheless, these conditions are somewhat restrictive and we leave removing them as future work.

## 6 Two-timescale EG globally finds local minimax points

Our analysis has thus far been local, but can be extended to a global statement. One can show that any accumulation point of the iterates computed by two-timescale EG is a stationary point, for instance under the *Minty variational inequality* (MVI) condition (Minty, 1967) which encompasses nonconvex-nonconcave settings; see Appendix J. With appropriate settings on $s_0$, $\eta$, and $\mathcal{T}^*$, Theorem 5.7 guarantees almost sure avoidance of strict non-minimax points. Hence, under these settings, with Assumptions 1 and 2 and assuming that the two-timescale EG converges and there are no non-strict non-minimax points, akin to the case of gradient descent as in (Lee et al., 2016, Theorem 11), we can conclude that the two-timescale EG globally and almost surely finds local minimax points.

As a consequential result, we now have that two-timescale EG is also capable of finding local minimax points even with a singular $D\boldsymbol{F}$, *i.e.*, some $\lambda_j$ of $\boldsymbol{H}_\tau$ in Theorem 4.3 are zero, as demonstrated in the following example with non-isolated local minimax points.

**Example 4.** *Consider the quadratic function $f_{\mathsf{q}}$ in Example 1. Similar to the case of Example 2, since $\mathrm{spec}(\boldsymbol{H}_\tau) = \{0, 2, \epsilon, \pm i\sqrt{\epsilon}\}$, all (non-strict) local minimax points are strict linearly unstable for discrete-time two-timescale GDA. Yet, while $0 \in \mathrm{spec}(\boldsymbol{H}_\tau)$, both $\mathbb{C} \setminus \overline{\mathcal{D}}_s$ and $\mathcal{P}_\eta$ never contain $0$. Thus, unlike Example 3, the optima are not strict linearly stable for two-timescale EG. However, recall that a point not being strict linearly stable does not imply that the point is strict linearly unstable. Indeed, for this $f_{\mathsf{q}}$ we have $s_0 = 0$ because $\{\lambda_j : j \in \mathcal{I}\} = \{\pm i\sqrt{\epsilon}\}$, and thus $\eta^* \le 0$. Therefore, the local minimax points are not avoided by two-timescale EG. In fact, any stationary point of this $f_{\mathsf{q}}$ is a local optimal point that satisfies the MVI condition (see Lemma J.3), hence the local optimal points are rather found by discrete-time two-timescale EG.*

We defer the final discussions and a review on the related works to the appendices.

---

[4]If we assume that all points in $\mathcal{T}^*$ are isolated and the iterates are bounded, then $\mathcal{T}^*$ is essentially finite.

ACKNOWLEDGMENTS

This work was supported in part by the National Research Foundation of Korea (NRF) grant funded by the Korea government (MSIT) (No. 2019R1A5A1028324, 2022R1C1C1003940), and the Samsung Science & Technology Foundation grant (No. SSTF-BA2101-02).

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

## CONTENTS

## A    Discussions

We demonstrated that two-timescale extragradient converges to local minimax points without a non-degeneracy condition on Hessian with respect to the maximization variable, from a dynamical system viewpoint. We have then shown that two-timescale extragradient almost surely avoids strict non-minimax points, using a theorem built upon the stable manifold theorem. To the best of our knowledge, this is the first result in minimax optimization that attempts to fully characterize the stability behavior near stationary points, comparable to that of gradient descent in Lee et al. (2016).

Still, our work is just a takeoff in exploring first-order methods for finding non-strict local minimax points, leaving several unanswered questions. While our results state that they hold whenever the timescale parameter $\tau$ is sufficiently large, we do not discuss how large exactly should $\tau$ be. A slight gap remains between the second order necessary and sufficient conditions of local minimax points, due to the additional condition on $h$ given in Proposition 3.2. The analysis of the limit points of two-timescale EG involves conditions on $\boldsymbol{S}_{\mathrm{res}}$ and $s_0$. For further discussion, we advise the reader to refer to (Chae et al., 2023). Investigations on these issues will lead to interesting future works.

## B    RELATED WORKS

**Equilibria in minimax problems**    Nash equilibrium is a widely used notion of (local) optimality in minimax optimization, especially when minimax problems are interpreted as simultaneous games, where both players act simultaneously (Jin et al., 2020). However, there are problems that do not have a local Nash equilibrium point including realistic GAN applications (Farnia & Ozdaglar, 2020). In fact, the correct way of interpreting minimax problems, especially in modern machine learning applications, are as sequential games (Jin et al., 2020). These necessitated a new notion of *local* optimality that reflects the sequential nature of minimax optimization. The notion of local optimality in sequential games, built upon Stackelberg equilibrium (von Stackelberg, 2011), was first introduced by Evtushenko (1974), but this was later found to be not *truly* local (Jin et al., 2020). Instead, Fiez et al. (2020) and Jin et al. (2020) considered another notion of (true) local optimum, called *strict* local minimax points, which are exactly the points satisfying the sufficient condition of local optimum proposed by Evtushenko (1974). This, however, only covers the cases where $\nabla^2_{yy} f$ is nondegenerate, hence disregards possibly meaningful "non-strict" local optimal points, *e.g.,* in over-parameterized training (Liu et al., 2022).

So, Jin et al. (2020) also introduced a more general definition of local optimum, named local minimax points; see Section 2.2 for more details. To the best of our knowledge, all existing methods in Evtushenko (1974); Fiez et al. (2020); Wang et al. (2020); Zhang et al. (2020); Chen et al. (2023), even that in Jin et al. (2020), focus on finding only *strict* local minimax points.

**Two-timescale methods**    Plain GDA can converge to nonoptimal points (Daskalakis & Panageas, 2020; Jin et al., 2020). So, Jin et al. (2020) adopted a (computationally cheap) timescale separation into GDA, *i.e.,* using different timescales, or step sizes, for each player's action. It is proven that two-timescale GDA with a sufficiently large timescale separation converges to strict local minimax points (Jin et al., 2020; Fiez & Ratliff, 2021). Recently, two-timescale EG was studied in Zhang et al. (2022); Mahdavinia et al. (2022), but it was only shown to behave similar to the two-timescale GDA; this paper demonstrates the particular usefulness of two-timescale EG when $\nabla^2_{yy} f$ is degenerate.

## C    PROOFS AND MISSING DETAILS FOR SECTION 2

### C.1    PROOF OF LEMMA 2.1

Letting $s = \eta/2$, the ODE derived by Lu (2022), mentioned in Section 2.3, can be written as

$$\begin{aligned}
\dot{\boldsymbol{z}}(t) &= -\boldsymbol{F}(\boldsymbol{z}(t)) + \frac{\eta}{2} D\boldsymbol{F}(\boldsymbol{z}(t))\boldsymbol{F}(\boldsymbol{z}(t)) \\
&= -\big(\boldsymbol{I} - sD\boldsymbol{F}(\boldsymbol{z}(t))\big)\boldsymbol{F}(\boldsymbol{z}(t)).
\end{aligned} \tag{9}$$

As we assume that $0 < s < \frac{1}{L}$, by Assumption 1, the matrix $\boldsymbol{I} + sD\boldsymbol{F}(z)$ is invertible. Applying the approximation $(\boldsymbol{I} + sD\boldsymbol{F}(\boldsymbol{z}(t)))^{-1} = \boldsymbol{I} - sD\boldsymbol{F}(\boldsymbol{z}(t)) + O(s^2)$, the ODE

$$\dot{\boldsymbol{z}}(t) = -\big(\boldsymbol{I} + sD\boldsymbol{F}(\boldsymbol{z}(t))\big)^{-1}\boldsymbol{F}(\boldsymbol{z}(t)) \tag{10}$$

can then be seen as an approximation of (9).

### C.2    FURTHER DISCUSSIONS ON THE NEW ODE

Let $\boldsymbol{g}(t) \coloneqq \boldsymbol{F}(\boldsymbol{z}(t))$. Using the chain rule, we have

$$D\boldsymbol{F}(\boldsymbol{z}(t))\dot{\boldsymbol{z}}(t) = \dot{\boldsymbol{g}}(t).$$

Therefore, (10) is equivalent to

$$\begin{aligned}
-\boldsymbol{g}(t) = -\boldsymbol{F}(\boldsymbol{z}(t)) &= \big(\boldsymbol{I} + sD\boldsymbol{F}(\boldsymbol{z}(t))\big)\dot{\boldsymbol{z}}(t) \\
&= \dot{\boldsymbol{z}}(t) + s\dot{\boldsymbol{g}}(t).
\end{aligned} \tag{11}$$

We would like to mention that special case of (11) when $s = 1$ has originally been studied as a continuous-time limit of the Newton's method by Attouch & Svaiter (2011), and also as that of the optimistic GDA by Ryu et al. (2019).

# D  PROOFS FOR SECTION 3

## D.1  PROOF OF PROPOSITION 3.1

We use $\boldsymbol{P}$, $\boldsymbol{\Gamma}$ and $\boldsymbol{U}$ that are introduced above Definition 4. Interpreting $\boldsymbol{P}$ as a change-of-basis matrix, we have $\boldsymbol{C}^\top \boldsymbol{v} \in \mathcal{R}(\boldsymbol{B})$ if and only if the last $(d_2 - r)$ components of $\boldsymbol{P}^\top \boldsymbol{C}^\top \boldsymbol{v}$ are zero. The latter holds if and only if $\boldsymbol{v}$ is orthogonal to the last $(d_2 - r)$ columns of $\boldsymbol{CP}$, or in other words, if and only if $\boldsymbol{v} \perp \mathcal{R}(\boldsymbol{\Gamma})$.

Let $\mathcal{W} := \mathcal{R}(\boldsymbol{\Gamma})^\perp$ and $w := \dim \mathcal{W}$. If $w = 0$, then this means that the zero vector is the only vector such that $\boldsymbol{C}^\top \boldsymbol{v} \in \mathcal{R}(\boldsymbol{B})$, so there is nothing to show. Assuming $w \geq 1$, by how $\boldsymbol{U}$ is constructed, we have $\boldsymbol{v} \in \mathcal{W}$ if and only if there exists $\boldsymbol{w} \in \mathbb{R}^w$ such that $\boldsymbol{v} = \boldsymbol{Uw}$. Hence, the condition $\boldsymbol{v}^\top (\boldsymbol{A} - \boldsymbol{CB}^\dagger \boldsymbol{C}^\top) \boldsymbol{v} \geq 0$ for any $\boldsymbol{v}$ satisfying $\boldsymbol{C}^\top \boldsymbol{v} \in \mathcal{R}(\boldsymbol{B})$ is equivalent to having

$$\boldsymbol{w}^\top \boldsymbol{U}^\top (\boldsymbol{A} - \boldsymbol{CB}^\dagger \boldsymbol{C}^\top) \boldsymbol{Uw} = \boldsymbol{w}^\top \boldsymbol{S}_{\mathrm{res}} \boldsymbol{w} \geq 0$$

for any $\boldsymbol{w} \in \mathbb{R}^w$.

## D.2  PROOF OF PROPOSITION 3.2

*Proof.* Let $\boldsymbol{A} := \nabla^2_{\boldsymbol{xx}} f(\boldsymbol{x}^*, \boldsymbol{y}^*)$, $\boldsymbol{B} := \nabla^2_{\boldsymbol{yy}} f(\boldsymbol{x}^*, \boldsymbol{y}^*)$, and $\boldsymbol{C} := \nabla^2_{\boldsymbol{xy}} f(\boldsymbol{x}^*, \boldsymbol{y}^*)$. Since $\boldsymbol{y}^*$ is a local maximum of $f(\boldsymbol{x}^*, \cdot)$, it holds that $\boldsymbol{B} \preceq \boldsymbol{0}$.

Because $(\boldsymbol{x}^*, \boldsymbol{y}^*)$ is a local minimax point, there exists a function $h$ such that (1) holds. Moreover, by the assumption we made on $h$, there exist $\delta_1$ and a constant $M$ such that $\frac{h(\delta)}{\delta} \leq M$ whenever $0 < \delta \leq \delta_1$. We may then replace $\delta_0$ in Definition 1 by $\min\{\delta_0, \delta_1\}$ without affecting the local optimality. In other words, without loss of generality we may assume that

$$\frac{h(\delta)}{\delta} \leq M \quad \text{whenever} \quad 0 < \delta \leq \delta_0 \tag{12}$$

holds in Definition 1. Let us denote $h'(\delta) = \max\{h(\delta), 2\|\boldsymbol{B}^\dagger \boldsymbol{C}^\top\|\delta\}$.

Using the first order necessary condition, we have

$$f(\boldsymbol{x}^* + \boldsymbol{\delta_x}, \boldsymbol{y}^* + \boldsymbol{\delta_y}) = f(\boldsymbol{x}^*, \boldsymbol{y}^*) + \frac{1}{2}\boldsymbol{\delta_x}^\top \boldsymbol{A}\boldsymbol{\delta_x} + \boldsymbol{\delta_x}^\top \boldsymbol{C}\boldsymbol{\delta_y} + \frac{1}{2}\boldsymbol{\delta_y}^\top \boldsymbol{B}\boldsymbol{\delta_y} + R(\boldsymbol{\delta_x}, \boldsymbol{\delta_y}) \tag{13}$$

where $R(\boldsymbol{\delta_x}, \boldsymbol{\delta_y})$ is the remainder term of a degree 2 Taylor series expansion. Now let us assume that $\|\boldsymbol{\delta_x}\| = \delta \in (0, \delta_0]$ and $\boldsymbol{C}^\top \boldsymbol{\delta_x} \in \mathcal{R}(\boldsymbol{B})$. Then, by the second order necessary assumption $\boldsymbol{B} \preceq \boldsymbol{0}$, it holds that

$$-\boldsymbol{B}^\dagger \boldsymbol{C}^\top \boldsymbol{\delta_x} = \operatorname*{argmax}_{\boldsymbol{\delta_y}} \ f(\boldsymbol{x}^*, \boldsymbol{y}^*) + \frac{1}{2}\boldsymbol{\delta_x}^\top \boldsymbol{A}\boldsymbol{\delta_x} + \boldsymbol{\delta_x}^\top \boldsymbol{C}\boldsymbol{\delta_y} + \frac{1}{2}\boldsymbol{\delta_y}^\top \boldsymbol{B}\boldsymbol{\delta_y}.$$

Meanwhile, let $\epsilon > 0$ be arbitrary. Then, noticing that

$$R(\boldsymbol{\delta_x}, \boldsymbol{\delta_y}) = o(\|\boldsymbol{\delta_x}\|^2 + \|\boldsymbol{\delta_y}\|^2),$$

we know that there exists $\delta_2 > 0$ such that

$$\|\boldsymbol{\delta_x}\|^2 + \|\boldsymbol{\delta_y}\|^2 < \delta_2 \quad \Longrightarrow \quad \left| \frac{R(\boldsymbol{\delta_x}, \boldsymbol{\delta_y})}{\|\boldsymbol{\delta_x}\|^2 + \|\boldsymbol{\delta_y}\|^2} \right| < \epsilon.$$

Here, if $\|\boldsymbol{\delta_y}\| \leq h(\delta)$, then by (12) and that $\delta \leq \delta_0$ we have

$$\frac{\|\boldsymbol{\delta_x}\|^2 + \|\boldsymbol{\delta_y}\|^2}{\|\boldsymbol{\delta_x}\|^2} = 1 + \frac{\|\boldsymbol{\delta_y}\|^2}{\|\boldsymbol{\delta_x}\|^2} \leq 1 + \left( \frac{h(\delta)}{\delta} \right)^2 \leq 1 + M^2 \tag{14}$$

which then implies

$$\left| \frac{R(\boldsymbol{\delta_x}, \boldsymbol{\delta_y})}{\|\boldsymbol{\delta_x}\|^2} \right| = \frac{\|\boldsymbol{\delta_x}\|^2 + \|\boldsymbol{\delta_y}\|^2}{\|\boldsymbol{\delta_x}\|^2} \left| \frac{R(\boldsymbol{\delta_x}, \boldsymbol{\delta_y})}{\|\boldsymbol{\delta_x}\|^2 + \|\boldsymbol{\delta_y}\|^2} \right| \leq (1 + M^2)\epsilon. \tag{15}$$

Note that, by (14), the condition $\|\boldsymbol{\delta_x}\|^2 + \|\boldsymbol{\delta_y}\|^2 < \delta_2$ holds whenever $\|\boldsymbol{\delta_x}\| \leq \frac{\delta_2}{\sqrt{1+M^2}}$. Thus, since $\epsilon$ was chosen arbitrarily, we conclude that $R(\boldsymbol{\delta_x}, \boldsymbol{\delta_y}) = o(\|\boldsymbol{\delta_x}\|^2)$, as long as $\|\boldsymbol{\delta_y}\| \leq h(\delta)$.

Therefore, again invoking that $(\boldsymbol{x}^*, \boldsymbol{y}^*)$ is a local minimax point, we have

$$
\begin{aligned}
0 &\leq \max_{\|\boldsymbol{\delta_y}\| \leq h(\delta)} f(\boldsymbol{x}^* + \boldsymbol{\delta_x}, \boldsymbol{y}^* + \boldsymbol{\delta_y}) - f(\boldsymbol{x}^*, \boldsymbol{y}^*) \\
&\leq \max_{\|\boldsymbol{\delta_y}\| \leq h(\delta)} \left( \frac{1}{2}\boldsymbol{\delta_x}^\top \boldsymbol{A}\boldsymbol{\delta_x} + \boldsymbol{\delta_x}^\top \boldsymbol{C}\boldsymbol{\delta_y} + \frac{1}{2}\boldsymbol{\delta_y}^\top \boldsymbol{B}\boldsymbol{\delta_y} \right) + \max_{\|\boldsymbol{\delta_y}\| \leq h(\delta)} R(\boldsymbol{\delta_x}, \boldsymbol{\delta_y}) \\
&\leq \max_{\|\boldsymbol{\delta_y}\| \leq h'(\delta)} \left( \frac{1}{2}\boldsymbol{\delta_x}^\top \boldsymbol{A}\boldsymbol{\delta_x} + \boldsymbol{\delta_x}^\top \boldsymbol{C}\boldsymbol{\delta_y} + \frac{1}{2}\boldsymbol{\delta_y}^\top \boldsymbol{B}\boldsymbol{\delta_y} \right) + o(\|\boldsymbol{\delta_x}\|^2) \\
&= \frac{1}{2}\boldsymbol{\delta_x}^\top (\boldsymbol{A} - \boldsymbol{C}\boldsymbol{B}^\dagger \boldsymbol{C}^\top)\boldsymbol{\delta_x} + o(\|\boldsymbol{\delta_x}\|^2).
\end{aligned}
$$

This inequality holds for any $\boldsymbol{\delta_x}$ satisfying $\boldsymbol{C}^\top \boldsymbol{\delta_x} \in \mathcal{R}(B)$, so we are done. $\qquad \square$

### D.3 PROOF OF EXAMPLE 1

- $f_{\mathsf{b}}(x, y) = xy$: The saddle-gradient of $f_{\mathsf{b}}$ is $\boldsymbol{F}(x, y) = (x, -y)$, where $(x^*, y^*) = (0, 0)$ is a unique stationary point. Since $f_{\mathsf{b}}(x^*, y) = f_{\mathsf{b}}(x^*, y^*) = f_{\mathsf{b}}(x, y^*) = 0$ for any $(x, y)$, we can say that $(0, 0)$ is a unique local minimax point by definition.

  Regarding Assumption 2, the Jacobian matrix of the saddle-gradient $\boldsymbol{F}$ is

  $$
  D\boldsymbol{F} = \left[ \begin{array}{c|c} \boldsymbol{A} & \boldsymbol{C} \\ -\boldsymbol{C}^\top & -\boldsymbol{B} \end{array} \right] = \left[ \begin{array}{c|c} 0 & 1 \\ \hline -1 & 0 \end{array} \right],
  $$

  where $\boldsymbol{B}$ is degenerate with $r = \mathrm{rank}(\boldsymbol{B}) = 0$. Since $d_1 = d_2 = 1$, $\boldsymbol{\Gamma} = [1] \in \mathbb{R}^{1 \times 1}$ and $q = \mathrm{rank}(\boldsymbol{\Gamma}) = 1$, the matrix $\boldsymbol{U}$ is of size $1 \times 0$. So, $\boldsymbol{S}_{\mathrm{res}}(D\boldsymbol{F})$ is then a $0 \times 0$ matrix. A linear map from a zero dimensional vector space to a zero dimensional vector space is trivially an identity map, so the corresponding $0 \times 0$ matrix is *trivially* nondegenerate.

- $f_{\mathsf{q}}(x_1, x_2, y_1, y_2, y_3) = \frac{1}{2}x_1^2 - \frac{1}{2}y_1^2 - \frac{1}{2}y_3^2 + x_2y_2 + y_1y_3$: The saddle-gradient of $f_{\mathsf{q}}$ is $\boldsymbol{F}(x_1, x_2, y_1, y_2, y_3) = (x_1, y_2, y_1 - y_3, -x_2, y_3 - y_1)$, where $\{t \in \mathbb{R} : (0, 0, t, 0, t)\}$ is a set of stationary points. Since the inequality

  $$
  f_{\mathsf{q}}(\boldsymbol{x}^*, \boldsymbol{y}) = -\frac{1}{2}(y_1 - y_3)^2 \leq f_{\mathsf{q}}(\boldsymbol{x}^*, \boldsymbol{y}^*) = 0 \leq f_{\mathsf{q}}(\boldsymbol{x}, \boldsymbol{y}^*) = \frac{1}{2}x_1^2
  $$

  holds for any $(\boldsymbol{x}^*, \boldsymbol{y}^*) \in \{t \in \mathbb{R} : (0, 0, t, 0, t)\}$ and for any $(\boldsymbol{x}, \boldsymbol{y})$, the set of local minimax points is by definition $\{t \in \mathbb{R} : (0, 0, t, 0, t)\}$.

  Regarding Assumption 2, the Jacobian matrix of the saddle-gradient $\boldsymbol{F}$ is

  $$
  D\boldsymbol{F} = \left[ \begin{array}{c|c} \boldsymbol{A} & \boldsymbol{C} \\ -\boldsymbol{C}^\top & -\boldsymbol{B} \end{array} \right] = \left[ \begin{array}{cc|ccc} 1 & 0 & 0 & 0 & 0 \\ 0 & 0 & 0 & 1 & 0 \\ \hline 0 & 0 & 1 & 0 & -1 \\ 0 & -1 & 0 & 0 & 0 \\ 0 & 0 & -1 & 0 & 1 \end{array} \right],
  $$

  where $\boldsymbol{B}$ is degenerate with $r = \mathrm{rank}(\boldsymbol{B}) = 2$. Since $d_1 = 2$, $d_2 = 3$, $\boldsymbol{\Gamma} = \left[ \begin{smallmatrix} 0 \\ 1 \end{smallmatrix} \right] \in \mathbb{R}^{2 \times 1}$, and $q = \mathrm{rank}(\boldsymbol{\Gamma}) = 1$, we have $\boldsymbol{U} = \left[ \begin{smallmatrix} 1 \\ 0 \end{smallmatrix} \right] \in \mathbb{R}^{2 \times 1}$, and thus $\boldsymbol{S}_{\mathrm{res}}(D\boldsymbol{F}) = [1] \in \mathbb{R}^{1 \times 1}$ is nondegenerate. Notice that for this example, the classical generalized Schur complement $\boldsymbol{S} = \boldsymbol{A} - \boldsymbol{C}\boldsymbol{B}^\dagger \boldsymbol{C}^\top = \left[ \begin{smallmatrix} 1 & 0 \\ 0 & 0 \end{smallmatrix} \right] \in \mathbb{R}^{2 \times 2}$ is degenerate.

## E PROOFS AND MISSING DETAILS FOR SECTION 4

### E.1 PROOF OF THEOREM 4.3

This theorem exactly reduces to Lemma 4.1 when $\nabla_{\boldsymbol{yy}}^2 f(\boldsymbol{z}^*)$ is nondegenerate, it is only left to show this theorem for the case where $\boldsymbol{S}_{\mathrm{res}}(D\boldsymbol{F}(\boldsymbol{z}^*))$ is nondegenerate.

We begin with the following observation. Suppose that restricted Schur complement $S_{\text{res}}$ is invertible. Then, one can show that $DF$ is similar to

$$G = \begin{bmatrix} A & C_1 & C_2 \\ -C_1^\top & D & 0 \\ -C_2^\top & 0 & 0 \end{bmatrix},$$

but $C_2$ may not be of full column rank. Let $\text{rank}(C_2) := q$, and let $C_2 = U\Sigma V^\top$ be the (full) singular value decomposition of $C_2$ where for some invertible diagonal matrix $\Sigma_q$ it holds that

$$\Sigma = \begin{bmatrix} \Sigma_q & 0 \\ 0 & 0 \end{bmatrix}.$$

Then, by defining $L_q = U\begin{bmatrix} \Sigma_q \\ 0 \end{bmatrix}$, notice that $U\Sigma = \begin{bmatrix} L_q & 0 \end{bmatrix}$. Thus, for $\tilde{Q} = \text{diag}\{I, I, V^\top\}$ we have

$$\tilde{Q}G\tilde{Q}^\top = \begin{bmatrix} A & C_1 & C_2 V \\ -C_1^\top & D & 0 \\ -V^\top C_2^\top & 0 & 0 \end{bmatrix} = \begin{bmatrix} A & C_1 & L_q & 0 \\ -C_1^\top & D & 0 & 0 \\ -L_q^\top & 0 & 0 & 0 \\ 0 & 0 & 0 & 0 \end{bmatrix}$$

and therefore

$$\det(\lambda I - DF) = \det(\lambda I - \tilde{Q}G\tilde{Q}^\top) = \lambda^{d_2 - r - q}\det\left(\lambda I - \begin{bmatrix} A & C_1 & L_q \\ -C_1^\top & D & 0 \\ -L_q^\top & 0 & 0 \end{bmatrix}\right).$$

So, the eigenvalues of $DF$ are either $0$ or the eigenvalues of

$$\Phi := \begin{bmatrix} A & C_1 & L_q \\ -C_1^\top & D & 0 \\ -L_q^\top & 0 & 0 \end{bmatrix}, \tag{16}$$

and analogously the eigenvalues of $H_\tau$ are either $0$ or the eigenvalues of $\Phi_\tau := \Lambda_\tau \Phi$.

Moreover, by how $L_q$ is defined, it holds that the singular values of $L_q$ are exactly the singular values of $C_2$, and $\mathcal{R}(L_q) = \mathcal{R}(C_2)$. In particular, there is no essential change in $S_{\text{res}}(H)$ when $C_2$ is replaced by $L_q$. Also, notice that $L_q$ is of full column rank.

Therefore, characterizing the eigenvalues of $\Phi_\tau$ is equivalent to characterizing the nonzero eigenvalues of $H_\tau$.

To prove Theorem 4.3 we need the following result, which shows a relation between the eigenvalues of the restricted Schur complement and a determinantal equation that resembles the eigenvalue equation of $\Phi_\tau$.

**Lemma E.1.** *A (possibly complex) number $\mu$ is an eigenvalue of the restricted Schur complement if and only if it is a root of the equation*

$$\det\begin{bmatrix} \mu I - A & -C_1 & -L_q \\ C_1^\top & -D & 0 \\ L_q^\top & 0 & 0 \end{bmatrix} = 0. \tag{17}$$

*Proof.* By the property of Moore-Penrose pseudoinverses, it holds that

$$CB^\dagger C^\top = \begin{bmatrix} C_1 & C_2 \end{bmatrix}\begin{bmatrix} -D^{-1} & 0 \\ 0 & 0 \end{bmatrix}\begin{bmatrix} C_1^\top \\ C_2^\top \end{bmatrix} = -C_1 D^{-1} C_1^\top.$$

Multiplying matrices of determinant 1 does not change the determinant, so by observing

$$\begin{bmatrix} I & -C_1 D^{-1} & 0 \\ 0 & I & 0 \\ 0 & 0 & I \end{bmatrix} \begin{bmatrix} \mu I - A & -C_1 & -L_q \\ C_1^\top & -D & 0 \\ L_q^\top & 0 & 0 \end{bmatrix} \begin{bmatrix} I & 0 & 0 \\ D^{-1} C_1^\top & I & 0 \\ 0 & 0 & I \end{bmatrix}$$

$$= \begin{bmatrix} \mu I - A - C_1 D^{-1} C_1^\top & 0 & -L_q \\ 0 & -D & 0 \\ L_q^\top & 0 & 0 \end{bmatrix}$$

$$= \begin{bmatrix} \mu I - A + C B^\dagger C^\top & 0 & -L_q \\ 0 & -D & 0 \\ L_q^\top & 0 & 0 \end{bmatrix}$$

$$= \begin{bmatrix} \mu I - S & 0 & -L_q \\ 0 & -D & 0 \\ L_q^\top & 0 & 0 \end{bmatrix}$$

we get that (17) is equivalent to the determinantal equation $\det T = 0$ where

$$T := \begin{bmatrix} \mu I - S & 0 & -L_q \\ 0 & -D & 0 \\ L_q^\top & 0 & 0 \end{bmatrix}. \tag{18}$$

Since permuting the rows and the columns change the determinant only up to a sign, we have

$$|\det T| = \left| \det \begin{bmatrix} \mu I - S & 0 & -L_q \\ 0 & -D & 0 \\ L_q^\top & 0 & 0 \end{bmatrix} \right|$$

$$= \left| \det \begin{bmatrix} \mu I - S & 0 & -L_q \\ L_q^\top & 0 & 0 \\ 0 & -D & 0 \end{bmatrix} \right|$$

$$= \left| \det \begin{bmatrix} \mu I - S & -L_q & 0 \\ L_q^\top & 0 & 0 \\ 0 & 0 & -D \end{bmatrix} \right|.$$

As $D$ is invertible by assumption, we obtain that $\det T = 0$ is further equivalent to

$$\det \begin{bmatrix} \mu I - S & -L_q \\ L_q^\top & 0 \end{bmatrix} = 0. \tag{19}$$

Now we choose an orthogonal matrix $Q$ such that it has a block matrix form

$$Q = \begin{bmatrix} U_1 & U_2 \end{bmatrix}$$

where $U_2$ is the $d_1 \times q$ matrix of left singular vectors appearing in the reduced singular value decomposition of $L_q = U_2 \Sigma_q V_2^\top$, and $U_1$ is consequently a $d_1 \times (d_1 - q)$ matrix with the columns consisting an orthonormal basis of $\mathcal{R}(L_q)^\perp$. Note that $U_1$ corresponds to the matrix $U$ that appears in the restricted Schur complement, and considering how $L_q$ is defined, we may take $V_2 = I$. Applying a similarity transformation, we have

$$\begin{bmatrix} Q^\top & 0 \\ 0 & I \end{bmatrix} \begin{bmatrix} \mu I - S & -L_q \\ L_q^\top & 0 \end{bmatrix} \begin{bmatrix} Q & 0 \\ 0 & I \end{bmatrix} = \begin{bmatrix} \mu I - Q^\top S Q & -Q^\top L_q \\ L_q^\top Q & 0 \end{bmatrix}. \tag{20}$$

Exploiting the block structure of $U$, observe that

$$Q^\top L_q = \begin{bmatrix} U_1^\top \\ U_2^\top \end{bmatrix} L_q = \begin{bmatrix} U_1^\top L_q \\ U_2^\top L_q \end{bmatrix} = \begin{bmatrix} 0 \\ \Sigma_q V_2^\top \end{bmatrix} = \begin{bmatrix} 0 \\ \Sigma_q \end{bmatrix}$$

by the orthogonality of $\mathcal{R}(U_1)$ and $\mathcal{R}(L_q)$, and

$$Q^\top S Q = \begin{bmatrix} U_1^\top \\ U_2^\top \end{bmatrix} S \begin{bmatrix} U_1 & U_2 \end{bmatrix} = \begin{bmatrix} U_1^\top S U_1 & U_1^\top S U_2 \\ U_2^\top S U_1 & U_2^\top S U_2 \end{bmatrix}.$$

Thus, the matrix in (20) can be also expressed as

$$M := \begin{bmatrix} \mu I - U_1^\top S U_1 & -U_1^\top S U_2 & 0 \\ -U_2^\top S U_1 & \mu I - U_2^\top S U_2 & -\Sigma_q \\ 0 & \Sigma_q & 0 \end{bmatrix}.$$

Similarity transformations preserve determinants, so (19) holds if and only if $\det M = 0$. Using again that the fact that permuting the rows and the columns change the determinant only up to a sign, we obtain

$$|\det M| = \left| \det \begin{bmatrix} \mu I - U_1^\top S U_1 & -U_1^\top S U_2 & 0 \\ -U_2^\top S U_1 & \mu I - U_2^\top S U_2 & -\Sigma_q \\ 0 & \Sigma_q & 0 \end{bmatrix} \right|$$

$$= \left| \det \begin{bmatrix} 0 & \Sigma_q & 0 \\ \mu I - U_1^\top S U_1 & -U_1^\top S U_2 & 0 \\ -U_2^\top S U_1 & \mu I - U_2^\top S U_2 & -\Sigma_q \end{bmatrix} \right|$$

$$= \left| \det \begin{bmatrix} \Sigma_q & 0 & 0 \\ -U_1^\top S U_2 & \mu I - U_1^\top S U_1 & 0 \\ \mu I - U_2^\top S U_2 & -U_2^\top S U_1 & -\Sigma_q \end{bmatrix} \right|.$$

Notice that both $\mu I - U_1^\top S U_1$ and $\Sigma_q$ are all square matrices, hence

$$\det M = \det(\mu I - U_1^\top S U_1) \det(\Sigma_q) \det(-\Sigma_q).$$

Because $L_q$ is of full rank, we know that $\Sigma_q$ is invertible. Therefore, $\det M = 0$ if and only if $\det(\mu I - U_1^\top S U_1) = 0$.

Combining all of the chain of equivalent equations established, we conclude that

$$\det \begin{bmatrix} \mu I - A & -C_1 & -L_q \\ C_1^\top & -D & 0 \\ L_q^\top & 0 & 0 \end{bmatrix} = 0 \quad \text{if and only if} \quad \det(\mu I - U_1^\top S U_1) = 0. \tag{21}$$

Noting that $U_1^\top S U_1$ is the restricted Schur complement, we are done. $\qquad\square$

The following is an immediate corollary of the previous theorem.

**Corollary E.2.** *Suppose that the restricted Schur complement is invertible. Then equation* (17) *does not have $\mu = 0$ as a solution. In particular, $\Phi$ is invertible.*

*Proof.* The first part directly follows from the equivalence (21). For the second part, because substituting $\mu = 0$ in (17) does not make the equation hold, we must have $\det(-\Phi) \neq 0$. $\qquad\square$

We also use the following lemma, which tells us that the roots of a polynomial is continuous with respect to its coefficients. For the proof, see the original reference.

**Lemma E.3** (Zedek (1965, Theorem 1)). *Given a polynomial $p_n(z) := \sum_{k=0}^n a_k z^k$, $a_n \neq 0$, an integer $m \geq n$ and a number $\epsilon > 0$, there exists a number $\delta > 0$ such that whenever the $m + 1$ complex numbers $b_k$, $0 \leq k \leq m$, satisfy the inequalities*

$$|b_k - a_k| < \delta \quad \text{for} \quad 0 \leq k \leq n, \quad \text{and} \quad |b_k| < \delta \quad \text{for} \quad n+1 \leq k \leq m,$$

*then the roots $\beta_k$, $1 \leq k \leq m$, of the polynomial $q_m(z) := \sum_{k=0}^m b_k z^k$ can be labeled in such a way as to satisfy, with respect to the zeros $\alpha_k$, $1 \leq k \leq n$, of $p_n(z)$, the inequalities*

$$|\beta_k - \alpha_k| < \epsilon \quad \text{for} \quad 1 \leq k \leq n, \quad \text{and} \quad |\beta_k| > \frac{1}{\epsilon} \quad \text{for} \quad n+1 \leq k \leq m.$$

*Proof of Theorem 4.3.* The eigenvalues of $\boldsymbol{\Phi}_\tau$ are the solutions of the equation

$$0 = p_\epsilon(\lambda) := \det \begin{bmatrix} \lambda\boldsymbol{I} - \epsilon\boldsymbol{A} & -\epsilon\boldsymbol{C}_1 & -\epsilon\boldsymbol{L}_q \\ \boldsymbol{C}_1^\top & \lambda\boldsymbol{I} - \boldsymbol{D} & \boldsymbol{0} \\ \boldsymbol{L}_q^\top & \boldsymbol{0} & \lambda\boldsymbol{I} \end{bmatrix} = \det(\lambda\boldsymbol{I} - \boldsymbol{\Phi}_\tau). \tag{22}$$

By Lemma E.3, constructing the functions $\lambda_j(\epsilon)$ so that they are continuous is possible. Also by that lemma, letting $\epsilon \to 0$, we see that the eigenvalues converges to the solutions of the equation

$$p_0(\lambda) = \det \begin{bmatrix} \lambda\boldsymbol{I} & \boldsymbol{0} & \boldsymbol{0} \\ \boldsymbol{C}_1^\top & \lambda\boldsymbol{I} - \boldsymbol{D} & \boldsymbol{0} \\ \boldsymbol{L}_q^\top & \boldsymbol{0} & \lambda\boldsymbol{I} \end{bmatrix} = 0.$$

Hence, the $r$ eigenvalues of $\boldsymbol{H}_\tau$ converges to the $r$ nonzero eigenvalues of $-\boldsymbol{B}$, and the other $d_1 + q$ eigenvalues converges to zero, as $\epsilon \to 0$.

To investigate the eigenvalues that converges to zero further, we begin by observing that, whenever $|\lambda|$ is small enough so that $\lambda\boldsymbol{I} - \boldsymbol{D}$ is invertible, it holds that

$$\det \begin{bmatrix} \lambda\boldsymbol{I} - \epsilon\boldsymbol{A} & -\epsilon\boldsymbol{C}_1 & -\epsilon\boldsymbol{L}_q \\ \boldsymbol{C}_1^\top & \lambda\boldsymbol{I} - \boldsymbol{D} & \boldsymbol{0} \\ \boldsymbol{L}_q^\top & \boldsymbol{0} & \lambda\boldsymbol{I} \end{bmatrix} = \det \begin{bmatrix} \lambda\boldsymbol{I} - \epsilon\boldsymbol{A} + \epsilon\boldsymbol{C}_1(\lambda\boldsymbol{I} - \boldsymbol{D})^{-1}\boldsymbol{C}_1^\top & \boldsymbol{0} & -\epsilon\boldsymbol{L}_q \\ \boldsymbol{0} & \lambda\boldsymbol{I} - \boldsymbol{D} & \boldsymbol{0} \\ \boldsymbol{L}_q^\top & \boldsymbol{0} & \lambda\boldsymbol{I} \end{bmatrix}$$

$$= \det(\lambda\boldsymbol{I} - \boldsymbol{D}) \det \begin{bmatrix} \lambda\boldsymbol{I} - \epsilon\big(\boldsymbol{A} - \boldsymbol{C}_1(\lambda\boldsymbol{I} - \boldsymbol{D})^{-1}\boldsymbol{C}_1^\top\big) & -\epsilon\boldsymbol{L}_q \\ \boldsymbol{L}_q^\top & \lambda\boldsymbol{I} \end{bmatrix}.$$

That is, any $\lambda_j$ that converges to zero as $\epsilon \to 0$ is a solution of the equation

$$0 = \det \begin{bmatrix} \lambda\boldsymbol{I} - \epsilon\big(\boldsymbol{A} - \boldsymbol{C}_1(\lambda\boldsymbol{I} - \boldsymbol{D})^{-1}\boldsymbol{C}_1^\top\big) & -\epsilon\boldsymbol{L}_q \\ \boldsymbol{L}_q^\top & \lambda\boldsymbol{I} \end{bmatrix}. \tag{23}$$

Now let us reparametrize (23) by $\lambda = \kappa\sqrt{\epsilon}$ to get

$$0 = \det \begin{bmatrix} \kappa\sqrt{\epsilon}\boldsymbol{I} - \epsilon\big(\boldsymbol{A} - \boldsymbol{C}_1(\lambda\boldsymbol{I} - \boldsymbol{D})^{-1}\boldsymbol{C}_1^\top\big) & -\epsilon\boldsymbol{L}_q \\ \boldsymbol{L}_q^\top & \kappa\sqrt{\epsilon}\boldsymbol{I} \end{bmatrix}$$

$$= \sqrt{\epsilon}^{d_1} \det \begin{bmatrix} \kappa\boldsymbol{I} - \sqrt{\epsilon}\big(\boldsymbol{A} - \boldsymbol{C}_1(\lambda\boldsymbol{I} - \boldsymbol{D})^{-1}\boldsymbol{C}_1^\top\big) & -\sqrt{\epsilon}\boldsymbol{L}_q \\ \boldsymbol{L}_q^\top & \kappa\sqrt{\epsilon}\boldsymbol{I} \end{bmatrix}$$

$$= \sqrt{\epsilon}^{d_1+d_2} \det \begin{bmatrix} \kappa\boldsymbol{I} - \sqrt{\epsilon}\big(\boldsymbol{A} - \boldsymbol{C}_1(\lambda\boldsymbol{I} - \boldsymbol{D})^{-1}\boldsymbol{C}_1^\top\big) & -\boldsymbol{L}_q \\ \boldsymbol{L}_q^\top & \kappa\boldsymbol{I} \end{bmatrix}.$$

Since $(\lambda\boldsymbol{I} - \boldsymbol{D})^{-1}$ converges as $\epsilon \to 0$, we have that if $\lambda_j \to 0$ as $\epsilon \to 0$ then $\lambda_j$ should be a solution of the equation

$$0 = \det \begin{bmatrix} \kappa\boldsymbol{I} - \sqrt{\epsilon}\big(\boldsymbol{A} - \boldsymbol{C}_1(\lambda\boldsymbol{I} - \boldsymbol{D})^{-1}\boldsymbol{C}_1^\top\big) & -\boldsymbol{L}_q \\ \boldsymbol{L}_q^\top & \kappa\boldsymbol{I} \end{bmatrix} \tag{24}$$

By Lemma E.3, we see that such eigenvalues divided by $\sqrt{\epsilon}$ converge to the roots of

$$0 = \det \begin{bmatrix} \kappa\boldsymbol{I} & -\boldsymbol{L}_q \\ \boldsymbol{L}_q^\top & \kappa\boldsymbol{I} \end{bmatrix}. \tag{25}$$

Following the first few statements in the proof of Lemma E.1, we get that $\boldsymbol{L}_q$ must be of full column rank. This implies that $\boldsymbol{L}_q$ has exactly $q$ singular values. Thus, the nonzero solutions of (25) are exactly $\pm i\sigma_k$, $k = 1,\ldots,q$ where $\sigma_k$ are the singular values of $\boldsymbol{L}_q$, and therefore there are $2q$ instances among $\lambda_j$ where each $\lambda_j/\sqrt{\epsilon}$ converges to each one of $\pm i\sigma_k$ as $\epsilon \to 0$.

So far, we have shown that $r$ eigenvalues have magnitude $\Theta(1)$, and $2q$ eigenvalues have magnitude $\Theta(\sqrt{\epsilon})$. Meanwhile, because we assume that the restricted Schur complement is invertible, from

$$\det \boldsymbol{\Phi}_\tau = \epsilon^{d_1} \det \begin{bmatrix} \boldsymbol{A} & \boldsymbol{C}_1 & \boldsymbol{L}_q \\ -\boldsymbol{C}_1^\top & \boldsymbol{D} & \boldsymbol{0} \\ -\boldsymbol{L}_q^\top & \boldsymbol{0} & \boldsymbol{0} \end{bmatrix}$$

and that the determinant on the right hand side is nonzero by Corollary E.2, we know that the product of all $\lambda_j$ should be of order $\Theta(\epsilon^{d_1})$. From these two observations, one can deduce that the product of the remaining $d_1 - q$ eigenvalues should be of order $\Theta(\epsilon^{d_1-q})$. We claim that these $d_1 - q$ eigenvalues are all exactly of order $\Theta(\epsilon)$. To this end, let us examine what properties would the eigenvalues of order $O(\epsilon)$ have. Reparametrizing $\lambda = \mu\epsilon$ in (22), we get

$$
0 = \begin{bmatrix} \mu\epsilon\boldsymbol{I} - \epsilon\boldsymbol{A} & -\epsilon\boldsymbol{C}_1 & -\epsilon\boldsymbol{L}_q \\ \boldsymbol{C}_1^\top & \mu\epsilon\boldsymbol{I} - \boldsymbol{D} & \boldsymbol{0} \\ \boldsymbol{L}_q^\top & \boldsymbol{0} & \mu\epsilon\boldsymbol{I} \end{bmatrix}
$$
$$
= \epsilon^{d_1} \det \begin{bmatrix} \mu\boldsymbol{I} - \boldsymbol{A} & -\boldsymbol{C}_1 & -\boldsymbol{L}_q \\ \boldsymbol{C}_1^\top & \mu\epsilon\boldsymbol{I} - \boldsymbol{D} & \boldsymbol{0} \\ \boldsymbol{L}_q^\top & \boldsymbol{0} & \mu\epsilon\boldsymbol{I} \end{bmatrix}.
$$

Then, by Lemma E.3, $\mu$ converges to a root of the equation

$$
0 = \det \begin{bmatrix} \mu\boldsymbol{I} - \boldsymbol{A} & \boldsymbol{C}_1 & -\boldsymbol{L}_q \\ \boldsymbol{C}_1^\top & -\boldsymbol{D} & \boldsymbol{0} \\ \boldsymbol{L}_q^\top & \boldsymbol{0} & \boldsymbol{0} \end{bmatrix}. \tag{26}
$$

Again by Corollary E.2, $\mu = 0$ cannot be a root of (26). This in particular implies that no $\lambda_j$ can be of order $o(\epsilon)$, or in other words, all eigenvalues of $\boldsymbol{H}_\tau$ are of order $\Omega(\epsilon)$. Therefore, for a product of $d_1 - q$ eigenvalues of $\boldsymbol{\Phi}_\tau$ to be of order $\Theta(\epsilon^{d_1-q})$, each of those eigenvalues must be of order exactly $\Theta(\epsilon)$, and the claim follows.

Notice that the discussions previous paragraph further imply the following two facts:

- If $\lambda$ is an eigenvalue of order $O(\epsilon)$ then $\lambda/\epsilon$ converges to a root of (26) as $\epsilon \to 0$.

- The right hand side of (26) is a polynomial of degree $d_1 - q$ in $\mu$, whose roots are nonzero.

It is now immediate that the $d_1 - q$ eigenvalues of $\boldsymbol{\Phi}_\tau$ that are of order $\Theta(\epsilon)$ is of the form $\lambda(\epsilon) = \mu\epsilon + o(\epsilon)$ for a (nonzero) root $\mu$ of (26).

The claim that $\lambda_j(\epsilon) \neq 0$ whenever $\epsilon > 0$ for any of the $d_1 + q + r$ eigenvalues that are not constantly zero is a direct consequence of the simple fact $\det(\boldsymbol{\Phi}_\tau) = \det(\boldsymbol{\Lambda}_\tau)\det(\boldsymbol{\Phi}) \neq 0$. □

### E.2 PROOFS ON THE PROPERTIES OF THE HEMICURVATURE

In Section 4.2.1, for $\lambda_j(\epsilon)$ with

$$
\begin{aligned}
\mathrm{Re}(\lambda_j(\epsilon)) &= \xi_j\epsilon^{\varrho_j} + o(\epsilon^{\varrho_j}), \\
\mathrm{Im}(\lambda_j(\epsilon)) &= \sigma_j\sqrt{\epsilon} + o(\sqrt{\epsilon}),
\end{aligned} \tag{27}
$$

and $\varrho_j > 1/2$, we defined its *hemicurvature* to be the quantity

$$
\iota_j := \lim_{\epsilon \to 0+} \frac{\xi_j\epsilon^{\varrho_j-1}}{\sigma_j^2}.
$$

As briefly mentioned therein, the term is named after the following observation.

**Proposition E.4.** *Let* $\alpha(\epsilon) = (\mathrm{Re}\,\lambda_j(\epsilon), \mathrm{Im}\,\lambda_j(\epsilon))$ *be a plane curve whose components are given as* (27). *Let* $\kappa(\epsilon)$ *be the (signed) curvature of* $\alpha$, *then it holds that*

$$
\iota_j = -\frac{1}{2}\lim_{\epsilon \to 0+}\kappa(\epsilon).
$$

*Proof.* For convenience, let us denote $x(\epsilon) = \mathrm{Re}(\lambda_j(\epsilon))$ and $y(\epsilon) = \mathrm{Im}(\lambda_j(\epsilon))$. Let us also omit the subscript $j$, as there is no confusion by doing so. We use the well known formula (do Carmo, 2016, p. 27)

$$
\kappa = \frac{x'y'' - x''y'}{((x')^2 + (y')^2)^{3/2}}
$$

to compute the curvature of a plane curve. To this end, observe that

$$x'(\epsilon) = \xi\varrho\epsilon^{\varrho-1} + o(\epsilon^{\varrho-1}), \qquad\qquad x''(\epsilon) = \xi\rho(\rho-1)\epsilon^{\varrho-2} + o(\epsilon^{\varrho-2}),$$
$$y'(\epsilon) = \frac{1}{2}\sigma\epsilon^{-1/2} + o(\epsilon^{-1/2}), \qquad\qquad y''(\epsilon) = -\frac{1}{4}\sigma\epsilon^{-3/2} + o(\epsilon^{-3/2}).$$

Substituting, and using the fact that $\varrho > 1/2$ hence $2\varrho - 2 > -1$, we get

$$\lim_{\epsilon\to 0+} \kappa(\epsilon) = \lim_{\epsilon\to 0+} \frac{-\frac{1}{4}\xi\varrho\sigma\epsilon^{\varrho-5/2} - \frac{1}{2}\xi\varrho(\varrho-1)\sigma\epsilon^{\varrho-5/2} + o(\epsilon^{\varrho-5/2})}{\left(\xi^2\varrho^2\epsilon^{2\varrho-2} + \frac{1}{4}\sigma^2\epsilon^{-1} + o(\epsilon^{-1})\right)^{3/2}}$$

$$= \lim_{\epsilon\to 0+} \frac{-2\xi\varrho(2\varrho-1)\sigma\epsilon^{\varrho-1} + o(\epsilon^{\varrho-1})}{\left(4\xi^2\varrho^2\epsilon^{2\varrho-1} + \sigma^2 + o(1)\right)^{3/2}}$$

$$= -2\varrho(2\varrho-1) \lim_{\epsilon\to 0+} \frac{\xi\epsilon^{\varrho-1}}{\sigma^2}.$$

If $1/2 < \varrho < 1$, then the limit is $\infty$, so the constant factor $\varrho(2\varrho - 1)$ does not affect the limit. Similarly, if $\varrho > 1$, then the limit is 0, so the constant factor $\varrho(2\varrho - 1)$ does not affect the limit also. Finally, if $\varrho = 1$, then $\varrho(2\varrho - 1) = 1$, so we are done. $\qquad\square$

In Section 4.2.1, we have also mentioned that one noteworthy property of the hemicurvature is that it is related to $\mathrm{Re}(1/\lambda_j)$. A precise statement on this is as follows.

**Proposition E.5.** *For $\lambda_j(\epsilon)$ as in (27), it holds that*

$$\iota_j = \lim_{\epsilon\to 0+} \mathrm{Re}\left(\frac{1}{\lambda_j(\epsilon)}\right). \tag{28}$$

*Proof.* We use the simple fact that if $x, y \in \mathbb{R}$ then

$$\mathrm{Re}\left(\frac{1}{x+iy}\right) = \mathrm{Re}\left(\frac{x-iy}{x^2+y^2}\right) = \frac{x}{x^2+y^2}. \tag{29}$$

Substituting (27) into the above leads to

$$\lim_{\epsilon\to 0+} \mathrm{Re}\left(\frac{1}{\lambda_j(\epsilon)}\right) = \lim_{\epsilon\to 0+} \frac{\mathrm{Re}\,\lambda_j(\epsilon)}{(\mathrm{Re}\,\lambda_j(\epsilon))^2 + (\mathrm{Im}\,\lambda_j(\epsilon))^2}$$

$$= \lim_{\epsilon\to 0+} \frac{\xi_j\epsilon^{\varrho_j} + o(\epsilon^{\varrho_j})}{\sigma_j^2\epsilon + o(\epsilon)}$$

$$= \lim_{\epsilon\to 0+} \frac{\xi_j\epsilon^{\varrho_j-1}}{\sigma_j^2} = \iota_j$$

where in the second line we use the fact that $\varrho > 1/2$ implies $\epsilon^{2\varrho} = o(\epsilon)$. $\qquad\square$

### E.3  ON THE NECESSITY OF CONSIDERING THE HEMICURVATURE

Let us present a depiction on why the tangential information alone may not be sufficient for our stability analyses: see Figure 1. Consider two disks $\overline{\mathcal{D}}_{s/2}$ and $\overline{\mathcal{D}}_s$ on the complex plane, both tangent to the imaginary axis at 0, but with different radii; $\frac{1}{s}$ and $\frac{1}{2s}$ respectively. The respective boundaries $\partial\overline{\mathcal{D}}_{s/2}$ and $\partial\overline{\mathcal{D}}_s$ are circles, so assuming without loss of generality that they are positively oriented, it is straightforward that the hemicurvatures of each boundary curves are each $-\frac{s}{2}$ and $-s$, respectively. Now suppose that some $\lambda_j(\epsilon)$ satisfies (27) with $\varrho_j = 1$ and $\frac{\xi_j}{\sigma_j^2} = -\frac{3s}{4}$. The shape of the trajectory of such $\lambda_j$ will be as drawn in the figure, and in particular, this trajectory is also tangent to the imaginary axis at 0. As $\epsilon \to 0+$ so that $\lambda_j(\epsilon) \to 0$, on one hand, we can visually see that the inclusion $\lambda_j(\epsilon) \in \overline{\mathcal{D}}_{s/2}$ eventually becomes true. On the other hand, although the fact that $\overline{\mathcal{D}}_{s/2}$ and $\overline{\mathcal{D}}_s$ share the same tangent line at 0 implies that they are indistinguishable locally at 0 if only the tangential information is used, somewhat interestingly it holds that $\lambda_j(\epsilon) \notin \overline{\mathcal{D}}_s$ for all (sufficiently small) $\epsilon > 0$.

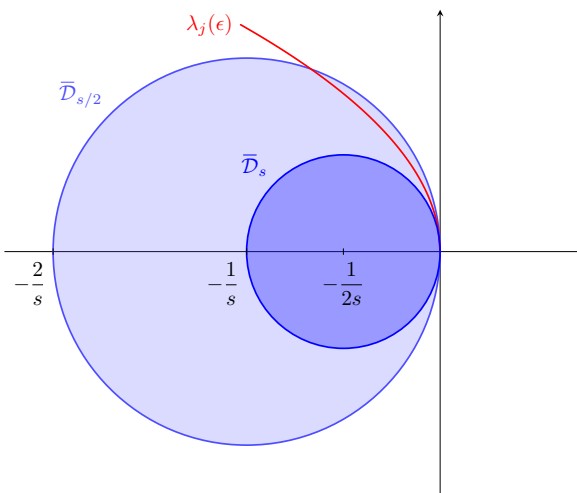

Figure 1: Two disks and a curve, all tangent to the same line at the same point.

### E.4  PROOF OF THEOREM 4.4

To prove Theorem 4.4 we need the following results.

**Proposition E.6.** *Under Assumption 1 and $0 < \eta < \frac{1}{L}$, then $\det(D\tilde{w}_\tau(z)) \neq 0$ for all $z$.*

*Proof.* Assumption 1 implies that any eigenvalue $\lambda$ of $DF(z)$ satisfies $|\lambda| \leq L$. So, assuming $0 < \eta < \frac{1}{L}$, any eigenvalue $\mu$ of $D\tilde{w}_\tau(z) = I - \eta\Lambda_\tau DF(z)$ satisfies $0 < |\mu| < 2$. Hence, we have $\det(D\tilde{w}_\tau(z)) > 0$. $\qquad\square$

**Proposition E.7.** *For any $z^* \in \mathcal{X}_{\mathrm{ns}}^*$, there exists a positive constant $\tau^\star > 0$ such that $z^* \in \mathcal{A}^*(\tilde{w}_\tau)$ for any $\tau > \tau^\star$.*

*Proof.* Since $D\tilde{w}_\tau = I - \eta H_\tau$, we have

$$
\begin{aligned}
\mathcal{A}^*(\tilde{w}_\tau) &= \{z^* \ : \ z^* = \tilde{w}_\tau(z^*), \ \rho(D\tilde{w}_\tau(z^*)) > 1\} \\
&= \{z^* \ : \ z^* = \tilde{w}_\tau(z^*), \ \exists\lambda \in \mathrm{spec}(H_\tau(z^*)) \ \text{s.t.} \ |1 - \eta\lambda| > 1\}.
\end{aligned}
$$

Observing that $\{z \in \mathbb{C} \ : \ |1 - \eta z| > 1\}$ can be equivalently formulated as $\{z \in \mathbb{C} \ : \ \mathrm{Re}\left(\frac{1}{z}\right) < \frac{\eta}{2}\}$, there exists some $\tau^\star > 0$ such that $z^* \in \mathcal{A}^*(\tilde{w}_\tau)$ for any $\tau > \tau^\star$, since $z^* \in \mathcal{X}_{\mathrm{ns}}^*$ satisfies $\lim_{\epsilon\to 0+} \mathrm{Re}(1/\lambda_j) = \iota_j < \frac{\eta}{2}$ for some $j \in \mathcal{I}$. $\qquad\square$

*Proof of Theorem 4.4.* Notice that $z^* \in \mathcal{A}^*(\tilde{w}_\tau)$ implies

$$
\{z_0 : \lim_{k\to\infty} w^k(z_0) = z^*\} \subset \{z_0 : \lim_{k\to\infty} w^k(z_0) \in \mathcal{A}^*(\tilde{w}_\tau)\}.
$$

Therefore, it follows from Theorem 2.2 that there exists a positive constant $\tau^\star > 0$ such that $\mu(\{z_0 : \lim_{k\to\infty} w^k(z_0) = z^*\}) = 0$ for any $\tau > \tau^\star$.

$\qquad\square$

### F  PROOF FOR SECTION 5.1

**Proposition F.1.** *Under Assumption 1, a point $z^*$ is an equilibrium point of (6) for $0 < s < {}^1\!/_L$, or respectively of (7) for $0 < \eta < {}^1\!/_L$, if and only if $F(z^*) = 0$.*

*Proof.* The "if" part is straightforward from both equations (6) and (7). To show the reverse direction, suppose that $z^*$ is an equilibrium point. For the case (6), we must have

$$
(I + s\Lambda_\tau DF(z^*))^{-1}\Lambda_\tau F(z^*) = 0.
$$

Under Assumption 1 and $0 < s < \frac{1}{L}$, $(\boldsymbol{I} + s\boldsymbol{\Lambda}_\tau D\boldsymbol{F}(\boldsymbol{z}^*))^{-1}$ is invertible, this is equivalent to $\boldsymbol{F}(\boldsymbol{z}^*) = \boldsymbol{0}$. For the case (7), we must have

$$\boldsymbol{F}(\boldsymbol{z}^* - \eta\boldsymbol{\Lambda}_\tau\boldsymbol{F}(\boldsymbol{z}^*)) = \boldsymbol{0}.$$

For the sake of contradiction, suppose that $\boldsymbol{F}(\boldsymbol{z}^*) \neq \boldsymbol{0}$, and let $\boldsymbol{w}^* := \boldsymbol{z}^* - \eta\boldsymbol{\Lambda}_\tau\boldsymbol{F}(\boldsymbol{z}^*)$. Note that $\boldsymbol{w}^* \neq \boldsymbol{z}^*$ by assumption. Then,

$$\begin{aligned}
\boldsymbol{z}^* - \eta\boldsymbol{\Lambda}_\tau\boldsymbol{F}(\boldsymbol{z}^*) &= \boldsymbol{z}^* - \eta\boldsymbol{\Lambda}_\tau\boldsymbol{F}(\boldsymbol{z}^* - \eta\boldsymbol{\Lambda}_\tau\boldsymbol{F}(\boldsymbol{z}^*)) - \eta\boldsymbol{\Lambda}_\tau\boldsymbol{F}(\boldsymbol{z}^*) \\
&= \boldsymbol{w}^* - \eta\boldsymbol{\Lambda}_\tau\boldsymbol{F}(\boldsymbol{w}^*),
\end{aligned}$$

hence we have $\boldsymbol{z}^* - \boldsymbol{w}^* = \eta\boldsymbol{\Lambda}_\tau(\boldsymbol{F}(\boldsymbol{z}^*) - \boldsymbol{F}(\boldsymbol{w}^*))$. Under Assumption 1, which is equivalent to the $L$-Lipschitz continuity of $\boldsymbol{F}$, and $0 < \eta < \frac{1}{L}$, we have

$$\begin{aligned}
\|\boldsymbol{z}^* - \boldsymbol{w}^*\| &= \eta\|\boldsymbol{\Lambda}_\tau(\boldsymbol{F}(\boldsymbol{z}^*) - \boldsymbol{F}(\boldsymbol{w}^*))\| \\
&\leq \eta L \|\boldsymbol{\Lambda}_\tau\|\|\boldsymbol{z}^* - \boldsymbol{w}^*\| \\
&\leq \eta L \|\boldsymbol{z}^* - \boldsymbol{w}^*\| \\
&< \|\boldsymbol{z}^* - \boldsymbol{w}^*\|
\end{aligned}$$

which is absurd. Therefore, we conclude that $\boldsymbol{F}(\boldsymbol{z}^*) = \boldsymbol{0}$. $\qquad\square$

## G   PROOFS AND MISSING DETAILS FOR SECTION 5.2

### G.1   PROOF OF PROPOSITION 5.1

The following lemmata will be used in the proof.

**Lemma G.1.** *At an equilibrium point $\boldsymbol{z}^*$ of ODE (6), it holds that*

$$\boldsymbol{J}_\tau(\boldsymbol{z}^*) = -(\boldsymbol{I} + s\boldsymbol{H}_\tau^*)^{-1}\boldsymbol{H}_\tau^*.$$

*Proof.* Differentiating the right hand side of (6) with respect to $z_i$, one gets that the $i^{\text{th}}$ column of the Jacobian matrix $\boldsymbol{J}_\tau$ is equal to

$$\begin{aligned}
[\boldsymbol{J}_\tau]_{:,i} &= \frac{d}{dz_i}\left(-(\boldsymbol{I} + s\boldsymbol{\Lambda}_\tau D\boldsymbol{F}(\boldsymbol{z}))^{-1}\boldsymbol{\Lambda}_\tau\boldsymbol{F}(\boldsymbol{z})\right) \\
&= -(\boldsymbol{I} + s\boldsymbol{\Lambda}_\tau D\boldsymbol{F}(\boldsymbol{z}))^{-1}\left(\boldsymbol{\Lambda}_\tau D_i\boldsymbol{F}(\boldsymbol{z}) - \left(s\boldsymbol{\Lambda}_\tau\frac{d}{dz_i}D\boldsymbol{F}(\boldsymbol{z})\right)(\boldsymbol{I} + s\boldsymbol{\Lambda}_\tau D\boldsymbol{F}(\boldsymbol{z}))^{-1}\boldsymbol{\Lambda}_\tau\boldsymbol{F}(\boldsymbol{z})\right)
\end{aligned}$$

where $D_i\boldsymbol{F}$ denotes the $i^{\text{th}}$ partial derivative of $\boldsymbol{F}$. Recall that $\boldsymbol{F}(\boldsymbol{z}^*) = \boldsymbol{0}$ by Proposition F.1. So, when we evaluate the expression above at $\boldsymbol{z}^*$, the last term vanishes, giving us what corresponds to the $i^{\text{th}}$ column of the matrix identity

$$\boldsymbol{J}_\tau(\boldsymbol{z}^*) = -(\boldsymbol{I} + s\boldsymbol{\Lambda}_\tau D\boldsymbol{F}(\boldsymbol{z}^*))^{-1}\boldsymbol{\Lambda}_\tau D\boldsymbol{F}(\boldsymbol{z}^*).$$

Note that this is exactly the claimed. $\qquad\square$

**Lemma G.2.** *A complex number $\lambda \in \mathbb{C} \setminus \{-\frac{1}{s}\}$ an eigenvalue of $\boldsymbol{H}_\tau^*$ if and only if $\mu = -\frac{\lambda}{1+s\lambda}$ is an eigenvalue of $\boldsymbol{J}_\tau^*$.*

*Proof.* The "only if" part is a direct consequence of applying Theorem 1.13 in (Higham, 2008) on the function $\lambda \mapsto -(1 + s\lambda)^{-1}\lambda$. For the "if" part, note that $\mu \neq -\frac{1}{s}$ as long as $\lambda \in \mathbb{C}$, hence $\boldsymbol{I} + s\boldsymbol{J}_\tau^*$ is always invertible. The assertion then follows from the fact that the map $X \mapsto -(1 + sX)^{-1}X$ is an involution. $\qquad\square$

*Proof of Proposition 5.1.* It is clear that $f : \mathbb{C} \cup \{\infty\} \to \mathbb{C} \cup \{\infty\}$ defined as $f(\lambda) = -\frac{\lambda}{1+s\lambda}$ is continuous. Also, it is an involution, hence a bijection. Therefore, $f$ is a homeomorphism.

Let us identify the image of the imaginary axis $i\mathbb{R}$ under the map $f$. Let $t \in \mathbb{R}$ be any real number, then observe that

$$f(it) + \frac{1}{2s} = \frac{-it}{1 + ist} + \frac{1}{2s} = \frac{i + st}{2s(i - st)}.$$

As $st \in \mathbb{R}$, it follows that

$$\left| f(it) + \frac{1}{2s} \right| = \frac{1}{2s} \qquad \forall t \in \mathbb{R}.$$

Moreover, $\lim_{t \to \infty} f(it) = -\frac{1}{s}$. This shows that $f(i\mathbb{R} \cup \{\infty\}) = \partial \overline{\mathcal{D}}_s$, where $\partial \overline{\mathcal{D}}_s$ denotes the boundary of $\overline{\mathcal{D}}_s$, a circle of radius $\frac{1}{2s}$ centered at $-\frac{1}{2s}$ in the complex plane.

In $\mathbb{C} \setminus i\mathbb{R}$ we have two connected components $\mathbb{C}_-^\circ$ and $\mathbb{C}_+^\circ$, and in $f(\mathbb{C} \setminus i\mathbb{R})$ we have two connected components $\overline{\mathcal{D}}_s \setminus \partial \overline{\mathcal{D}}_s$ and $\mathbb{C} \setminus \overline{\mathcal{D}}_s$. To see which is mapped to which, notice that

$$f(1) = -\frac{1}{1+s}$$

and thus $-\frac{1}{s} < f(1) < 0$. It follows that $f(1) \in \overline{\mathcal{D}}_s \setminus \partial \overline{\mathcal{D}}_s$, and therefore,

$$f(\mathbb{C}_-^\circ) = \mathbb{C} \setminus \overline{\mathcal{D}}_s,$$
$$f(\mathbb{C}_+^\circ) = \overline{\mathcal{D}}_s \setminus \partial \overline{\mathcal{D}}_s.$$

In particular, $f$ is a bijective mapping between $\mathbb{C}_-^\circ$ and $\mathbb{C} \setminus \overline{\mathcal{D}}_s$. So, by Lemma G.2, the spectrum of $\boldsymbol{J}_\tau^*$ lies inside the open left half plane if and only if $\mathrm{spec}(\boldsymbol{H}_\tau^*) \subset \mathbb{C} \setminus \overline{\mathcal{D}}_s$, which is equivalent to the claimed statement. $\qquad \square$

We would like to remark that $\mathrm{spec}(D\boldsymbol{F}) \cap \overline{\mathcal{D}}_s = \varnothing$, appearing in the statement of Proposition 5.1, is equivalent to the so-called *negative comonotonicity* condition on $\boldsymbol{F}$, which is a condition that guarantees the convergence of (discrete) EG, as demonstrated by Gorbunov et al. (2023). While their discussions rely on algebraic manipulations, our work offers a dynamical systems perspective that sheds light on the significance of this condition.

### G.2    TARGET SETS OF CONTINUOUS-TIME METHODS

Using Proposition 5.1, we can depict the *target sets* of two-timescale EG. By *target sets* we mean the region in the complex plane that the spectrum of $\mathrm{spec}(\boldsymbol{H}_\tau^*)$ must be contained in, in order to make $\mathrm{spec}(\boldsymbol{J}_\tau^*) \subset \mathbb{C}_-^\circ$ hold. See Figure 2a for a depiction. To additionally demonstrate how the target set of $\tau$-EG depends on the parameter $s$, we overlaid two target sets on the same plot.

For continuous-time $\tau$-GDA, as studied by Fiez & Ratliff (2021), the target set is the open right half plane. To contrast this set to the set in the case of $\tau$-EG, we depicted the target set for $\tau$-GDA in Figure 2b. An interesting thing to note here is that the target set of $\tau$-GDA can be seen as the (set-theoretic) limit of the target set of $\tau$-EG when $s \to 0+$, since the limit of $\mathcal{D}_s$ as $s \to 0+$ is the open left half plane. (Here we consider $\overline{\mathcal{D}}_s$ as the closure of $\mathcal{D}_s$, where the latter denotes the open disk obtained by taking the interior of $\overline{\mathcal{D}}_s$.)

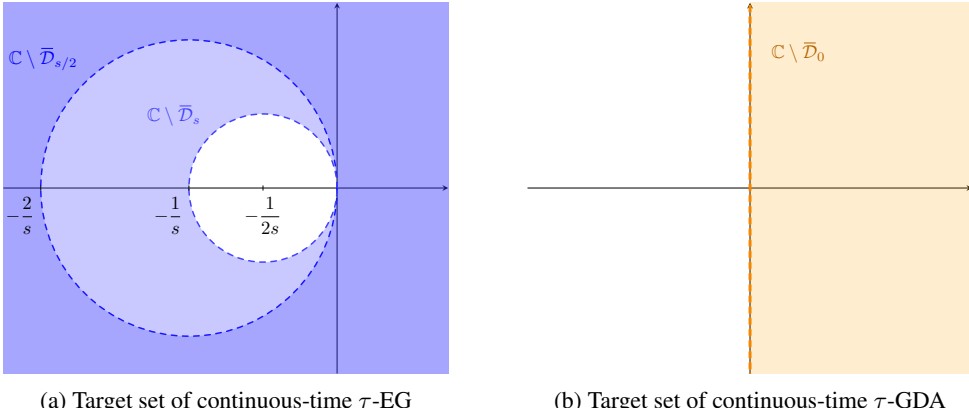

(a) Target set of continuous-time $\tau$-EG            (b) Target set of continuous-time $\tau$-GDA

Figure 2: A depiction of the target sets of continuous two-timescale methods. Notice that for $\tau$-EG we have overlaid two target sets with different choices of $s$ on the same plot.

### G.3 A CONSEQUENCE OF ASSUMPTION $2'$

As all of the remaining theorems in Section 5.2 employ Assumption $2'$, let us begin with studying what do we get by additionally assuming the invertibility of $D\boldsymbol{F}$.

**Proposition G.3.** *Suppose that Assumption $2'$ holds. Then $\boldsymbol{S}_{\mathrm{res}}(D\boldsymbol{F}(\boldsymbol{z}^*))$ is necessarily invertible.*

*Proof.* Because Assumption $2'$ implies Assumption 2, either one of $\boldsymbol{S}_{\mathrm{res}}(D\boldsymbol{F}(\boldsymbol{z}^*))$ or $\nabla^2_{\boldsymbol{yy}}f(\boldsymbol{z}^*)$ is nondegenerate. For the sake of contradiction, suppose that $\boldsymbol{S}_{\mathrm{res}}(D\boldsymbol{F}(\boldsymbol{z}^*))$ is singular, which asserts that $\boldsymbol{B} = \nabla^2_{\boldsymbol{yy}}f(\boldsymbol{z}^*)$ is nondegenerate. As $D\boldsymbol{F}$ is invertible, a classical result on Schur complements tells us that $\boldsymbol{S} = \boldsymbol{A} - \boldsymbol{C}\boldsymbol{B}^{-1}\boldsymbol{C}^\top$ is also invertible; see, *e.g.*, (Horn & Johnson, 2012, Section 0.8.5). Meanwhile, if $\boldsymbol{B}$ is invertible then $r = d_2$, so $\boldsymbol{\Gamma}$ is a matrix with zero columns. Hence, $\mathcal{R}(\boldsymbol{\Gamma})^\perp = \mathbb{R}^{d_2}$, and so the matrix $\boldsymbol{U}$ that defines the restricted Schur complement through the relation $\boldsymbol{S}_{\mathrm{res}}(D\boldsymbol{F}) = \boldsymbol{U}^\top \boldsymbol{S}\boldsymbol{U}$ becomes a square orthogonal matrix, which is in particular invertible. However, this implies that the restricted Schur complement $\boldsymbol{S}_{\mathrm{res}}(D\boldsymbol{F}(\boldsymbol{z}^*))$ is a product of invertible matrices, which cannot be singular. Therefore, our hypothesis must be false, and $\boldsymbol{S}_{\mathrm{res}}(D\boldsymbol{F}(\boldsymbol{z}^*))$ must be invertible. $\qquad\square$

Thus, assuming the invertibility of $D\boldsymbol{F}$ ensures that all $\mu_j$ are nonzero in Theorem 4.3. Moreover, because $\boldsymbol{H}_\tau = \boldsymbol{\Lambda}_\tau D\boldsymbol{F}$ is then also invertible hence cannot have a 0 as its eigenvalue, we must have $d_2 - q - r = 0$, and all eigenvalues of $\boldsymbol{H}_\tau$ fall into one of type (i), (ii), (iii) eigenvalues characterized in Theorem 4.3.

### G.4 PROOF OF THEOREM 5.2

**Lemma G.4.** *Let $s_0 := \max_{j\in\mathcal{I}}(-\iota_j)$. If $s > s_0$, then there exists some $\tau^\star > 0$ such that whenever $\tau > \tau^\star$, all the eigenvalues $\lambda_j$ with $j \in \mathcal{I}$ lie outside of the disk $\overline{\mathcal{D}}_s$. Conversely, if $s < s_0$, then for any sufficiently large $\tau$, there exists some $j \in \mathcal{I}$ such that $\lambda_j(\epsilon) \in \overline{\mathcal{D}}_s$.*

*Proof.* Fix any $j \in \{1, \ldots, d_2 - r\}$. Observing that $\overline{\mathcal{D}}_s$ can be equivalently formulated as

$$\overline{\mathcal{D}}_s = \left\{ z \in \mathbb{C} : \mathrm{Re}\left(\frac{1}{z}\right) \leq -s \right\} \tag{30}$$

we may alternatively analyze how the real part of $1/\lambda_j$ behaves. By Proposition E.5, it holds that

$$\lim_{\epsilon \to 0+} \mathrm{Re}\left(\frac{1}{\lambda_j}\right) = \iota_j. \tag{31}$$

For $j \in \{d_1 + 1, \ldots, d_1 + d_2 - r\}$, by simply replacing $\sigma_j$ by $-\sigma_j$, we also have (31).

As we define $s_0 = \max_{j\in\mathcal{I}}(-\iota_j)$, we equivalently have $-s_0 = \min_{j\in\mathcal{I}} \iota_j$. Hence, if $-s < -s_0$, then for all $j \in \mathcal{I}$, the quantity $\mathrm{Re}(1/\lambda_j(\epsilon))$ will eventually be larger than $-s$ as $\tau \to \infty$. That is, in view of (30), there exists some $\tau^\star > 0$ such that whenever $\tau > \tau^\star$, all the eigenvalues $\lambda_j$ with $j \in \mathcal{I}$ lie outside of the disk $\overline{\mathcal{D}}_s$.

Conversely, if $-s > -s_0$, then because $\mathcal{I}$ is a finite set, there exists some $j \in \mathcal{I}$ such that $\mathrm{Re}(1/\lambda_j(\epsilon)) \to -s_0$ as $\tau \to \infty$. In other words, for such $j$, whenever $\tau$ is sufficiently large it holds that $\mathrm{Re}(1/\lambda_j(\epsilon)) < -s$, or equivalently, $\lambda_j(\epsilon) \in \overline{\mathcal{D}}_s$, according to (30). $\qquad\square$

**Remark G.5.** *If $\xi_j \geq 0$ for all $j \in \mathcal{I}$, then we have $s_0 \leq 0$. In that case, since we always assume that $s > 0$, we trivially have $s > s_0$. On the other hand, if $\varrho_j < 1$ and $\xi_j < 0$ for some $j \in \mathcal{I}$, then $\iota_j = -\infty$, so as a consequence $s_0 = \infty$. In that case, since $s$ is finite, we trivially have $s < s_0$.*

*Proof of Theorem 5.2.* We analyze the necessary and sufficient condition for each set of eigenvalues $\lambda_j$ of $\boldsymbol{H}_\tau^*$ categorized in Theorem 4.3 to lie outside $\overline{\mathcal{D}}_s$ below, and combining them leads to the claimed statement.

(i) Consider $\lambda_j$ with $j \in \mathcal{I}$. Suppose that $s_0 < \frac{1}{L}$. Then, by Lemma G.4, for any step size $s$ such that $s_0 < s < \frac{1}{L}$, it holds that $\lambda_j(\epsilon) \notin \overline{\mathcal{D}}_s$ whenever $\tau$ is sufficiently large. Conversely, suppose that $s_0 \geq \frac{1}{L}$. Then, for any $s$ such that $0 < s < \frac{1}{L}$, we have $s < s_0$. In that case, by Lemma G.4, for any sufficiently large timescale $\tau > 0$ there exists some $j \in \mathcal{I}$ such that $\lambda_j(\epsilon) \in \overline{\mathcal{D}}_s$. Therefore, $s_0 < \frac{1}{L}$ if and only if there exists some $0 < s^\star < \frac{1}{L}$ such that, for any $s$ satisfying $s^\star < s < \frac{1}{L}$, $\tau$ being sufficiently large implies $\lambda_j(\epsilon) \notin \overline{\mathcal{D}}_s$ for all $j \in \mathcal{I}$.

(ii) Consider $\lambda_j = \epsilon\mu_k + o(\epsilon)$ for some $k$. Note that they converge to zero asymptotically along the real axis. In particular, whenever $\epsilon$ is sufficiently small, if $\mu_k > 0$ then $\lambda_j(\epsilon) \notin \overline{\mathcal{D}}_s$, and if $\mu_k < 0$ then $\lambda_j(\epsilon) \in \overline{\mathcal{D}}_s$. Recall that in Theorem 4.3 we have established that $\mu_k$ are the eigenvalues of the restricted Schur complement $\boldsymbol{S}_{\mathrm{res}}$. One can then deduce that $\boldsymbol{S}_{\mathrm{res}} \succeq \boldsymbol{0}$ if and only if there exists some $\tau^\star$ where $\tau > \tau^\star$ implies that every $\lambda_j$ of order $\Theta(\epsilon)$ satisfies $\lambda_j(\epsilon) \notin \overline{\mathcal{D}}_s$.

(iii) Consider $\lambda_j = \nu_k + o(1)$ for some $k$. As we take Assumption 1, for any $\tau > 1$ it holds that

$$\|\boldsymbol{H}_\tau^*\| = \|\boldsymbol{\Lambda}_\tau D\boldsymbol{F}(\boldsymbol{z}^*)\| \leq \|\boldsymbol{\Lambda}_\tau\| \|D\boldsymbol{F}(\boldsymbol{z}^*)\| \leq L.$$

As $\lambda_j(\epsilon)$ is an eigenvalue of $\boldsymbol{H}_\tau^*$, it follows that $|\lambda_j| \leq L$. Similarly, for $\boldsymbol{I}_{d_2}$ denoting the $d_2 \times d_2$ identity matrix, observe that

$$\|\boldsymbol{B}\| = \left\|\begin{bmatrix} \boldsymbol{0} & \boldsymbol{0} \\ \boldsymbol{0} & \boldsymbol{B} \end{bmatrix}\right\| = \left\|\begin{bmatrix} \boldsymbol{0} & \boldsymbol{0} \\ \boldsymbol{0} & \boldsymbol{I}_{d_2} \end{bmatrix}\begin{bmatrix} \boldsymbol{A} & \boldsymbol{C} \\ -\boldsymbol{C}^\top & \boldsymbol{B} \end{bmatrix}\begin{bmatrix} \boldsymbol{0} & \boldsymbol{0} \\ \boldsymbol{0} & \boldsymbol{I}_{d_2} \end{bmatrix}\right\| \leq \|D\boldsymbol{F}(\boldsymbol{z}^*)\| = L.$$

It follows that $|\nu_k| \leq L$ for the eigenvalues $\nu_k$ of $-\boldsymbol{B}$. Now, let $0 < s < \frac{1}{L}$, and suppose that $\boldsymbol{B} \npreceq \boldsymbol{0}$, so that there exists some $k$ such that $-L \leq \nu_k < 0$. Then, since the half-open interval $[-L, 0)$ is contained in the interior of $\overline{\mathcal{D}}_s$, for sufficiently large $\tau$ we will have $\lambda_j(\epsilon) \in \overline{\mathcal{D}}_s$. Conversely, if $\boldsymbol{B} \preceq \boldsymbol{0}$, then $\nu_k > 0$ for all $k$, so as a consequence, whenever $s > 0$ and $\tau$ is sufficiently large, we will have $\lambda_j(\epsilon) \notin \overline{\mathcal{D}}_s$ for all $j$ such that $\lambda_j(\epsilon) = \nu_k + o(1)$. Therefore, $\boldsymbol{B} \preceq \boldsymbol{0}$ if and only if there exists some $0 < s < \frac{1}{L}$ such that $\tau$ being sufficiently large implies every $\lambda_j$ of order $\Theta(1)$ satisfies $\lambda_j(\epsilon) \notin \overline{\mathcal{D}}_s$.

Combining all the discussions made, we conclude that $\boldsymbol{S}_{\mathrm{res}} \succeq \boldsymbol{0}$, $\boldsymbol{B} \preceq \boldsymbol{0}$, and $s_0 < \frac{1}{L}$ if and only if there exists some $0 < s^\star < \frac{1}{L}$ such that, for any $s$ satisfying $s^\star < s < \frac{1}{L}$, $\tau$ being sufficiently large implies $\lambda_j(\tau) \notin \overline{\mathcal{D}}_s$ for all $j$. The conclusion then follows from Proposition 5.1. $\qquad\square$

### G.5 Proof of Theorem 5.3

The proof of this theorem utilizes several theorems on the locations of eigenvalues. For completeness, we hereby include their statements.

**Theorem G.6** (Weyl). *Let $\boldsymbol{A}$, $\boldsymbol{B}$ be Hermitian matrices, and let the respective eigenvalues of $\boldsymbol{A}$, $\boldsymbol{B}$, and $\boldsymbol{A} + \boldsymbol{B}$ be $\{\lambda_k(\boldsymbol{A})\}$, $\{\lambda_k(\boldsymbol{B})\}$, and $\{\lambda_k(\boldsymbol{A} + \boldsymbol{B})\}$, each sorted in increasing order. Then for each $k$, it holds that*

$$\lambda_{k-j+1}(\boldsymbol{A}) + \lambda_j(\boldsymbol{B}) \leq \lambda_k(\boldsymbol{A} + \boldsymbol{B})$$

*for any $j = 1, \ldots, k$.*

**Definition 7.** *For any matrix $\boldsymbol{A}$, let $\boldsymbol{A}^{\mathsf{H}}$ denote the* conjugate transpose *of $\boldsymbol{A}$. When $\boldsymbol{A}$ is a square matrix, the matrix $\frac{1}{2}(\boldsymbol{A} + \boldsymbol{A}^{\mathsf{H}})$ is called the* Hermitian part *of $\boldsymbol{A}$.*

**Theorem G.7** (Bendixson). *Let $\boldsymbol{A}$ be any square matrix, and let $\boldsymbol{R}$ be its Hermitian part. Then for every eigenvalue $\lambda$ of $\boldsymbol{A}$ it holds that*

$$\lambda_{\min}(\boldsymbol{R}) \leq \operatorname{Re}\lambda \leq \lambda_{\max}(\boldsymbol{R})$$

*where $\lambda_{\min}(\boldsymbol{R})$ and $\lambda_{\max}(\boldsymbol{R})$ are the minimum and the maximum eigenvalues of $\boldsymbol{R}$, respectively.*

**Theorem G.8** (Rayleigh). *Let $\boldsymbol{A}$ be a Hermitian matrix, and denote by $\lambda_{\max}(\boldsymbol{A})$ the maximum eigenvalue of $\boldsymbol{A}$. Then it holds that*

$$\lambda_{\max}(\boldsymbol{A}) = \max_{\|\boldsymbol{x}\|=1} \boldsymbol{x}^{\mathsf{H}}\boldsymbol{A}\boldsymbol{x}. \tag{32}$$

Theorems G.6 and G.8 can be found in (Horn & Johnson, 2012) as Theorem 4.3.1 and Theorem 4.2.2, respectively. Theorem G.7 can be found as Theorem 6.9.15 in (Stoer & Bulirsch, 2002). For the proof of these theorems, see the references.

To be more precise, in terms of Rayleigh's theorem, we need the following corollary rather than the theorem itself.

**Corollary G.9.** *Let $\boldsymbol{E} = \mathrm{diag}\{\vartheta_1, \ldots, \vartheta_n\}$ be a real diagonal matrix. Choose $\beta$ to be a nonnegative number such that $-\beta \leq \min\{\vartheta_1, \ldots, \vartheta_n\}$. Then for any real matrix $\boldsymbol{C} \in \mathbb{R}^{m \times n}$, it holds that*

$$\lambda_{\min}(\boldsymbol{C}\boldsymbol{E}\boldsymbol{C}^\top) \geq -\beta \|\boldsymbol{C}\|^2$$

*where $\lambda_{\min}(\boldsymbol{C}\boldsymbol{E}\boldsymbol{C}^\top)$ denotes the minimum eigenvalue of $\boldsymbol{C}\boldsymbol{E}\boldsymbol{C}^\top$.*

*Proof.* Let $\lambda_{\max}(\,\cdot\,)$ denote the largest eigenvalue of a given matrix. By how we chose $\beta$, we have $-\vartheta_k \leq \beta$ for all $k$. Hence, for any fixed $\boldsymbol{x}$, let $\boldsymbol{C}^\top \boldsymbol{x} = \boldsymbol{y} = [y_1 \; \ldots \; y_m]^\top$, then

$$\boldsymbol{x}^\mathsf{H} \boldsymbol{C}(-\boldsymbol{E})\boldsymbol{C}^\top \boldsymbol{x} = \boldsymbol{y}^\top (-\boldsymbol{E})\boldsymbol{y} = \sum_{k=1}^{m} -\vartheta_k y_k^2$$

$$\leq \beta \sum_{k=1}^{m} y_k^2 = \beta \|\boldsymbol{y}\|^2 = \beta\|\boldsymbol{C}^\top \boldsymbol{x}\|^2 \leq \beta\|\boldsymbol{C}\|^2 \|\boldsymbol{x}\|^2.$$

Taking the maximum over $\|\boldsymbol{x}\| = 1$, by Rayleigh's theorem we get

$$\lambda_{\max}(-\boldsymbol{C}\boldsymbol{E}\boldsymbol{C}^\top) = \max_{\|\boldsymbol{x}\|=1} \boldsymbol{x}^\mathsf{H} \boldsymbol{C}(-\boldsymbol{E})\boldsymbol{C}^\top \boldsymbol{x} \leq \beta\|\boldsymbol{C}\|^2.$$

As $\lambda$ is an eigenvalue of $-\boldsymbol{C}\boldsymbol{E}\boldsymbol{C}^\top$ if and only if $-\lambda$ is an eigenvalue of $\boldsymbol{C}\boldsymbol{E}\boldsymbol{C}^\top$, it holds that $\lambda_{\max}(-\boldsymbol{C}\boldsymbol{E}\boldsymbol{C}^\top) = -\lambda_{\min}(\boldsymbol{C}\boldsymbol{E}\boldsymbol{C}^\top)$, so we are done. $\qquad\square$

In order to make the proof clearer, let us state the following simple but technical fact also as a separate lemma.

**Lemma G.10.** *Let $\nu$ be a nonnegative real number. Then, for any $\lambda \in \mathbb{C}^\circ_-$, it holds that*

$$-1 \leq \mathrm{Re}\left(\frac{1}{\frac{\nu}{\lambda} - 1}\right) \leq 0.$$

*Proof.* Let $\lambda = x + iy$ for $x, y \in \mathbb{R}$, and then note that $x \leq 0$. As it holds that

$$\frac{\nu}{\lambda} - 1 = \left(\frac{\nu x}{x^2 + y^2} - 1\right) - i\frac{\nu y}{x^2 + y^2}$$

we have $\mathrm{Re}\left(\frac{\nu}{\lambda} - 1\right) \leq -1$, and consequently $\left|\frac{\nu}{\lambda} - 1\right| \geq 1$. Let $\frac{\nu}{\lambda} - 1 = \xi + i\upsilon$ for $\xi, \upsilon \in \mathbb{R}$, then as we know that $\xi \leq -1$, we obtain

$$\mathrm{Re}\left(\frac{1}{\frac{\nu}{\lambda} - 1}\right) = \frac{\xi}{\xi^2 + \upsilon^2} < 0.$$

Meanwhile, since the size of the real part of a complex number cannot be larger than its absolute value, it also holds that

$$\left|\mathrm{Re}\left(\frac{1}{\frac{\nu}{\lambda} - 1}\right)\right| \leq \left|\frac{1}{\frac{\nu}{\lambda} - 1}\right| \leq 1.$$

Combining the two bounds gives us the result. $\qquad\square$

*Proof of Theorem 5.3(i).* For any $j$ such that $\lambda_j(\epsilon) = \nu_k + o(1)$, since $\boldsymbol{B} \preceq 0$ implies $\nu_k > 0$, for any sufficiently large $\tau$ we have $\lambda_j(\epsilon) \in \mathbb{C}^\circ_+$. In particular, for any $s > 0$, it holds that $\lambda_j(\epsilon) \notin \overline{\mathcal{D}}_s$ whenever $\tau$ is sufficiently large.

For any $j$ such that $\lambda_j(\epsilon) = \epsilon\mu_k + o(\epsilon)$, notice that $\boldsymbol{S} \succeq 0$ implies $\boldsymbol{S}_{\mathrm{res}} \succeq 0$, so we have $\mu_k > 0$. Because $\frac{1}{\epsilon}\lambda_j(\epsilon)$ converges to $\mu_k$, for any sufficiently large $\tau$ we have $\frac{1}{\epsilon}\lambda_j(\epsilon) \in \mathbb{C}^\circ_+$, which implies that $\lambda_j(\epsilon) \in \mathbb{C}^\circ_+$. Therefore, for any $s > 0$, it holds that $\lambda_j(\epsilon) \notin \overline{\mathcal{D}}_s$ whenever $\tau$ is sufficiently large.

Now for convenience, let $\theta_j(\epsilon) := \frac{1}{\epsilon}\lambda_j(\epsilon) = \tau\lambda_j(\epsilon)$. Our next step is to show that

$$\lim_{\epsilon \to 0+} \mathrm{Re}\,(\theta_j(\epsilon)) \geq 0 \qquad \forall j \in \mathcal{I}.$$

If that is the case, for all $j \in \mathcal{I}$ we would have

$$0 \leq \frac{1}{\sigma_j^2}\lim_{\epsilon \to 0+}\mathrm{Re}\,(\theta_j(\epsilon)) = \lim_{\epsilon \to 0+}\frac{1}{\sigma_j^2}\mathrm{Re}\left(\frac{1}{\epsilon}\lambda_j(\epsilon)\right) = \lim_{\epsilon \to 0+}\frac{\xi_j\epsilon^\varrho + o(\epsilon^\varrho)}{\epsilon\sigma_j^2} = \iota_j,$$

and consequently $s_0 \leq 0$. Then, for any $s > 0$, we trivially have $s > s_0$. Hence, by applying Lemma G.4 and Proposition 5.1, it is possible to conclude that $z^*$ is a strict linearly stable point.

Notice that $\theta_j$ are the eigenvalues of the matrix

$$\tau H_\tau^* = \begin{bmatrix} A & C \\ -\tau C^\top & -\tau B \end{bmatrix}.$$

As in Section 4.2, by applying a similarity transformation, we may assume without loss of generality that $B$ is a diagonal matrix. Following the notation therein, let $-B = \mathrm{diag}\{\nu_1, \ldots, \nu_r, 0, \ldots, 0\}$.

Fix any $j \in \mathcal{I}$. Recall that we have a convergent Puiseux series expansion $\lambda_j(\epsilon) = \sum_{k=1}^\infty \alpha_j^{(k)}\epsilon^{k/p_j}$, and notice that it suffices to assume that $\epsilon \in \mathbb{R}$, as we are studying the limit as $\epsilon \to 0+$. Then we see that, as $\epsilon \to 0+$, either $\mathrm{Re}\left(\frac{1}{\epsilon}\lambda_j(\epsilon)\right) = \mathrm{Re}\,(\theta_j(\epsilon))$ converges to a finite number, diverges to $+\infty$, or diverges to $-\infty$. For the sake of contradiction, suppose that $\lim_{\tau \to \infty}\mathrm{Re}\,(\theta_j(\epsilon)) < 0$. By the Schur determinantal formula, the eigenvalue $\theta_j$ of $\tau H_\tau^*$ must be a root of the equation

$$\begin{aligned}
\det(\tau H_\tau^* - \theta I) &= \det\left(\begin{bmatrix} A - \theta I & C \\ -\tau C^\top & -\tau B - \theta I \end{bmatrix}\right) \\
&= \det(-\tau B - \theta I)\det\left(A - \theta I - \tau C(\tau B + \theta I)^{-1}C^\top\right) \\
&= \det(-\tau B - \theta I)\det\left(A - \theta I - C\left(B + \frac{\theta}{\tau}I\right)^{-1}C^\top\right).
\end{aligned}$$

Since $B \preceq 0$, and as for sufficiently large $\tau$ we would have $\mathrm{Re}(\theta_j(\epsilon)) < 0$, it must be the case where $\det(-\tau B - \theta_j I) \neq 0$. With this in mind, observe that

$$\begin{aligned}
&A - \theta I - C\left(B + \frac{\theta}{\tau}I\right)^{-1}C^\top \\
&= A - \theta I - C\left(\mathrm{diag}\left\{-\nu_1 + \frac{\theta}{\tau}, \ldots, -\nu_r + \frac{\theta}{\tau}, \frac{\theta}{\tau}, \ldots, \frac{\theta}{\tau}\right\}\right)^{-1}C^\top \\
&= A - \theta I + C\,\mathrm{diag}\left\{\frac{\tau}{\tau\nu_1 - \theta}, \ldots, \frac{\tau}{\tau\nu_r - \theta}, -\frac{\tau}{\theta}, \ldots, -\frac{\tau}{\theta}\right\}C^\top \\
&= A - CB^\dagger C^\top - \theta I + C\,\mathrm{diag}\left\{\frac{1}{\nu_1} \cdot \frac{1}{\frac{\tau\nu_1}{\theta} - 1}, \ldots, \frac{1}{\nu_r} \cdot \frac{1}{\frac{\tau\nu_r}{\theta} - 1}, -\frac{\tau}{\theta}, \ldots, -\frac{\tau}{\theta}\right\}C^\top.
\end{aligned}$$

To see why the last line holds, observe that $\frac{\tau}{\tau\nu - \theta} = \frac{1}{\nu} + \frac{1}{\nu(\frac{\tau\nu}{\theta} - 1)}$. Let us denote

$$E := \mathrm{diag}\left\{\frac{1}{\nu_1} \cdot \frac{1}{\frac{\tau\nu_1}{\theta} - 1}, \ldots, \frac{1}{\nu_r} \cdot \frac{1}{\frac{\tau\nu_r}{\theta} - 1}, -\frac{\tau}{\theta}, \ldots, -\frac{\tau}{\theta}\right\}$$

so that $\theta_j(\epsilon)$ becomes a root of the equation

$$\det(A - CB^\dagger C^\top - \theta I + CEC^\top) = 0. \tag{33}$$

Now suppose that $\lim_{\tau \to \infty}\mathrm{Re}\,(\theta_j(\epsilon)) = -\infty$. For sufficiently large $\tau$, having $\mathrm{Re}(\theta_j(\epsilon)) < 0$ implies $\mathrm{Re}(\theta_j(\epsilon)/\tau) < 0$. So by Lemma G.10,

$$-1 \leq \mathrm{Re}\left(\frac{1}{\frac{\tau\nu_k}{\theta_j} - 1}\right) \leq 0 \tag{34}$$

for all $k = 1, \ldots, r$. Here, note that the Hermitian part of $\boldsymbol{C}\boldsymbol{E}\boldsymbol{C}^\top$ is

$$\frac{1}{2}\left(\boldsymbol{C}\boldsymbol{E}\boldsymbol{C}^\top + (\boldsymbol{C}\boldsymbol{E}\boldsymbol{C}^\top)^{\mathsf{H}}\right) = \boldsymbol{C}\left(\frac{\boldsymbol{E} + \boldsymbol{E}^{\mathsf{H}}}{2}\right)\boldsymbol{C}^\top$$
$$= \boldsymbol{C}\operatorname{Re}(\boldsymbol{E})\boldsymbol{C}^\top$$

where $\operatorname{Re}(\boldsymbol{E})$ denotes the *elementwise* real part of $\boldsymbol{E}$, by the fact that $\boldsymbol{E}$ is diagonal. Hence, using Corollary G.9 with (34), we get that the Hermitian part of $\boldsymbol{C}\boldsymbol{E}\boldsymbol{C}^\top$ has a spectrum that is bounded below by a constant, uniformly on $\tau$. As we assume that $\boldsymbol{A} - \boldsymbol{C}\boldsymbol{B}^\dagger\boldsymbol{C}^\top \succeq \boldsymbol{0}$, by Weyl's theorem, the spectrum of the Hermitian part of $\boldsymbol{A} - \boldsymbol{C}\boldsymbol{B}^\dagger\boldsymbol{C}^\top + \boldsymbol{C}\boldsymbol{E}\boldsymbol{C}^\top$ is also bounded below by a constant, uniformly on $\tau$. Finally, noticing that the Hermitian part of $\theta\boldsymbol{I}$ is $\operatorname{Re}(\theta)\boldsymbol{I}$, and that we assume $\lim_{\tau\to\infty}\operatorname{Re}(\theta_j(\epsilon)) = -\infty$, Bendixson's theorem tells us that $\tau$ being sufficiently large implies every eigenvalue of $\boldsymbol{A} - \boldsymbol{C}\boldsymbol{B}^\dagger\boldsymbol{C}^\top - \theta_j\boldsymbol{I} + \boldsymbol{C}\boldsymbol{E}\boldsymbol{C}^\top$ having a strictly positive real part. However, this means that $\boldsymbol{A} - \boldsymbol{C}\boldsymbol{B}^\dagger\boldsymbol{C}^\top - \theta_j\boldsymbol{I} + \boldsymbol{C}\boldsymbol{E}\boldsymbol{C}^\top$ is invertible, hence (33) cannot hold.

We are left with the case where $\operatorname{Re}(\theta_j(\epsilon))$ converges to a negative finite value, say $-\psi_j$, as $\tau \to \infty$. In that case, notice that

$$\lim_{\tau\to\infty}\frac{1}{\frac{\tau\nu_k}{\theta_j} - 1} = 0$$

for all $k = 1, \ldots, r$. As we assume that $\boldsymbol{A} - \boldsymbol{C}\boldsymbol{B}^\dagger\boldsymbol{C}^\top \succeq \boldsymbol{0}$, following the same logic used in the previous paragraph, this time we can find $\tau_0$ such that for any $\tau > \tau_0$ the spectrum of the Hermitian part of $\boldsymbol{A} - \boldsymbol{C}\boldsymbol{B}^\dagger\boldsymbol{C}^\top + \boldsymbol{C}\boldsymbol{E}\boldsymbol{C}^\top$ becomes bounded below by $-\psi_j/2$. Meanwhile, there also exists $\tau_1$ such that $\tau > \tau_1$ implies $-\operatorname{Re}(\theta_j(\epsilon)) > \frac{2}{3}\psi_j$. Therefore, by Bendixson's theorem, whenever $\tau > \max\{\tau_0, \tau_1\}$ the eigenvalues of $\boldsymbol{A} - \boldsymbol{C}\boldsymbol{B}^\dagger\boldsymbol{C}^\top - \theta_j\boldsymbol{I} + \boldsymbol{C}\boldsymbol{E}\boldsymbol{C}^\top$ will have a positive real part that is greater than $\psi_j/6$. But then, this again implies that (33) does not hold, which is absurd.

Therefore, for any $j$ it holds that $\lim_{\tau\to\infty}\operatorname{Re}(\theta_j(\epsilon)) \geq 0$. As claimed in the above, this completes the proof. □

*Proof of Theorem 5.3(ii).* Suppose that $\boldsymbol{S}_{\mathrm{res}} \not\succeq \boldsymbol{0}$, or equivalently, there exists some $k$ such that $\mu_k < 0$. Then, by Theorem 4.3, we have $\lambda_j(\epsilon) = \epsilon\mu_k + o(\epsilon)$ for some $j$. In other words, for some $j$ it holds that $\lambda_j(\epsilon)/\epsilon \to \mu_k$ as $\epsilon \to 0+$. So by choosing $s = \frac{1}{2|\mu_k|} > 0$ we must have $\lambda_j(\epsilon)/\epsilon \in \overline{\mathcal{D}}_s$ whenever $\tau$ is sufficiently large. Since $\epsilon\overline{\mathcal{D}}_s \subset \overline{\mathcal{D}}_s$ whenever $0 < \epsilon \leq 1$, it follows that $\lambda_j(\epsilon) \in \overline{\mathcal{D}}_s$ whenever $\tau$ is sufficiently large. However by Proposition 5.1 this implies that $\boldsymbol{z}^*$ is not strict linearly stable, which is absurd. Therefore, it is necessary that $\boldsymbol{S}_{\mathrm{res}} \succeq \boldsymbol{0}$.

Next, suppose that $\boldsymbol{B} \not\preceq \boldsymbol{0}$, or equivalently, there exists some $k$ such that $\nu_k < 0$. Then, by Theorem 4.3, as $\lambda_j(\epsilon)$ converges to $\nu_k$ for some $j$, if we choose $s = \frac{1}{2|\nu_k|} > 0$ we must have $\lambda_j(\epsilon) \in \overline{\mathcal{D}}_s$ whenever $\tau$ is sufficiently large. However again by Proposition 5.1 this implies that $\boldsymbol{z}^*$ is not strict linearly stable, which is absurd. Therefore, it is necessary that $\boldsymbol{B} \preceq \boldsymbol{0}$. □

### G.6  PROOF OF THEOREM 5.4

Let us first identify what happens if $\sigma_j$ are all distinct. In doing so, the following proposition, which summarizes the discussion made in (Kato, 1995, Section II.1.2), will be useful. For further details and the proof of the statements, see the original reference.

**Proposition G.11.** *Consider a convergent perturbation series $\boldsymbol{T}(\varkappa) = \boldsymbol{T}_0 + \varkappa\boldsymbol{T}_1 + \varkappa^2\boldsymbol{T}_2 + \cdots$, and suppose that $0$ is in the spectrum of the unperturbed operator $\boldsymbol{T}_0$. Let $\lambda_1(\varkappa), \ldots, \lambda_s(\varkappa)$ be the eigenvalues of $\boldsymbol{T}(\varkappa)$ with $\lambda_j(0) = 0$ for all $j = 1, \ldots, s$. We may assume that such functions are holomorphic on a deleted neighborhood of $0$.*

*Furthermore, we may assume that those functions can be grouped into $\{\lambda_1(\varkappa), \ldots, \lambda_{g_1}(\varkappa)\}, \{\lambda_{g_1+1}(\varkappa), \ldots, \lambda_{g_1+g_2}(\varkappa)\}, \ldots$ so that within each group, by letting $g$ to be the size of that group and relabelling the indices of the functions in that group as $\lambda_{j_1}, \ldots, \lambda_{j_g}$, the Puiseux series expansion*

$$\lambda_{j_h}(\epsilon) = \sum_{\rho=0}^{\infty} a_\rho\left(\omega^h\epsilon^{1/g}\right)^\rho$$

*holds for all $h = 1, \ldots, g$. Here, $\omega$ is a primitive $g^{th}$ root of unity.*

The fact that the eigenvalues can be grouped in a way that is described above is crucial, leading to the following observation.

**Lemma G.12.** *Suppose that all $\sigma_j$ are distinct. Then, the Puiseux series $\lambda_j(\epsilon) = \sum_{k=1}^{\infty} \alpha_j^{(k)} \epsilon^{k/p_j}$ essentially becomes a power series of $\epsilon^{1/2}$. In particular, $\varrho_j \geq 1$.*

*Proof.* Fix any $\lambda_j$ with $j \in \mathcal{I}$. By Proposition G.11, there exists some positive integer $g$ such that we can find $\lambda_{j_1} = \lambda_j, \ldots, \lambda_{j_g}$ whose Puiseux series expansions are

$$\lambda_{j_s}(\epsilon) = \sum_{\rho=0}^{\infty} a_\rho \left( \omega^s \epsilon^{1/g} \right)^\rho$$

for $\omega$ a primitive $g^{\text{th}}$ root of unity. In particular, the absolute values of the coefficients corresponding to the terms of the same order in each of these $g$ series must be all equal. Since all $\sigma_j$ values are distinct by assumption, and since they are all real and positive, we conclude that $g = 2$. It is also now clear that the second lowest order term in the Puiseux series expansion is of order $O(\epsilon)$. □

Now, observe that if $\varrho_j > 1$, then $\iota_j = 0$. Hence, in the case where $\varrho_j > 1$, there is no loss of generality when considering it as having $\varrho_j = 1$ and $\zeta_j = 0$. In other words, by allowing $\zeta_j = 0$, we may assume that the Puiseux expansion of $\lambda_j$ is of the form

$$\lambda_j = \pm i\sigma_j \sqrt{\epsilon} + \zeta_j \epsilon + o(\epsilon).$$

This suggests a reparametrization, say for example letting $\sqrt{\epsilon} = \delta$, so that the Puiseux series can be dealt as if it is an ordinary power series.

In case where an eigenvalue of a perturbed linear operator (on a finite dimensional space) admits a power series expansion, much more is known on quantitative results regarding the coefficients of that power series. For instance, the following proposition summarizes the discussions made in Sections I.5.3, II.2.2, and II.3.1 in (Kato, 1995). For the details and the proof of the statements, see the original reference.

**Proposition G.13.** *Consider a convergent perturbation series $\boldsymbol{T}(\varkappa) = \boldsymbol{T}_0 + \varkappa \boldsymbol{T}_1 + \varkappa^2 \boldsymbol{T}_2 + \cdots$. Suppose that $\lambda_0$ is a simple eigenvalue of $\boldsymbol{T}_0$, and define*

$$\boldsymbol{P}_{\lambda_0} \coloneqq \frac{1}{2\pi i} \int_\Gamma (\zeta \boldsymbol{I} - \boldsymbol{T}_0)^{-1} d\zeta \tag{35}$$

*where $\Gamma$ is a positively oriented simple closed contour around $\lambda_0$ which does not contain any other eigenvalues of $\boldsymbol{T}_0$. Then, there exists an eigenvalue $\lambda(\varkappa)$ of a perturbed operator $\boldsymbol{T}(\varkappa)$ such that $\lambda(0) = \lambda_0$, which admits a power series expansion*

$$\lambda(\varkappa) = \lambda_0 + \varkappa \lambda^{(1)} + \varkappa^2 \lambda^{(2)} + \cdots. \tag{36}$$

*Here, it holds that*

$$\lambda^{(1)} = \mathrm{tr}(\boldsymbol{T}_1 \boldsymbol{P}_{\lambda_0}).$$

*Moreover, let $R$ be the radius of convergence of the power series (36). Then, the fact that the perturbation series is convergent guarantees the existence of nonnegative constants $a$ and $c$ satisfying $\|\boldsymbol{T}_n\| \leq ac^{n-1}$ for all $n = 1, 2, \cdots$. For such $a$ and $c$ it holds that*

$$R \geq r_0 \coloneqq \min_{\zeta \in \Gamma} \frac{1}{a \|\zeta \boldsymbol{I} - \boldsymbol{T}_0\| + c}, \tag{37}$$

*and there exists a positive constant $\rho$ such that*

$$\left| \lambda(\varkappa) - \lambda_0 - \varkappa \lambda^{(1)} \right| \leq \frac{\rho |\varkappa|^2}{r_0(r_0 - |\varkappa|)}. \tag{38}$$

The operator $\boldsymbol{P}_{\lambda_0}$ defined as in (35) is called the *eigenprojection* of $\boldsymbol{T}_0$ for $\lambda_0$. If $\boldsymbol{T}_0$ is additionally assumed to be normal, we have the following characterization of the eigenprojection for $\lambda_0$.

**Lemma G.14.** *Suppose that $\lambda_0$ is a simple eigenvalue of a normal operator $\boldsymbol{T}_0$. Let $\boldsymbol{w}$ be the unit norm eigenvector of $\boldsymbol{T}_0$ associated with $\lambda_0$. Then, $\boldsymbol{P}_{\lambda_0} = \boldsymbol{w}\boldsymbol{w}^\top$.*

*Proof.* Because $\boldsymbol{T}_0$ is normal, it admits a diagonalization $\boldsymbol{T}_0 = \boldsymbol{W}\boldsymbol{\Theta}\boldsymbol{W}^\top$, where we may assume without loss of generality that $\boldsymbol{\Theta} = \mathrm{diag}\{\theta_1, \theta_2, \dots\}$ with $\theta_1 = \lambda_0$ and $\boldsymbol{W}$ is a unitary matrix with its first column being $\boldsymbol{w}$. Then, by definition (35), it holds that

$$
\begin{aligned}
\boldsymbol{P}_{\lambda_0} &= \frac{1}{2\pi i}\int_\Gamma (\zeta\boldsymbol{I} - \boldsymbol{T}_0)^{-1}d\zeta \\
&= \boldsymbol{W}\left(\frac{1}{2\pi i}\int_\Gamma (\zeta\boldsymbol{I} - \boldsymbol{\Theta})^{-1}d\zeta\right)\boldsymbol{W}^\top \\
&= \boldsymbol{W}\left(\mathrm{diag}\left\{\frac{1}{2\pi i}\int_\Gamma \frac{d\zeta}{\zeta - \theta_1}, \frac{1}{2\pi i}\int_\Gamma \frac{d\zeta}{\zeta - \theta_2}, \dots\right\}\right)\boldsymbol{W}^\top \\
&= \boldsymbol{W}\,\mathrm{diag}\{1, 0, \dots, 0\}\boldsymbol{W}^\top \\
&= \boldsymbol{w}\boldsymbol{w}^\top
\end{aligned}
$$

where in the fourth line we use the fact that $\Gamma$ encloses $\lambda_0 = \theta_1$ but no other eigenvalues of $\boldsymbol{T}_0$. $\quad\square$

*Proof of Theorem 5.4.* Recall that, in the proof of Theorem 4.3, we have shown that any eigenvalue such that $\lambda_j \to 0$ as $\epsilon \to 0$ should be a solution of the equation (24), namely the equation

$$
0 = \det\begin{bmatrix} \kappa\boldsymbol{I} - \sqrt{\epsilon}\big(\boldsymbol{A} - \boldsymbol{C}_1(\lambda\boldsymbol{I} - \boldsymbol{D})^{-1}\boldsymbol{C}_1^\top\big) & -\boldsymbol{C}_2 \\ \boldsymbol{C}_2^\top & \kappa\boldsymbol{I} \end{bmatrix}.
$$

Reparametrizing the above by $\delta = \sqrt{\epsilon}$ gives us

$$
0 = \det\begin{bmatrix} \kappa\boldsymbol{I} - \delta\big(\boldsymbol{A} - \boldsymbol{C}_1(\kappa\delta\boldsymbol{I} - \boldsymbol{D})^{-1}\boldsymbol{C}_1^\top\big) & -\boldsymbol{C}_2 \\ \boldsymbol{C}_2^\top & \kappa\boldsymbol{I} \end{bmatrix}.
$$

Now fix any index $j \in \{1, 2, \dots, d_2 - r\}$. The cases where $j \in \{d_1 + 1, \dots, d_1 + d_2 - r\}$ can be dealt in the exact same way that is presented hereinafter, by simply changing every instance of $\sigma_j$ to $-\sigma_j$. We a priori know that $\kappa_j(\delta) := \lambda_j(\delta)/\delta = i\sigma_j + \zeta_j\delta + o(\delta)$. Fix $\delta_0 > 0$ arbitrarily, but small enough so that $\lambda_j(\delta_0)\|\boldsymbol{D}\| < 1$. Setting $\hat{\kappa} = \lambda_j(\delta_0)/\delta_0$, we have that $\kappa_j$ is a solution of the equation

$$
0 = \det\begin{bmatrix} \kappa\boldsymbol{I} - \delta_0\big(\boldsymbol{A} - \boldsymbol{C}_1(\hat{\kappa}\delta_0\boldsymbol{I} - \boldsymbol{D})^{-1}\boldsymbol{C}_1^\top\big) & -\boldsymbol{C}_2 \\ \boldsymbol{C}_2^\top & \kappa\boldsymbol{I} \end{bmatrix}.
$$

In other words, $\kappa_j(\delta_0)$ is an eigenvalue of

$$
\begin{aligned}
&\begin{bmatrix} \boldsymbol{0} & \boldsymbol{C}_2 \\ -\boldsymbol{C}_2^\top & \boldsymbol{0} \end{bmatrix} + \delta_0\begin{bmatrix} \boldsymbol{A} - \boldsymbol{C}_1(\hat{\kappa}\delta_0\boldsymbol{I} - \boldsymbol{D})^{-1}\boldsymbol{C}_1^\top & \boldsymbol{0} \\ \boldsymbol{0} & \boldsymbol{0} \end{bmatrix} \\
&= \begin{bmatrix} \boldsymbol{0} & \boldsymbol{C}_2 \\ -\boldsymbol{C}_2^\top & \boldsymbol{0} \end{bmatrix} + \delta_0\begin{bmatrix} \boldsymbol{A} + \boldsymbol{C}_1\boldsymbol{D}^{-1}\boldsymbol{C}_1^\top & \boldsymbol{0} \\ \boldsymbol{0} & \boldsymbol{0} \end{bmatrix} + \sum_{k=2}^\infty \delta_0^k\begin{bmatrix} \hat{\kappa}^{k-1}\boldsymbol{C}_1\boldsymbol{D}^{k-2}\boldsymbol{C}_1^\top & \boldsymbol{0} \\ \boldsymbol{0} & \boldsymbol{0} \end{bmatrix}.
\end{aligned}
$$

Notice that the infinite series expansion above is valid, as $\delta_0$ is chosen to satisfy the condition $\hat{\kappa}\delta_0\|\boldsymbol{D}\| = \lambda_j(\delta_0)\|\boldsymbol{D}\| < 1$. Now, because $\hat{\kappa} \to i\sigma_j$ as $\delta_0 \to 0$, by restricting the range of $\delta_0$ further if necessary, we may assume that $\hat{\kappa}$ is uniformly bounded on $\delta_0$. In particular, we can set the range of $\delta_0$ so that $|\hat{\kappa}| \le \sigma_j + 1$ uniformly on $\delta_0$. Then, the coefficients of the perturbation series are bounded by

$$
\left\| \begin{bmatrix} \hat{\kappa}^{k-1}\boldsymbol{C}_1\boldsymbol{D}^{k-2}\boldsymbol{C}_1^\top & \boldsymbol{0} \\ \boldsymbol{0} & \boldsymbol{0} \end{bmatrix} \right\| \le \|\boldsymbol{C}_1\|^2 (\sigma_j + 1)^{k-1}\|\boldsymbol{D}\|^{k-2}.
$$

Notice that these bounds are independent of $\delta_0$. Invoking Proposition G.13, with restricting the range of $\delta_0$ further if necessary in order to guarantee the convergence of the power series (36), we obtain

$$
\kappa_j(\delta_0) = i\sigma_j + \delta_0\,\mathrm{tr}\left(\begin{bmatrix} \boldsymbol{A} + \boldsymbol{C}_1\boldsymbol{D}^{-1}\boldsymbol{C}_1^\top & \boldsymbol{0} \\ \boldsymbol{0} & \boldsymbol{0} \end{bmatrix}\boldsymbol{P}\right) + \delta_0^2\kappa^{(2)} + \cdots.
$$

Moreover, by further restricting the range of $\delta_0$ if necessary so that $\delta_0 \leq r_0/2$, where $r_0$ is the quantity introduced in (37), we can apply the bound (38) to get a constant $C$ that is independent of $\delta_0$ and satisfies

$$\left| \kappa_j(\delta_0) - i\sigma_j - \delta_0 \operatorname{tr}\left( \begin{bmatrix} \boldsymbol{A} + \boldsymbol{C}_1 \boldsymbol{D}^{-1} \boldsymbol{C}_1^\top & \boldsymbol{0} \\ \boldsymbol{0} & \boldsymbol{0} \end{bmatrix} \boldsymbol{P} \right) \right| \leq C\delta_0^2.$$

Note that this estimate holds for all sufficiently small $\delta_0$. Therefore, we can conclude that

$$\zeta_j = \operatorname{tr}\left( \begin{bmatrix} \boldsymbol{A} + \boldsymbol{C}_1 \boldsymbol{D}^{-1} \boldsymbol{C}_1^\top & \boldsymbol{0} \\ \boldsymbol{0} & \boldsymbol{0} \end{bmatrix} \boldsymbol{P} \right).$$

Observe that, for $\boldsymbol{u}_j$ and $\boldsymbol{v}_j$ the left and right singular vectors of $\boldsymbol{C}_2$ associated with the singular value $\sigma_j$ respectively, the vector $\begin{bmatrix} \boldsymbol{u}_j \\ i\boldsymbol{v}_j \end{bmatrix}$ is the eigenvector of the unperturbed operator $\begin{bmatrix} \boldsymbol{0} & \boldsymbol{C}_2 \\ -\boldsymbol{C}_2^\top & \boldsymbol{0} \end{bmatrix}$ that is associated to the eigenvalue $i\sigma_j$. Normalizing the eigenvector gives us

$$\hat{\boldsymbol{w}} = \frac{1}{\sqrt{\|\boldsymbol{u}_j\|^2 + \|\boldsymbol{v}_j\|^2}} \begin{bmatrix} \boldsymbol{u}_j \\ i\boldsymbol{v}_j \end{bmatrix} = \frac{1}{\sqrt{2}} \begin{bmatrix} \boldsymbol{u}_j \\ i\boldsymbol{v}_j \end{bmatrix},$$

and Lemma G.14, we moreover have $\boldsymbol{P} = \hat{\boldsymbol{w}}\hat{\boldsymbol{w}}^\top$. Then, by the cyclic invariance of the trace, we get

$$\zeta_j = \operatorname{tr}\left( \begin{bmatrix} \boldsymbol{A} + \boldsymbol{C}_1 \boldsymbol{D}^{-1} \boldsymbol{C}_1^\top & \boldsymbol{0} \\ \boldsymbol{0} & \boldsymbol{0} \end{bmatrix} \boldsymbol{P} \right)$$

$$= \hat{\boldsymbol{w}}_j^\top \begin{bmatrix} \boldsymbol{A} + \boldsymbol{C}_1 \boldsymbol{D}^{-1} \boldsymbol{C}_1^\top & \boldsymbol{0} \\ \boldsymbol{0} & \boldsymbol{0} \end{bmatrix} \hat{\boldsymbol{w}}_j$$

$$= \frac{1}{2} \boldsymbol{u}_j^\top (\boldsymbol{A} + \boldsymbol{C}_1 \boldsymbol{D}^{-1} \boldsymbol{C}_1^\top) \boldsymbol{u}_j.$$

By the structure of $\boldsymbol{B}$, we have $\boldsymbol{C}_1 \boldsymbol{D}^{-1} \boldsymbol{C}_1^\top = -\boldsymbol{C}\boldsymbol{B}^\dagger \boldsymbol{C}^\top$, so we are done. $\qquad\square$

## H  PROOFS AND MISSING DETAILS FOR SECTION 5.3

### H.1  ON THE PEANUT-SHAPED REGION $\mathcal{P}_\eta$

The peanut-shaped region $\mathcal{P}_\eta$ defined in (8) is illustrated in Figure 3. This $\mathcal{P}_\eta$ is a dilation of $\mathcal{P}_1$ by a factor of $\frac{1}{\eta}$, and contains a subset of the imaginary axis $\left\{ it : t \in \left(-\frac{1}{\eta}, 0\right) \cup \left(0, \frac{1}{\eta}\right) \right\} \subset i\mathbb{R}$.

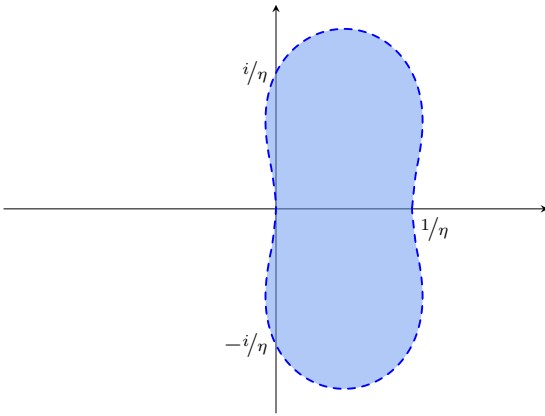

Figure 3: The peanut-shaped region $\mathcal{P}_\eta$ in the complex plane

Similar to the analysis in Appendix E.2, the tangential information alone may not be sufficient for discrete-time case stability analysis: see Figure 4. Consider two peanut-shaped regions $\mathcal{P}_{\eta/16}$ and $\mathcal{P}_\eta$ on the complex plane. Both are tangent to the imaginary axis at 0, but have different dilation parameter; $\frac{1}{\eta}$ and $\frac{16}{\eta}$ respectively. Suppose that some $\lambda_j(\epsilon)$ satisfies (27) with $\varrho_j = 1$ and $\frac{\xi_j}{\sigma_j^2} = -\frac{\eta}{9}$. Then we can visually see that $\lambda_j(\epsilon) \in \mathcal{P}_\eta$ and $\lambda_j(\epsilon) \notin \mathcal{P}_{\eta/16}$ becomes true as $\epsilon \to 0+$.

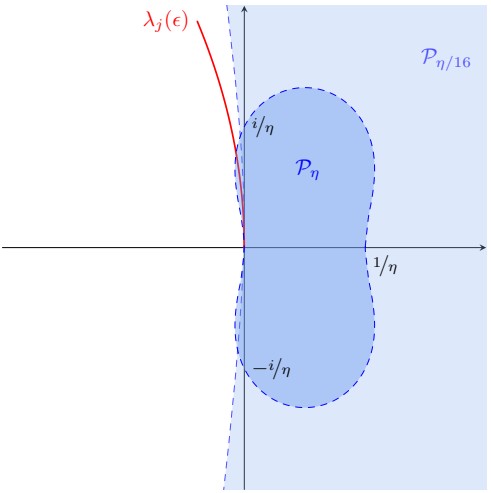
(a) An overall view of the depiction.

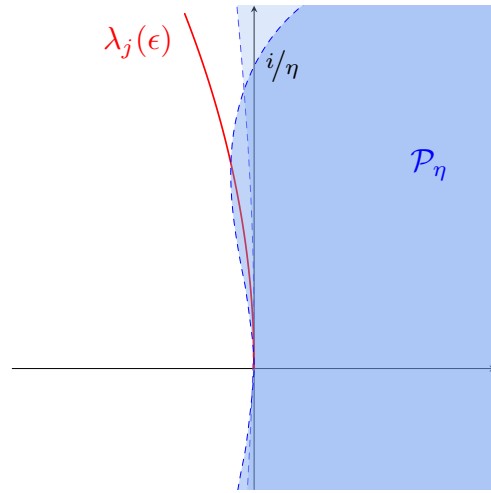
(b) Enlarged version of the figure on the left.

Figure 4: Two peanut-shaped region and a curve, all tangent to the same line at the same point.

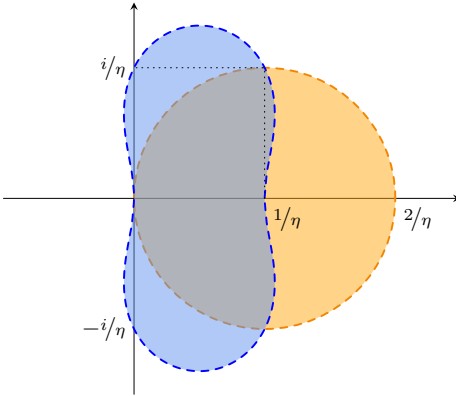

Figure 5: Target sets of discrete-time $\tau$-EG and $\tau$-GDA, both using $\eta$ as their step sizes. The (blue) peanut-shaped region is the target set of $\tau$-EG, and the (orange) disk is the target set of $\tau$-GDA.

## H.2  TARGET SETS OF DISCRETE-TIME METHODS

Similar to the continuous-time case, let us compare this to the target set of $\tau$-GDA. As studied by Fiez & Ratliff (2021), the Jacobian of (discrete) $\tau$-GDA is $\boldsymbol{I} - \eta\boldsymbol{H}_\tau$, so the target set of $\tau$-GDA becomes the open disk with radius $1/\eta$ centered at $1/\eta$. In Figure 5 we have overlaid the depictions of the target set of $\tau$-EG and the target set of $\tau$-EG, when both methods use $\eta$ as their step size.

Notice that it is now visually apparent why $\boldsymbol{0}$ in Example 2 is not a strict linearly stable point for $\tau$-GDA. Clearly, the target set of $\tau$-GDA—the disk—will never contain $\pm i\sqrt{\epsilon}$ for any $\epsilon > 0$.

## H.3  PROOF OF PROPOSITION 5.5

For convenience, let us define a subset of the complex plane
$$\mathcal{D}'_{\eta/2} := \left\{ z \in \mathbb{C} : \left| z - \frac{1}{\eta} \right| < \frac{1}{\eta} \right\}$$
which is an open disk centered at $\frac{1}{\eta}$ with radius $\frac{1}{\eta}$.

At an equilibrium point $\boldsymbol{z}^*$, it holds that
$$\boldsymbol{J}_\tau(\boldsymbol{z}^*) = \boldsymbol{I} - \eta\boldsymbol{H}_\tau^*(\boldsymbol{I} - \eta\boldsymbol{H}_\tau^*),$$

and one can easily show that $\rho(\boldsymbol{J}_\tau^*) < 1$ if and only if $\mathrm{spec}(\boldsymbol{H}_\tau^*(\boldsymbol{I} - \eta\boldsymbol{H}_\tau^*)) \in \mathcal{D}_{\eta/2}'$. The only remaining part to be proven is that $\mathrm{spec}(\boldsymbol{H}_\tau^*(\boldsymbol{I} - \eta\boldsymbol{H}_\tau^*)) \subset \mathcal{D}_{\eta/2}'$ is equivalent to $\mathrm{spec}(\boldsymbol{H}_\tau^*) \subset \mathcal{P}_\eta$.

Let $\lambda$ be an eigenvalue of $\boldsymbol{H}_\tau^*$. Then $\mathrm{spec}(\boldsymbol{H}_\tau^*(\boldsymbol{I} - \eta\boldsymbol{H}_\tau^*)) \subset \mathcal{D}_{\eta/2}'$ implies $\lambda - \eta\lambda^2 \in \mathcal{D}_{\eta/2}'$. By letting $\lambda = x + iy$, where $x$ denotes the real part and $y$ denotes the imaginary part of $\lambda$, we have $\lambda - \eta\lambda^2 = x - \eta x^2 + \eta y^2 + i(y - 2\eta xy)$. Then, the condition $\lambda - \eta\lambda^2 \in \mathcal{D}_{\eta/2}'$ can be equivalently written as

$$(\eta x - \eta^2 x^2 + \eta^2 y^2 - 1)^2 + (\eta y - 2\eta^2 xy)^2 < 1.$$

This can be further simplified as

$$\left(\eta x - \frac{1}{2}\right)^2 + \eta^2 y^2 + \frac{3}{4} < \sqrt{1 + 3\eta^2 y^2},$$

and the assertion then follows from the definition of $\mathcal{P}_\eta$.

## H.4 PROOF OF THEOREM 5.6

In proving Theorem 5.6, the following two results will be used.

**Proposition H.1.** *For $\lambda_j(\epsilon)$ as in (27), it holds that*

$$\lim_{\epsilon \to 0+} \mathrm{Re}\left(\frac{1}{\lambda_j(1 - \eta\lambda_j)}\right) = \iota_j + \eta. \tag{39}$$

*Proof.* Using (29) and substituting (27) into the above leads to

$$
\begin{aligned}
\lim_{\epsilon \to 0+} \mathrm{Re}\left(\frac{1}{\lambda_j(1 - \eta\lambda_j)}\right) &= \lim_{\epsilon \to 0+} \frac{\mathrm{Re}(\lambda_j(1 - \eta\lambda_j))}{\mathrm{Re}(\lambda_j(1 - \eta\lambda_j))^2 + \mathrm{Im}(\lambda_j(1 - \eta\lambda_j))^2} \\
&= \lim_{\epsilon \to 0+} \frac{\xi_j \epsilon^{\varrho_j - 1} + o(\epsilon^{\varrho_j - 1}) + \eta\sigma_j^2 + o(1)}{\sigma_j^2 + o(1)} \\
&= \iota_j + \eta
\end{aligned}
$$

where the second line uses the fact that $\varrho > 1/2$ implies $\epsilon^{2\varrho} = o(\epsilon)$. $\qquad\square$

**Lemma H.2.** *Let $s_0 := \max_{j \in \mathcal{I}}(-\iota_j)$. If $\frac{\eta}{2} > s_0$, then there exists some $\tau^* > 0$ such that whenever $\tau > \tau^*$, all the eigenvalues $\lambda_j$ with $j \in \mathcal{I}$ lie in $\mathcal{P}_\eta$. Conversely, if $\frac{\eta}{2} < s_0$, then for any sufficiently large $\tau$, there exists some $j \in \mathcal{I}$ such that $\lambda_j(\epsilon) \notin \mathcal{P}_\eta$.*

*Proof of Lemma H.2.* Recall that, in the proof of Proposition 5.5, we have shown that the peanut-shaped region can be represented as follows

$$\mathcal{P}_\eta = \left\{z \in \mathbb{C} : \mathrm{Re}\left(\frac{1}{z(1 - \eta z)}\right) > \frac{\eta}{2}\right\}.$$

Therefore, if $-s_0 = \min_{j \in \mathcal{I}} \iota_j > -\frac{\eta}{2}$ then for any sufficiently large $\tau$ and for all $j$ we will have $\mathrm{Re}\left(\frac{1}{\lambda_j(1 - \eta\lambda_j)}\right) > \frac{\eta}{2}$. Hence, all eigenvalues will lie in $\mathcal{P}_\eta$ for a sufficiently large $\tau$. On the other hand, if $-s_0 = \min_{j \in \mathcal{I}} \iota_j < -\frac{\eta}{2}$ then for some $j$ we will have $\mathrm{Re}\left(\frac{1}{\lambda_j(1 - \eta\lambda_j)}\right) < \frac{\eta}{2}$ whenever $\tau$ is sufficiently large. Hence, there exists some $j$ such that $\lambda_j \notin \mathcal{P}_\eta$ as $\tau \to \infty$. $\qquad\square$

*Proof of Theorem 5.6.* Similar to the proof of the Theorem 5.2, we analyze the necessary and sufficient condition for each set of eigenvalues $\lambda_j$ of $\boldsymbol{H}_\tau^*$ categorized in Theorem 4.3 to lie in $\mathcal{P}_\eta$ below, and combining them leads to the claimed statement.

(i) Consider $\lambda_j$ with $j \in \mathcal{I}$. Suppose that $s_0 < \frac{1}{2L}$. Then, by Lemma H.2, for any step size $\eta$ such that $s_0 < \frac{\eta}{2} < \frac{1}{2L}$, it holds that $\lambda_j(\epsilon) \in \mathcal{P}_\eta$ whenever $\tau$ is sufficiently large. Conversely, suppose that $s_0 \geq \frac{1}{2L}$. Then, for any $\eta$ such that $0 < \eta < \frac{1}{L}$, we have

$\frac{\eta}{2} < s_0$. In that case, by Lemma H.2, for any sufficiently large timescale $\tau > 0$ there exists some $j \in \mathcal{I}$ such that $\lambda_j(\epsilon) \notin \mathcal{P}_\eta$. Therefore, $s_0 < \frac{1}{2L}$ if and only if there exists some $0 < \eta^\star < \frac{1}{L}$ such that for any $\eta$ satisfying $\eta^\star < \eta < \frac{1}{L}$, $\tau$ being sufficiently large implies $\lambda_j(\epsilon) \in \mathcal{P}_\eta$ for all $j \in \mathcal{I}$.

(ii) Consider $\lambda_j = \epsilon\mu_k + o(\epsilon)$ for some $k$. Note that they converge to zero asymptotically along the real axis. In particular, whenever $\epsilon$ is sufficiently small, if $\mu_k > 0$ then $\lambda_j(\epsilon) \in \mathcal{P}_\eta$, and if $\mu_k < 0$ then $\lambda_j(\epsilon) \notin \mathcal{P}_\eta$. Recall that in Theorem 4.3 we have established that $\mu_k$ are the eigenvalues of the restricted Schur complement $\boldsymbol{S}_{\mathrm{res}}$. One can then deduce that $\boldsymbol{S}_{\mathrm{res}} \succeq \boldsymbol{0}$ if and only if there exists some $\tau^\star$ where $\tau > \tau^\star$ implies that every $\lambda_j$ of order $\Theta(\epsilon)$ satisfies $\lambda_j(\epsilon) \in \mathcal{P}_\eta$.

(iii) Consider $\lambda_j = \nu_k + o(1)$ for some $k$. By the proof of Theorem 5.2, we have $\|\lambda_j(\epsilon)\| \leq L$ for eigenvalues of $\boldsymbol{H}_\tau^*$ and $\|\nu_k\| \leq L$ for the eigenvalues of $-\boldsymbol{B}$. Now, let $0 < \eta < \frac{1}{L}$, and suppose that $\boldsymbol{B} \npreceq \boldsymbol{0}$, so that there exists some $k$ such that $-L \leq \nu_k < 0$. Then, since the half-open interval $[-L, 0)$ is contained in the interior of complement of $\mathcal{P}_\eta$, for sufficiently large $\tau$ we will have $\lambda_j(\epsilon) \notin \mathcal{P}_\eta$. Conversely, if $\boldsymbol{B} \preceq \boldsymbol{0}$, then $\nu_k > 0$ for all $k$, so as a consequence, whenever $\eta > 0$ and $\tau$ is sufficiently large, we will have $\lambda_j(\epsilon) \in \mathcal{P}_\eta$ for all $j$ such that $\lambda_j(\epsilon) = \nu_k + o(1)$. Therefore, $\boldsymbol{B} \preceq \boldsymbol{0}$ if and only if there exists some $0 < \eta < \frac{1}{L}$ such that $\tau$ being sufficiently large implies every $\lambda_j$ of order $\Theta(1)$ satisfies $\lambda_j(\epsilon) \in \mathcal{P}_\eta$.

Combining all the discussions made, we conclude that $\boldsymbol{S}_{\mathrm{res}} \succeq \boldsymbol{0}$, $\boldsymbol{B} \preceq \boldsymbol{0}$, and $s_0 < \frac{1}{2L}$ if and only if there exists some $0 < \eta^\star < \frac{1}{L}$ such that, for any $\eta$ satisfying $\eta^\star < \eta < \frac{1}{L}$, $\tau$ being sufficiently large implies $\lambda_j(\tau) \in \mathcal{P}_\eta$ for all $j$. The conclusion then follows from Proposition 5.5. □

## H.5 Additional stability conditions of two-timescale EG in discrete-time

**Theorem H.3.** *Let $\boldsymbol{z}^*$ be an equilibrium point. Suppose Assumptions 1 and 2′ hold.*

(i) *(Sufficient condition) Suppose that $\boldsymbol{z}^*$ satisfies $\boldsymbol{S} \succeq \boldsymbol{0}$ and $\boldsymbol{B} \preceq \boldsymbol{0}$. Then for any step size $0 < \eta < 1/L$, the point $\boldsymbol{z}^*$ is a strict linearly stable point of $\infty$-EG.*

(ii) *(Necessary condition) Suppose that $\boldsymbol{z}^*$ is a strict linearly stable equilibrium point of $\infty$-EG for any step size $0 < \eta < 1/L$. Then, it satisfies $\boldsymbol{S}_{\mathrm{res}} \succeq \boldsymbol{0}$ and $\boldsymbol{B} \preceq \boldsymbol{0}$.*

**Theorem H.4.** *Under Assumptions 1 and 2′, suppose that all the values of $\sigma_j$ are distinct. Then, an equilibrium point $\boldsymbol{z}^*$ satisfies $\boldsymbol{S}_{\mathrm{res}} \succeq \boldsymbol{0}$, $\boldsymbol{B} \preceq \boldsymbol{0}$, and $\boldsymbol{u}_j^\top \boldsymbol{S} \boldsymbol{u}_j \geq 0$ for every left singular vector $\boldsymbol{u}_j$ of $\boldsymbol{C}_2$ if and only if $\boldsymbol{z}^*$ is a strict linearly stable point of $\infty$-EG for any $0 < \eta < 1/L$.*

## H.6 Proof of Theorem H.3

*Proof of Theorem H.3(i).* For any $j$ such that $\lambda_j(\epsilon) = \nu_k + o(1)$, $\boldsymbol{B} \preceq 0$ implies that $\nu_k > 0$. Since $|\nu_k| \leq L$ and $0 < \eta < \frac{1}{L}$ implies that $\mathcal{P}_\eta$ contains a half-open interval $(0, L]$, for any sufficiently large $\tau$ we have $\lambda_j(\epsilon) \in \mathcal{P}_\eta$.

And for any $j$ such that $\lambda_j(\epsilon) = \epsilon\mu_k + o(\epsilon)$, since $\boldsymbol{A} - \boldsymbol{C}\boldsymbol{B}^\dagger\boldsymbol{C}^\top \succeq 0$ implies $\boldsymbol{S}_{\mathrm{res}} \succeq 0$, for any sufficiently large $\tau$ we have $\lambda_j(\epsilon) \in \mathcal{P}_\eta$.

On the other hand, for any $j \in \mathcal{I}$ proof of Theorem 5.3(i) implies that $\lim_{\epsilon \to 0+} \frac{\xi_j \epsilon^{e_j-1}}{\sigma_j^2} \geq 0$ hence $s_0 = \max_{j \in \mathcal{I}}(-\iota_j) \leq 0$. The conclusion then follows from Lemma H.2. □

To prove Theorem H.3(ii), we use the following technical lemma. For convenience, let us define a subset of the complex plane, for a positive constant $a > 0$,

$$\mathcal{O}_a := \left\{ z \in \mathbb{C} \ : \ |z + a| < \frac{a}{2} \right\},$$

which an open disk centered at $-a$ with radius $\frac{a}{2}$.

**Lemma H.5.** $\mathcal{O}_a \cap \mathcal{P}_\eta = \varnothing$ *for any positive constant $a > 0$.*

*Proof.* Noticing that $\mathcal{O}_a$ lies in the left half plane, the only region to care about is $\mathbb{C}_-^\circ$. Thus, if the disk $\mathcal{O}_a$ and the peanut-shaped $\mathcal{P}_\eta$ do not intersect on that region, then the assertion follows immediately.

Consider a circle centered at origin with radius $R$, denoted by $\mathcal{O}^*$. Then, if the circle $\mathcal{O}^*$ and the boundary of $\mathcal{O}_a$ intersects, then it intersects at a point $z_1$ with a real part $\operatorname{Re} z_1 = -(\frac{R^2}{2a} + \frac{3a}{8})$. Similarly, if the circle $\mathcal{O}^*$ and the boundary of $\mathcal{P}_\eta$ intersects, then it intersects at a point $z_2$ with a real part $\operatorname{Re} z_2 = \frac{1}{4\eta} + \frac{\eta R^2}{4} - \sqrt{\frac{1}{16\eta^2} - \frac{3\eta^2 R^4}{16} + \frac{3R^2}{8}}$. For $\mathcal{O}_a$ and $\mathcal{P}_\eta$ to have an overlap, there must exist some $R$ such that $\operatorname{Re} z_1 = \operatorname{Re} z_2$. We show that such $R$ does not exist for any positive $a$ and $\eta$, by proving the following statement

$$\operatorname{Re} z_2 - \operatorname{Re} z_1 = \frac{1}{4\eta} + \frac{\eta R^2}{4} + \frac{R^2}{2a} + \frac{3a}{8} - \sqrt{\frac{1}{16\eta^2} - \frac{3\eta^2 R^4}{16} + \frac{3R^2}{8}} > 0$$

for any positivie values $a$, $\eta$ and $R$. This is done by showing that the following

$$\left(\frac{1}{4\eta} + \frac{\eta R^2}{4} + \frac{R^2}{2a} + \frac{3a}{8}\right)^2 - \left(\frac{1}{16\eta^2} - \frac{3\eta^2 R^4}{16} + \frac{3R^2}{8}\right)$$
$$= \frac{9a^2}{64} + \frac{3a}{16\eta} + R^2\left(\frac{3a\eta}{16} + \frac{1}{4a\eta} + \frac{1}{8}\right) + R^4\left(\frac{1}{4a^2} + \frac{\eta}{4a} + \frac{\eta^2}{4}\right) > 0$$

holds for any positive $a$, $\eta$ and $R$, and this concludes the proof. $\qquad\square$

*Proof of Theorem H.3(ii).* Suppose that $\boldsymbol{S}_{\mathrm{res}} \not\succeq \boldsymbol{0}$, or equivalently, there exists some $k$ such that $\mu_k < 0$. Then, by Theorem 4.3, we have $\lambda_j(\epsilon) = \epsilon\mu_k + o(\epsilon)$ for some $j$. So by choosing sufficiently large $\tau$, Lemma H.5 implies that $\lambda_j(\epsilon) \notin \mathcal{P}_\eta$. However by Proposition 5.5 this implies that $\boldsymbol{z}^*$ is not strict linearly stable, which is absurd. Therefore, it is necessary that $\boldsymbol{S}_{\mathrm{res}} \succeq \boldsymbol{0}$.

Next, suppose that $\boldsymbol{B} \not\preceq \boldsymbol{0}$, or equivalently, there exists some $k$ such that $\nu_k < 0$. Then, by Theorem 4.3, as $\lambda_j(\epsilon)$ converges to $\nu_k$ for some $j$, if we choose sufficiently large $\tau$, we must have $\lambda_j(\epsilon) \notin \mathcal{P}_\eta$. However again by Proposition 5.5 this implies that $\boldsymbol{z}^*$ is not strict linearly stable, which is absurd. Therefore, it is necessary that $\boldsymbol{B} \preceq \boldsymbol{0}$. $\qquad\square$

### H.7    PROOF OF THEOREM H.4

Note that $\zeta_j = \frac{1}{2}\boldsymbol{U}_j^\top(\boldsymbol{A} - \boldsymbol{C}\boldsymbol{B}^\dagger\boldsymbol{C}^\top)\boldsymbol{U}_j$ by the proof of Theorem 5.4. Then the fact that $s_0 \leq 0$ directly follows from the assumption. Lemma H.2 implies that the eigenvalues of the form $\lambda_j(\epsilon) = \pm i\sigma_j\sqrt{\epsilon} + o(\sqrt{\epsilon})$ lie in $\mathcal{P}_\eta$ for any positive $\eta$. And the remaining eigenvalues lie in $\mathcal{P}_\eta$ if and only if $\boldsymbol{S}_{\mathrm{res}} \succeq \boldsymbol{0}$ and $\boldsymbol{B} \preceq \boldsymbol{0}$. The assertion then follows from the Proposition 5.5.

Conversely, suppose that $\boldsymbol{S}_{\mathrm{res}} \not\succeq \boldsymbol{0}$ or $\boldsymbol{B} \not\preceq \boldsymbol{0}$. Then unstability of the $\boldsymbol{z}^*$ directly follows from the proof of Theorem H.3(ii).

Next, suppose that there exists some $j$ such that $\zeta_j < 0$. Then the fact that $s_0 > 0$ directly follows from Lemma G.12. Hence for step size $\frac{\eta}{2} < s_0$, the $\boldsymbol{z}^*$ is not strict linearly stable point by Lemma H.2 and Proposition 5.5.

## I    PROOF FOR SECTION 5.4

### I.1    PROOF OF THEOREM 5.7

To prove Theorem 5.7 we need the following results.

**Proposition I.1.** *Under Assumption 1 and $0 < \eta < \frac{\sqrt{5}-1}{2L}$, we have $\det(D\boldsymbol{w}_\tau(\boldsymbol{z})) \neq 0$ for all $\boldsymbol{z}$.*

*Proof.* We begin by observing that

$$D\boldsymbol{w}_\tau(z) = \boldsymbol{I} - \eta\boldsymbol{\Lambda}_\tau D\boldsymbol{F}(\boldsymbol{z} - \eta\boldsymbol{\Lambda}_\tau\boldsymbol{F}(\boldsymbol{z}))(\boldsymbol{I} - \eta\boldsymbol{\Lambda}_\tau D\boldsymbol{F}(\boldsymbol{z})).$$

Since $\|\mathbf{\Lambda}_\tau\| \le 1$, Assumption 1 implies that $\|\mathbf{\Lambda}_\tau D\boldsymbol{F}\| \le \|\mathbf{\Lambda}_\tau\| \|D\boldsymbol{F}\| \le L$. Therefore, whenever $0 < \eta < \frac{\sqrt{5}-1}{2L}$, for clarity by letting $\check{\boldsymbol{z}} = \boldsymbol{z} - \eta\mathbf{\Lambda}_\tau\boldsymbol{F}(\boldsymbol{z})$, we obtain the bound

$$
\begin{aligned}
\|\eta\mathbf{\Lambda}_\tau D\boldsymbol{F}(\boldsymbol{z} - \eta\mathbf{\Lambda}_\tau\boldsymbol{F}(\boldsymbol{z}))(\boldsymbol{I} - \eta\mathbf{\Lambda}_\tau D\boldsymbol{F}(\boldsymbol{z}))\| &\le \|\eta\mathbf{\Lambda}_\tau D\boldsymbol{F}(\boldsymbol{z} - \eta\mathbf{\Lambda}_\tau\boldsymbol{F}(\boldsymbol{z}))\| \|\boldsymbol{I} - \eta\mathbf{\Lambda}_\tau D\boldsymbol{F}(\boldsymbol{z})\| \\
&\le \|\eta\mathbf{\Lambda}_\tau D\boldsymbol{F}(\check{\boldsymbol{z}})\| \left(1 + \|\eta\mathbf{\Lambda}_\tau D\boldsymbol{F}(\boldsymbol{z})\|\right) \\
&\le \eta L \left(1 + \eta L\right) \\
&< 1.
\end{aligned}
$$

It follows that any eigenvalue of $\eta\mathbf{\Lambda}_\tau D\boldsymbol{F}(\boldsymbol{z} - \eta\mathbf{\Lambda}_\tau\boldsymbol{F}(\boldsymbol{z}))(\boldsymbol{I} - \eta\mathbf{\Lambda}_\tau D\boldsymbol{F}(\boldsymbol{z}))$ has an absolute value strictly less than 1, and hence, 0 cannot be an eigenvalue of $D\boldsymbol{w}_\tau(\boldsymbol{z})$. The conclusion is now immediate. $\qquad\square$

**Proposition I.2.** *For any $\boldsymbol{z}^* \in \mathcal{T}^*$, under Assumption 2, there exists a positive constant $\tau^\star > 0$ such that $\boldsymbol{z}^* \in \mathcal{A}^*(\boldsymbol{w}_\tau)$ for any $\tau > \tau^\star$.*

*Proof.* By Proposition 5.5, we have

$$
\begin{aligned}
\mathcal{A}^*(\boldsymbol{w}_\tau) &= \{\boldsymbol{z}^* \ : \ \boldsymbol{z}^* = \boldsymbol{w}_\tau(\boldsymbol{z}^*), \ \rho(D\boldsymbol{w}_\tau(\boldsymbol{z}^*)) > 1\} \\
&= \{\boldsymbol{z}^* \ : \ \boldsymbol{z}^* = \boldsymbol{w}_\tau(\boldsymbol{z}^*), \ \exists\lambda \in \mathrm{spec}(\boldsymbol{H}_\tau(\boldsymbol{z}^*)) \ \text{s.t.} \ \lambda \notin \mathcal{P}_\eta\}.
\end{aligned}
$$

For any strict non-minimax point $\boldsymbol{z}^* \in \mathcal{T}^*$, either $\boldsymbol{S}_{\mathrm{res}}(\boldsymbol{z}^*)$ or $-\boldsymbol{B}(\boldsymbol{z}^*)$ has at least one strictly negative eigenvalue. First, suppose that $\boldsymbol{S}_{\mathrm{res}}(\boldsymbol{z}^*)$ has a strictly negative eigenvalue $\mu < 0$. By Theorem 4.3, there exists a constant $\tau^\star$ such that at least one eigenvalue of $\boldsymbol{H}_\tau^*$ lies in a disk $\mathcal{D}_{\mu\epsilon}^\sharp$ for any $\tau > \tau^\star$. So by Lemma H.5, we would have $\mathcal{D}_{\mu\epsilon}^\sharp \cap \mathcal{P}_\eta = \varnothing$. On the other hand, suppose that $-\boldsymbol{B}(\boldsymbol{z}^*)$ has a strictly negative eigenvalue $\nu < 0$. Similarly, by Theorem 4.3, there exists a constant $\tau^\star$ such that at least one eigenvalue of $\boldsymbol{H}_\tau^*$ lies in a disk $\mathcal{D}_\nu^\sharp$ for any $\tau > \tau^\star$. So by Lemma H.5, we would have $\mathcal{D}_\nu^\sharp \cap \mathcal{P}_\eta = \varnothing$. Therefore, we can conclude that for any $\boldsymbol{z}^* \in \mathcal{T}^*$, there exists a constant $\tau^\star$ such that $\boldsymbol{z}^* \in \mathcal{A}^*(\boldsymbol{w}_\tau)$ for any $\tau > \tau^\star$. $\qquad\square$

*Proof of Theorem 5.7.* Because $\boldsymbol{z}^* \in \mathcal{A}^*(\boldsymbol{w}_\tau)$ implies

$$
\left\{\boldsymbol{z}_0 : \lim_{k\to\infty}\boldsymbol{w}^k(\boldsymbol{z}_0) = \boldsymbol{z}^*\right\} \subset \left\{\boldsymbol{z}_0 : \lim_{k\to\infty}\boldsymbol{w}^k(\boldsymbol{z}_0) \in \mathcal{A}^*(\boldsymbol{w}_\tau)\right\},
$$

by Theorem 2.2, there exists a positive constant $\tau^\star > 0$ such that

$$
\mu\left(\left\{\boldsymbol{z}_0 : \lim_{k\to\infty}\boldsymbol{w}^k(\boldsymbol{z}_0) = \boldsymbol{z}^*\right\}\right) = 0
$$

for any $\tau > \tau^\star$.

Moreover, if $\mathcal{T}^*$ is finite, then a maximum of $\tau^\star$ for all $\boldsymbol{z}^* \in \mathcal{T}^*$ is also finite. Let us denote such maximum by $\tau_{\mathrm{max}}^*$. Then, for any $\tau > \tau_{\mathrm{max}}^\star$ we have $\mathcal{T}^* \subset \mathcal{A}^*(\boldsymbol{w}_\tau)$. This implies that

$$
\left\{\boldsymbol{z}_0 : \lim_{k\to\infty}\boldsymbol{w}^k(\boldsymbol{z}_0) \in \mathcal{T}^*\right\} \subset \left\{\boldsymbol{z}_0 : \lim_{k\to\infty}\boldsymbol{w}^k(\boldsymbol{z}_0) \in \mathcal{A}^*(\boldsymbol{w}_\tau)\right\},
$$

for any $\tau > \tau_{\mathrm{max}}^\star$, and by Theorem 2.2, we can conclude that

$$
\mu\left(\left\{\boldsymbol{z}_0 : \lim_{k\to\infty}\boldsymbol{w}^k(\boldsymbol{z}_0) \in \mathcal{T}^*\right\}\right) = 0 \qquad\qquad \square.
$$

## J  GLOBAL CONVERGENCE ANALYSIS OF TWO-TIMESCALE EG

Let us consider the *Minty variational inequality* (MVI) condition, defined as follows, as a requirement that holds for nonconvex-nonconcave setting.

**Assumption 4** (Minty variational inequality). *For the saddle-gradient operator $\boldsymbol{F}$, a stationary point $\boldsymbol{z}^* = (\boldsymbol{x}^*, \boldsymbol{y}^*)$ under consideration satisfies $\langle\boldsymbol{F}(\boldsymbol{z}), \boldsymbol{z} - \boldsymbol{z}^*\rangle \ge 0$ for all $\boldsymbol{z}$.*

Let us now show that the two-timescale EG

$$\begin{cases} \boldsymbol{u}_k = \boldsymbol{x}_k - \dfrac{\eta}{\tau}\nabla_{\boldsymbol{x}}\,f(\boldsymbol{x}_k, \boldsymbol{y}_k), \\[2mm] \boldsymbol{v}_k = \boldsymbol{y}_k + \eta\nabla_{\boldsymbol{y}}\,f(\boldsymbol{x}_k, \boldsymbol{y}_k) \end{cases}, \qquad \begin{cases} \boldsymbol{x}_{k+1} = \boldsymbol{x}_k - \dfrac{\eta}{\tau}\nabla_{\boldsymbol{x}}\,f(\boldsymbol{u}_k, \boldsymbol{v}_k) \\[2mm] \boldsymbol{y}_{k+1} = \boldsymbol{y}_k + \eta\nabla_{\boldsymbol{y}}\,f(\boldsymbol{u}_k, \boldsymbol{v}_k) \end{cases}.$$

finds a stationary point satisfying the MVI condition, built upon the proof for the plain EG (Diakonikolas et al., 2021). Note that Diakonikolas et al. (2021) consider a variant of EG, named EG+, and shows that it works for a even weaker condition, namely the *weak MVI* condition. The ideas behind our dynamical system analysis and the proof in the section are general enough to be further extended to EG+ and the weak MVI condition, but we leave a more detailed study in this direction as a future work.

The objective function $f$ is said to be *L-smooth* if the inequality $\|\nabla f(\boldsymbol{z}) - \nabla f(\boldsymbol{w})\| \le L\|\boldsymbol{z} - \boldsymbol{w}\|$ holds for any two points $\boldsymbol{z}$ and $\boldsymbol{w}$ in the domain of $f$. If $f \in C^2$, then it is known that $f$ is $L$-smooth if and only if $\|\nabla^2 f\| \le L$; see, *e.g.*, (Beck, 2017, Theorem 5.12). As $\nabla f$ and $\boldsymbol{F}$ are equal up to a sign difference, one can observe that Assumption 1 is equivalent to the assertion that $f$ is $L$-smooth.

**Theorem J.1.** *Suppose that $f$ is L-smooth, and Assumption 4 holds. Then, the iterates $\{\boldsymbol{x}_k\}_{k\ge 0}$ and $\{\boldsymbol{y}_k\}_{k\ge 0}$ of two-timescale EG with $\eta < \frac{1}{L}$ and $\tau \ge 1$ satisfies $\|\nabla_{\boldsymbol{x}}\,f(\boldsymbol{x}_k, \boldsymbol{y}_k)\| \to 0$ and $\|\nabla_{\boldsymbol{y}}\,f(\boldsymbol{x}_k, \boldsymbol{y}_k)\| \to 0$ as $k \to \infty$.*

*Proof.* We have the inequality

$$\|\boldsymbol{x}_{k+1} - \boldsymbol{x}^*\|^2 + \frac{1}{\tau}\|\boldsymbol{y}_{k+1} - \boldsymbol{y}^*\|^2$$

$$= \left\|\boldsymbol{x}_k - \frac{\eta}{\tau}\nabla_{\boldsymbol{x}}f(\boldsymbol{u}_k, \boldsymbol{v}_k) - \boldsymbol{x}^*\right\|^2 + \frac{1}{\tau}\|\boldsymbol{y}_k + \eta\nabla_{\boldsymbol{y}}f(\boldsymbol{u}_k, \boldsymbol{v}_k) - \boldsymbol{y}^*\|^2$$

$$= \|\boldsymbol{x}_k - \boldsymbol{x}^*\|^2 + \frac{1}{\tau}\|\boldsymbol{y}_k - \boldsymbol{y}^*\|^2 - \frac{2\eta}{\tau}(\langle\nabla_{\boldsymbol{x}}f(\boldsymbol{u}_k, \boldsymbol{v}_k), \boldsymbol{x}_k - \boldsymbol{x}^*\rangle - \langle\nabla_{\boldsymbol{y}}f(\boldsymbol{u}_k, \boldsymbol{v}_k), \boldsymbol{y}_k - \boldsymbol{y}^*\rangle)$$

$$\quad + \frac{\eta^2}{\tau^2}\|\nabla_{\boldsymbol{x}}f(\boldsymbol{u}_k, \boldsymbol{v}_k)\|^2 + \frac{\eta^2}{\tau}\|\nabla_{\boldsymbol{y}}f(\boldsymbol{u}_k, \boldsymbol{v}_k)\|^2$$

$$\le \|\boldsymbol{x}_k - \boldsymbol{x}^*\|^2 + \frac{1}{\tau}\|\boldsymbol{y}_k - \boldsymbol{y}^*\|^2 - \frac{2\eta}{\tau}(\langle\nabla_{\boldsymbol{x}}f(\boldsymbol{u}_k, \boldsymbol{v}_k), \boldsymbol{x}_k - \boldsymbol{u}_k\rangle - \langle\nabla_{\boldsymbol{y}}f(\boldsymbol{u}_k, \boldsymbol{v}_k), \boldsymbol{y}_k - \boldsymbol{v}_k\rangle)$$

$$\quad + \frac{\eta^2}{\tau^2}\|\nabla_{\boldsymbol{x}}f(\boldsymbol{u}_k, \boldsymbol{v}_k)\|^2 + \frac{\eta^2}{\tau}\|\nabla_{\boldsymbol{y}}f(\boldsymbol{u}_k, \boldsymbol{v}_k)\|^2$$

$$= \|\boldsymbol{x}_k - \boldsymbol{x}^*\|^2 + \frac{1}{\tau}\|\boldsymbol{y}_k - \boldsymbol{y}^*\|^2 - \frac{2\eta}{\tau}\left(\left\langle\nabla_{\boldsymbol{x}}f(\boldsymbol{u}_k, \boldsymbol{v}_k), \frac{\eta}{\tau}\nabla_{\boldsymbol{x}}f(\boldsymbol{x}_k, \boldsymbol{y}_k)\right\rangle - \langle\nabla_{\boldsymbol{y}}f(\boldsymbol{u}_k, \boldsymbol{v}_k), -\eta\nabla_{\boldsymbol{y}}f(\boldsymbol{x}_k, \boldsymbol{y}_k)\rangle\right)$$

$$\quad + \frac{\eta^2}{\tau^2}\|\nabla_{\boldsymbol{x}}f(\boldsymbol{u}_k, \boldsymbol{v}_k)\|^2 + \frac{\eta^2}{\tau}\|\nabla_{\boldsymbol{y}}f(\boldsymbol{u}_k, \boldsymbol{v}_k)\|^2$$

$$= \|\boldsymbol{x}_k - \boldsymbol{x}^*\|^2 + \frac{1}{\tau}\|\boldsymbol{y}_k - \boldsymbol{y}^*\|^2 + \frac{\eta^2}{\tau^2}\left(\|\nabla_{\boldsymbol{x}}f(\boldsymbol{u}_k, \boldsymbol{v}_k) - \nabla_{\boldsymbol{x}}f(\boldsymbol{x}_k, \boldsymbol{y}_k)\|^2 - \|\nabla_{\boldsymbol{x}}f(\boldsymbol{x}_k, \boldsymbol{y}_k)\|^2\right)$$

$$\quad + \frac{\eta^2}{\tau}\left(\|\nabla_{\boldsymbol{y}}f(\boldsymbol{u}_k, \boldsymbol{v}_k) - \nabla_{\boldsymbol{y}}f(\boldsymbol{x}_k, \boldsymbol{y}_k)\|^2 - \|\nabla_{\boldsymbol{y}}f(\boldsymbol{x}_k, \boldsymbol{y}_k)\|^2\right)$$

$$\le \|\boldsymbol{x}_k - \boldsymbol{x}^*\|^2 + \frac{1}{\tau}\|\boldsymbol{y}_k - \boldsymbol{y}^*\|^2 + \frac{\eta^2}{\tau}(\|\nabla_{\boldsymbol{x}}f(\boldsymbol{u}_k, \boldsymbol{v}_k) - \nabla_{\boldsymbol{x}}f(\boldsymbol{x}_k, \boldsymbol{y}_k)\|^2 + \|\nabla_{\boldsymbol{y}}f(\boldsymbol{u}_k, \boldsymbol{v}_k) - \nabla_{\boldsymbol{y}}f(\boldsymbol{x}_k, \boldsymbol{y}_k)\|^2)$$

$$\quad - \frac{\eta^2}{\tau^2}\|\nabla_{\boldsymbol{x}}f(\boldsymbol{x}_k, \boldsymbol{y}_k)\|^2 - \frac{\eta^2}{\tau}\|\nabla_{\boldsymbol{y}}f(\boldsymbol{x}_k, \boldsymbol{y}_k)\|^2$$

$$\le \|\boldsymbol{x}_k - \boldsymbol{x}^*\|^2 + \frac{1}{\tau}\|\boldsymbol{y}_k - \boldsymbol{y}^*\|^2 + \frac{\eta^2 L^2}{\tau}(\|\boldsymbol{u}_k - \boldsymbol{x}_k\|^2 + \|\boldsymbol{v}_k - \boldsymbol{y}_k\|^2)$$

$$\quad - \frac{\eta^2}{\tau^2}\|\nabla_{\boldsymbol{x}}f(\boldsymbol{x}_k, \boldsymbol{y}_k)\|^2 - \frac{\eta^2}{\tau}\|\nabla_{\boldsymbol{y}}f(\boldsymbol{x}_k, \boldsymbol{y}_k)\|^2$$

$$\le \|\boldsymbol{x}_k - \boldsymbol{x}^*\|^2 + \frac{1}{\tau}\|\boldsymbol{y}_k - \boldsymbol{y}^*\|^2 - \frac{\eta^2}{\tau^2}\left(1 - \frac{\eta^2 L^2}{\tau}\right)\|\nabla_{\boldsymbol{x}}f(\boldsymbol{x}_k, \boldsymbol{y}_k)\|^2 - \frac{\eta^2}{\tau}\left(1 - \eta^2 L^2\right)\|\nabla_{\boldsymbol{y}}f(\boldsymbol{x}_k, \boldsymbol{y}_k)\|^2$$

where the first inequality uses Assumption 4, the second inequality uses $\tau \ge 1$, the third inequality uses the $L$-smoothness of $f$, and the last inequality uses the update rules.

By taking a telescoping summation, we have

$$\frac{\eta^2}{\tau^2}\left(1-\frac{\eta^2 L^2}{\tau}\right)\sum_{i=0}^{k}\|\nabla_{\boldsymbol{x}}f(\boldsymbol{x}_i,\boldsymbol{y}_i)\|^2 + \frac{\eta^2}{\tau}\left(1-\eta^2 L^2\right)\sum_{i=0}^{k}\|\nabla_{\boldsymbol{y}}f(\boldsymbol{x}_i,\boldsymbol{y}_i)\|^2$$

$$\leq \|\boldsymbol{x}_0-\boldsymbol{x}^*\|^2 + \frac{1}{\tau}\|\boldsymbol{y}_0-\boldsymbol{y}^*\|^2 - \|\boldsymbol{x}_{k+1}-\boldsymbol{x}^*\|^2 - \frac{1}{\tau}\|\boldsymbol{y}_{k+1}-\boldsymbol{y}^*\|^2$$

$$\leq \|\boldsymbol{x}_0-\boldsymbol{x}^*\|^2 + \frac{1}{\tau}\|\boldsymbol{y}_0-\boldsymbol{y}^*\|^2.$$

Since both the series $\sum_{i=0}^{k}\|\nabla_{\boldsymbol{x}}f(\boldsymbol{x}_i,\boldsymbol{y}_i)\|^2$ and $\sum_{i=0}^{k}\|\nabla_{\boldsymbol{y}}f(\boldsymbol{x}_i,\boldsymbol{y}_i)\|^2$ are monotone and bounded, they converge. Therefore we have $\|\nabla_{\boldsymbol{x}}f(\boldsymbol{x}_k,\boldsymbol{y}_k)\|^2 \to 0$ and $\|\nabla_{\boldsymbol{y}}f(\boldsymbol{x}_k,\boldsymbol{y}_k)\|^2 \to 0$ as $k \to \infty$. $\square$

In general, even when a sequence $\{\boldsymbol{z}_k\}_{k\geq 0}$ satisfies $\lim_{k\to 0}\|\boldsymbol{F}\boldsymbol{z}_k\| \to 0$, there is nothing we can say about the convergence of the sequence itself. In other words, the statement of Theorem J.1 alone does not guarantee the convergence of the iterates. However, what the condition $\lim_{k\to 0}\|\boldsymbol{F}\boldsymbol{z}_k\| \to 0$ does guarantee, regardless of the MVI condition, is that any accumulation point of the sequence $\{\boldsymbol{z}_k\}_{k\geq 0}$ becomes a stationary point of the objective function $f$. Recall that a point $\boldsymbol{z}'$ is called an accumulation point of a sequence $\{\boldsymbol{z}_k\}_{k\geq 0}$ if, for any given $\varepsilon > 0$, there exist infinitely many indices $k \geq 0$ such that $\|\boldsymbol{z}_k - \boldsymbol{z}'\| < \varepsilon$.

**Lemma J.2.** *Suppose that $f$ is L-smooth, and let $\{\boldsymbol{z}_k\}_{k\geq 0}$ be a sequence such that $\|\boldsymbol{F}(\boldsymbol{z}_k)\| \to 0$ as $k \to \infty$. Then, any accumulation point of the sequence $\{\boldsymbol{z}_k\}_{k\geq 0}$ is a stationary point of the objective function $f$.*

*Proof.* Let $\boldsymbol{z}'$ be an accumulation point of the sequence $\{\boldsymbol{z}_k\}_{k\geq 0}$. In that case, there exists a subsequence $\{\boldsymbol{z}_{k_j}\}_{j\geq 0}$ of $\{\boldsymbol{z}_k\}_{k\geq 0}$ that converges to $\boldsymbol{z}'$ (Tao, 2016, Proposition 1.4.5). Then for any $j \geq 0$, by triangle inequality it holds that

$$0 \leq \|\boldsymbol{F}(\boldsymbol{z}')\| \leq \|\boldsymbol{F}(\boldsymbol{z}') - \boldsymbol{F}(\boldsymbol{z}_{k_j})\| + \|\boldsymbol{F}(\boldsymbol{z}_{k_j})\|$$

$$\leq L\|\boldsymbol{z}' - \boldsymbol{z}_{k_j}\| + \|\boldsymbol{F}(\boldsymbol{z}_{k_j})\|.$$

As both $\|\boldsymbol{z}' - \boldsymbol{z}_{k_j}\|$ and $\|\boldsymbol{F}(\boldsymbol{z}_{k_j})\|$ converge to 0 as $j \to \infty$, we must have $\|\boldsymbol{F}(\boldsymbol{z}')\| = 0$. $\square$

Finally, let us demonstrate that all local minimax points of the quadratic function $f_{\mathsf{q}}$ used in Example 4 satisfy the MVI condition.

**Lemma J.3.** *Consider the quadratic function $f(x_1, x_2, y_1, y_2, y_3) = \frac{1}{2}x_1^2 - \frac{1}{2}y_1^2 - \frac{1}{2}y_3^2 + x_2y_2 + y_1y_3$. Any stationary point of $f$ is a local minimax point that satisfies the MVI condition.*

*Proof.* We already saw in Section D.3 that the stationary points of $f$ are exactly the points of the form $(0, 0, t, 0, t)$ where $t \in \mathbb{R}$, and that every such point is a local minimax point.

Now fix any $t \in \mathbb{R}$ and let $\boldsymbol{z}^* = (0, 0, t, 0, t)$. One way to show that this point satisfies the MVI condition is to note that $\boldsymbol{F}\boldsymbol{z}^* = \boldsymbol{0}$, while $f$ is in fact convex-concave hence its saddle gradient $\boldsymbol{F}$ becomes a so-called *monotone* operator; see (Ryu & Yin, 2022, Section 2.2.3). Alternatively, one can also verify this by a direct computation: for any $\boldsymbol{z} = (x_1, x_2, y_1, y_2, y_3)$, it holds that

$$\langle \boldsymbol{F}\boldsymbol{z}, \boldsymbol{z} - \boldsymbol{z}^* \rangle = \langle (x_1, y_2, y_1 - y_3, -x_2, -y_1 + y_3), (x_1, x_2, y_1 - t, y_2, y_3 - t) \rangle$$

$$= x_1^2 + y_1^2 + y_3^2 - 2y_1y_3$$

$$= x_1^2 + (y_1 - y_3)^2$$

$$\geq 0$$

so the Minty variational inequality holds. $\square$

