# OpenReview forum: "Two-timescale Extragradient for Finding Local Minimax Points"
_ICLR.cc/2024/Conference — ICLR 2024 poster_

### Official Review · Reviewer_8GfH · 2023-10-26

**Soundness:** 3 good
**Presentation:** 2 fair
**Contribution:** 3 good
**Rating:** 6
**Confidence:** 3

**Summary:**

The paper studies the local minimax points in minimax optimization when the objective is smooth and possibly nonconvex-nonconcave. They first examine the previous second-order necessary and sufficient conditions for minimax points and propose a new necessary condition that allows $\nabla^2_{yy} f$ to be degenerate. Using dynamical system tools, they further show that two-timescale extragradient converges to points satisfying this condition under mild assumptions and almost surely avoids strict non-minimax points, while two-timescale GDA may avoid some local minimax points that are degenerate.

**Strengths:**

The paper proposes a new second-order necessary condition for local minimax points, which allows the Hessian $\nabla^2_{yy} f$ to be degenerate. For some local minimax points with degenerate $\nabla^2_{yy} f$, it is shown that two-timescale extragradient will converge to them while two-timescale GDA may avoid them. This provides a better understanding of the behaviors of classical first-order algorithms for minimax optimization. I believe this contribution is interesting enough to the community.

**Weaknesses:**

1. Although the proposed restricted Schur complement $S_{res}$ allows the Hessian $\nabla^2_{yy} f$ to be degenerate, the obtained necessary condition still requires some other assumptions on $h(\delta)$, and characterizing the limit points of extragradient through $S_{res}$ also requires other conditions like $s_0$, $\eta^*$, $\sigma_j$ or $\iota_j$. These conditions, including $S_{res}$, could be computationally more heavy and cumbersome to check. Some high-level intuition about how $S_{res}$ is designed and why it is helpful could be good as well.

2. What does the assumption that $S_{res}$ is nondegenerate imply? What assumptions on $H$ are necessary to make sure $S_{res}$ is nondegenerate?

3. I think it would help improve the clarity of the paper by explicitly saying continuous-time GDA/EG or discrete-time GDA/EG to distinguish them. In Example 2, the paper uses both Definition 2(i) and 3 when justifying the behaviors of $xy$. However, 2(i) is for continuous time and 3 is for discrete time. Actually, $\rho\\{\pm i\sqrt{\epsilon}\\}\leq 1$ implies linearly stability if applying Definition 3. Similar confusions between continuous time and discrete time exist elsewhere as well.

4. It would improve the clarity of the paper if some notations were introduced before using them, although the meaning is well-known and can be guessed from the context.  For example, $C^2$ and $DF$ in Assumption 1, $\lambda_j$ in Lemma 5.1, etc.

**Questions:**

See Weaknesses.

---

> ### Author Response · Authors · 2023-11-23
>
> We thank the reviewer for a positive evaluation and constructive feedback.
>
> 1. **Computation of quantities such as $\boldsymbol{S}_\text{res}$, $s_0$, $\eta ^ * $, $\sigma_j$, and $\iota_j$ are expensive (W1)**
>
>     The reviewer’s concern on the computational difficulties on the quantities we introduced (*e.g.*, *$\boldsymbol{S}_\text{res}$*, $s_0$, etc.) is indeed a valid point. However, we would also like to stress that the main point of our work is on the theoretical side; establishing a sound study on local minimax points and demonstrating the first ever result on how they can be found by a first order method, when the strong invertibility condition on $\nabla_{\boldsymbol{y}\boldsymbol{y}}^2 f$ is removed. Nonetheless, note that we also provide the results (Theorem 6.4 and the discrete-time counterpart in the appendix) that relaxes much of the computational load by removing the dependencies on $s_0$ (and $\eta_0$ in the discrete-time), under the mild assumption of $\sigma_j$ all being distinct.
>
> 2. **Intuition on $\boldsymbol{S}_\text{res}$ (W1)**
>
>     The design of *$\boldsymbol{S}_\text{res}$* is done so that the (classical) Schur complement $\boldsymbol{S}$ being "positive definite on a certain subspace" is equivalent to $\boldsymbol{S}_\text{res}$  being positive semidefinite (*cf*. Proposition 4.1). Such a property is directly related to the "refined second-order necessary condition" for local minimax points; see Proposition 4.2. While this line of thoughts can be inferred from Section 4, we acknowledge that our concise presentation on this part might make it hard for the readers to capture this idea. However, we regretfully ask the reviewer (and all the other readers of this paper) to mercifully understand that we cannot help but keep this part to be compactly presented as it is done now, in order to comply with the page limit.
>
> 3. **Nondegeneracy of $\boldsymbol{S}_\text{res}$ (W2)**
>
>     From the theoretical point of view, the nondegeneracy of *$\boldsymbol{S}_\text{res}$* enables us to establish the stability analyses. It is well known that determining local optimality for nonconvex-nonconcave problems efficiently without imposing any assumptions is a nearly impossible task; already for minimization problems determining if a stationary point is a minimum or not is a co-NP complete problem (see, *e.g.*, Lee et al., 2019). The current understanding of ours is that the nondegeneracy of *$\boldsymbol{S}_\text{res}$* is the necessary ingredient that enables the spectrum analyses.
>
>     As for an intuitive explanation, considering that when $\nabla_{\boldsymbol{y}\boldsymbol{y}}^2 f$ is invertible the ordinary Schur complement is strongly related to the Hessian of the primal function (Fiez et al., 2020), we conjecture that the restricted Schur complement should also take an analogous role, but unfortunately we currently do not have a rigorous proof of this claim.
>
>     Still, we would like to stress that we have extended the stability analysis by introducing the new nondegeneracy assumption on the restricted Schur complement which is much less restrictive than the previously assumed nondegeneracy of $\nabla_{\boldsymbol{y}\boldsymbol{y}}^2 f$. For now, we shall leave the investigations on the implications and/or intuitive interpretations of the invertibility of  *$\boldsymbol{S}_\text{res} (\boldsymbol{H})$*  and the studies on whether such condition can be further relaxed as future works.
>
> 4. **Continuous/Discrete and EG/GDA (W3)**
>
>     Admittedly, Example 2 was indeed confusingly written in the original manuscript, not clearly distinguishing continuous-time and discrete-time discussions. The arguments in Example 2 are now fixed to address this issue, and similar revisions are made across the paper.
>
>     Meanwhile, we would like to point out that the stability analysis of discrete GDA should be—considering the update rule of GDA—done with **$\boldsymbol{I} - \eta \boldsymbol{H}_{\tau}^{*}$** instead of  $\boldsymbol{H}_\tau^*$  itself. Hence, the equilibrium $(0, 0)$ of $\min_x \max_y \ xy$ is a local minimax point but is strict linearly unstable for (discrete-time) GDA even with timescale separation. This is now (visually) demonstrated in Appendix G.2, which we added to enhance the contrast between (discrete-time) GDA and EG.
>
> 5. **Clarifying notations (W4)**
>
>     Thank you for your suggestion. We have added the introduction of notations in the definitions and theorem statements.

---

### Official Review · Reviewer_dqsM · 2023-10-30

**Soundness:** 4 excellent
**Presentation:** 2 fair
**Contribution:** 4 excellent
**Rating:** 8
**Confidence:** 3

**Summary:**

This paper focuses on nonconvex-nonconcave minimax optimization when the Hessian $\nabla_{yy}^2 f$ is possibly degenerate. The authors introduce the concept of restricted Schur complement to refine the second-order condition and provide a characterization of the eigenvalues of the Jacobian matrix of the saddle gradient in an asymptotic sense. In particular, the degeneracy leads to pairs of nearly imaginary eigenvalues. To describe these eigenvalues, the authors investigate the curvature information through the *hemicurvature*. Based on this, it is established that the limit points of two-timescale EG in continuous time are local minimax points under mild conditions. Moreover, two-time-scale GDA may avoid non-strict minimax points, while two-timescale EG could find them, which demonstrates the superiority of two-timescale EG over two-timescale GDA.

**Strengths:**

The methods and conclusions in this article are valuable in terms of both originality and significance.

1. In terms of originality, the authors propose a new concept, the restricted Schur complement, to refine the second-order conditions. To study the stability, the authors introduce the concept of the *hemicurvature* to characterize the eigenvalues. These concepts and tools seem novel and fascinating.

2. Regarding significance, this paper improves upon previous results by eliminating the nondegenerate condition on the Hessian $\nabla_{yy} f$ and provides solid evidence to demonstrate the superiority of two-timescale EG over two-timescale GDA. I believe the methodology and conclusions in this paper are helpful for future research in degenerate cases.

3. I have checked some parts of the appendices and think the derivation is rigorous.

**Weaknesses:**

This is room for improvement in the presentation.

1. The organization is less satisfactory and some results lack intuitive interpretation in the main text. I provide several examples below.

* (i) The concept of hemicurvature is a bit opaque and the result of Proposition 6.7 is not intuitive. However, the figures in the appendix are a good illustration and could be put in the main text.
* (ii) The authors mention that they adopt the hemicurvature instead of the curvature because of the property in Proposition C.4. From the proof, this property is indeed important and the authors should elaborate on this in the main text.
* (iii) Theorem 6.4 is based on two additional conditions: the distinction of $\sigma_i$ and $u_j^\top S u_j \ge 0 $. The authors should give more explanations on these conditions, e.g., why we need these conditions or what their role is.
* (iv) In Theorems 6.2 and 6.6, there appear $s^*$ and $\eta^*$. It is worth discussing their values. For example, what is the relationship between them and $s_0$?

In summary, I advise the authors to compress the review of previous results and add more interpretation in Sections 5 and 6. The methods and conclusions are valuable and deserve more elaboration. Moreover, the abstract is a bit uninformative and should also be extended.

2. To demonstrate the superiority of two-timescale EG over two-timescale GDA, the authors could provide the results of GDA corresponding to Propositions 6.1 and 6.5, and then plot these regions in the complex plane as a better illustration.

Minor concerns:
In the second instance in Example 1, the local minimax points should be $(0,0,t,0,t)$.

**Questions:**

1. In Example 1, the restricted Schur complement for the two cases is zero or degenerate. Could the authors provide an example such that the restricted Schur complement is non-zero and non-singular?

2. If Assumption 2 is removed, do some results in Theorem 5.3 still hold?

---

> ### Author Response · Authors · 2023-11-23
>
> We thank the reviewer for a positive evaluation of our work and constructive feedback.
>
> 1. **Organizations of the paper (W1)**
>
>     Thank you for the considerate suggestions. Here are our thoughts on your recommendations.
>
>     * As correctly pointed out by the reviewer, the hemicurvature does play an interesting role in the analyses conducted in Section 6. Yet, we ask the reviewer to kindly understand our decision to keep the details in the appendices, in order to focus more on the results and the discussions in Section 6; the protagonists of our paper. Still, we would like to mention that at the end of Section 5.2.1 there does exist a brief sketch on the properties, with pointers to the relevant sections in the appendices.
>
>     * Currently, our understandings on the role of additional assumptions of $\sigma_j$ being distinct is mostly from a technical point of view; the main reason we take it is to have Lemma E.11 (which is now Lemma F.11 in the revised manuscript). Still, we think this assumption is mild one, because when the size of a matrix is fixed, the set of matrices with non-distinct singular values is of measure zero.
>
>     * The condition *$\boldsymbol{u}_j^\top \boldsymbol{S} \boldsymbol{u}_j \geq 0$* is more of a result of the theorem than an assumption, because it appears as one of the conditions for a necessary and sufficient condition for strict linear stability. Nonetheless, an interesting point to observe is that $\boldsymbol{S}_\text{res} \succeq \boldsymbol{0}$ with the additional *$\boldsymbol{u}_j^\top \boldsymbol{S} \boldsymbol{u}_j \geq 0$* gives us a condition that is somewhere between the necessary ($\boldsymbol{S}_\text{res} \succeq \boldsymbol{0}$ only) and the sufficient ($\boldsymbol{S} \succeq \boldsymbol{0}$) conditions from Theorem 6.3.
>
>     * There does exist a latent relationship between $s^*$ and $s_0$, as elaborated in the proof of Theorem 6.2. However, in the perspective of the statement of Theorem 6.2, we don’t really have to focus on how those two are related; rather, it is more natural to consider them appearing from two different conditions, which are shown to be equivalent by Theorem 6.2. The same goes to $\eta^*$ and $s_0$ in Theorem 6.6. But always, one can refer to the proofs of those theorems to find a full explanation on the quantities $s^*$  and $\eta^*$ .
>
>     * As suggested, we made the abstract more instructive, reduced the related work section, and added more explanation in Section 5. We will make the paper more readable in our next revision.
>
> 2. **Benefits of EG over GDA (W2)**
>
>     We added the results from (Fiez & Ratliff, 2021) on GDA that corresponds to our Proposition 6.1 and 6.5, with some remarks, in Appendices F.2 and G.2. There we compare GDA and EG, in particular contrasting the regions where the spectrum of $\boldsymbol{H}_\tau^∗$ can lie on for $\boldsymbol{z}^∗$ to be stable for each methods. We hope this enhances the presentation of our work.
>
> 3. **Typo in Example 1**
>
>     You are correct; the local minimax points are $(0, 0, t, 0, t)$. We appreciate your careful proofreading. The typo is now fixed.
>
> 4. **Example concerning the restricted Schur complement (Q1)**
>
>     The second example in Example 1 (which is also Example 3) is in fact a case where the restricted Schur complement is *nonsingular*, while the (classical) generalized Schur complement is degenerate. We kindly refer the reviewer to the details presented in Appendix B.3 (which is now Appendix C.3).
>
> 5. **Relation between Assumption 2 and Theorem 5.3 (Q2)**
>
>     Analyzing the spectrum of a perturbed linear operator is in general a highly challenging task, but there will still be some things we can say about Theorem 5.3 without Assumption 2. For instance, following our proof of the theorem, one should be able to draw similar conclusions about type (i) and type (iii) eigenvalues. Recalling that the type (i) eigenvalues are the ones newly identified—that is, the ones that do not appear in (Jin et al., 2020)—to some extent this also demonstrates the effectiveness of our new spectral analysis. Yet, it is also true that some conclusions will no longer hold without Assumption 2; for instance, we would not be able to conclude whether the remaining eigenvalues (*i.e.*, the ones that would have been the type (ii) eigenvalues if Assumption 2 was present) are of order $\Theta(\epsilon)$ or not.

---

### Official Review · Reviewer_yB9R · 2023-10-30

**Soundness:** 3 good
**Presentation:** 3 good
**Contribution:** 3 good
**Rating:** 8
**Confidence:** 3

**Summary:**

The paper studies the local properties of the extragradient method with timescale separation between the minimizing and maximizing player. They crucially drop the previous requirement of nondegenerate $\nabla^2_{yy} f$, by instead considered conditions on what they call the _restricted_ shur complement of the Jacobian of the saddle gradient. They show convergence to local minimax points for the discrete two timescale extragradient method and a second order continuous approximation. They further show avoidance of strict non-minimax points almost surely.

**Strengths:**

The paper addresses the important problem of local behaviour of two-timescale extragradient in nonmonotone problems, it is technically strong and well-written.

**Weaknesses:**

I only have the following remark:

It seems that Thm. 6.2-6.4, Thm 6.6 and Thm. F.3 all treats _strict_ linearly stable points. The main claim that two-timescale extragradient finds (nonstrict degenerate) local minimax points (e.g. example 3) is found in Remark 6.8. The argument relies on showing avoidance of non local minimax point coupled with a _global_ convergence guarantee to a fixed point.

The remark makes it appear as if global convergence for general nonconvex-nonconcave is solved, while Diakonikolas et al., 2021 only applies to the structured problems satisfying the weak Minty variational inequality. Furthermore, the proof provided in the appendix only applies to the Minty variational inequality (MVI).

I suggest toning down the claim and instead state that "_when_ global convergence can be guaranteed (as e.g. under MVI), Thm. 6.7 implies convergence to (degenerate) local minimax points".

Apart from this one case, the paper is otherwise very transparent about the claim that it makes.

Minor suggestions:

- Some definitions are hard to find (e.g. $H_\tau$ on page 5). I suggest moving definitions that are needed for the theorem statements to a central place to the extend that it is possible.
- It is probably worth mentioning [Bauschke et al. 2019](https://arxiv.org/pdf/1902.09827.pdf) work on linear operators regarding the relationship to comonotonicity.
- It is maybe worth contrasting the timescale separation between players with timescale separation ala [Hsieh et al. 2020](https://arxiv.org/pdf/2003.10162.pdf) (also used for weak MVI).

**Questions:**

- Thm. 4.4 of [Zhang et al. 2022](https://jmlr.csail.mit.edu/papers/volume23/20-918/20-918.pdf) seem to consider similar conditions on the restricted Shur complement. How does your results compare?
- How does the choice of $\tau$ propagate to e.g. Thm 6.2 or Thm 6.6? You make the final claims in terms of infinite time separation ($\infty$-EG). Can you claim anything about finite time separation?

---

> ### Author Response · Authors · 2023-11-23
>
> We thank the reviewer for a positive evaluation of our work and constructive feedback.
>
> 1. **On the global convergence**
>
>     We added the phrase "under a nonconvex-nonconcave setting, such as the MVI, where a global convergence to a stationary point is guaranteed" in Remark 6.8, so that the readers know the limitation of our global convergence analysis, coming from the existing limitation of the global convergence of EG.
>
>     As noted by the reviewer, Appendix I for the global convergence of the two-timescale EG is proven under the MVI condition, but not the weak MVI condition considered in (Diakonikolas et al., 2021). We would like to note that, as briefly mentioned in Appendix I, one can extend the results to the weak MVI for the EG+.
>
> 2. **On the suggestions**
>
>     * We agree that some definitions are hard to follow, which is mostly due to the page limit. We at least made the definition of $\boldsymbol{H}_\tau$ more visible.
>
>     * The work by Bauschke et al. (2019) is notable, but considering that the concept of comonotonicity in our work is introduced as an interesting side remark (and also the strict page limit of 9 pages), it seems for us that mentioning (Gorbunov et al., 2023) is sufficient for our presentation.
>
>     * Contrasting the timescale separation between players with the double step strategy in (Hsieh et al., 2020) will only make the paper clearer, but please understand that we are already running out of space, and the distinction between double step (as in (Hsieh et al., 2020)) and timescale separation (as in ours) seems clear from equation (4).
>
> 3. **Comparison with (Zhang et al., 2022) (Q1)**
>
>     Zhang et al. (2022) indeed came up with a similar concept to study the nature of local optimal points for quadratic problems, but our study applies to general minimax problems. We added mentioning Zhang et al. (2022) when introducing the restricted Schur complement, as it admittedly seems that we missed giving Zhang et al. (2022) the necessary credits for establishing a prototypical analysis in this direction.
>
> 4. **Finiteness timescale separation (Q2)**
>
>     The notation $\infty$-EG is introduced purely for convenience, and does not actually mean that the timescale separation parameter $\tau$ should be infinitely large. As stated in Definition 6, our results hold whenever we choose a *finite* time separation $\tau$ that is larger than a certain threshold $\tau^*$. The infinity symbol $\infty$ is to emphasize that the timescale separation $\tau$ being larger than a certain threshold is sufficient, and it does not matter how large $\tau$ actually is.

---

### Official Review · Reviewer_aBEe · 2023-11-10

**Soundness:** 3 good
**Presentation:** 3 good
**Contribution:** 2 fair
**Rating:** 6
**Confidence:** 2

**Summary:**

This paper delves into the realm of minimax optimization, offering a comparative analysis of the two-time scale extra gradient against the two-time scale gradient descent ascent. The authors demonstrate that while the two-time scale GDA successfully converges to a specific minimax point, it encounters difficulties in the presence of a degenerate Hessian. Viewing the problem through the lens of a continuous dynamical system, the two-time scale extra gradient emerges as the superior method. It converges to the minimax point, maintaining its effectiveness even when faced with a degenerate Hessian.

**Strengths:**

- This paper is the first to remove the non degenerate assumption in literature. It defines nature new notions of restricted Schur complement and strict non-minimax point in correspondence with their assumption.

- The paper adopts the high order ODE of EG, to resolve the issue of avoiding nonminimax points. This approach utilizes continuous dynamics techniques, which are then adeptly extended to the analysis of discrete dynamics.

**Weaknesses:**

- The authors could enhance their presentation by including additional examples or illustrative figures that emphasize the significance of the non-degenerate assumption and its impact on the algorithm's practical applicability. i.e. are there any examples that two-time scale GDA fails while two-time scale EG works?

- The absence of the conclusion and discussion sections from the main text disrupts the flow and detracts from the overall reading experience.

**Questions:**

- The timescale separation technique is proposed in Jin et al. (2020) to solve the convergence of GDA, is it still necessary in the analysis of EG? How will your analysis change if using a single-timescale EG? Is it possible to still obtain similar results?

- Is it possible to extend some of the analysis to the stochastic EG setting?

---

> ### Author Response · Authors · 2023-11-23
>
> We thank the reviewer for a positive evaluation of our work and constructive feedback.
>
> 1. **Benefits of EG over GDA (W1)**
>
>     We would like to point out that Examples 2 and 3 describe two illustrative examples (with degenerate $\nabla_{\boldsymbol{y}\boldsymbol{y}}^2 f$) that two-timescale GDA fail while two-timescale EG work.
>
>     Also, Appendices F.2 and G.2 are added to compare EG to GDA, in particular contrasting the regions where the spectrum of $\boldsymbol{H}_\tau^*$ can lie on for $\boldsymbol{z}^*$ to be stable for each methods. We hope this enhances the presentation of our work.
>
> 2. **Conclusions and Discussions Section (W2)**
>
>      We do also acknowledge that the current position of the final discussions section is unnatural. However, due to the strict page limit, we could not help but to move that section to the appendix. We at least moved the discussion section furthest forward in the appendix to make it closest to the main text.
>
> 3. **Timescale separation with EG (Q1)**
>
>     The primary intention of introducing timescale separation is indeed to achieve convergence, but more specifically, convergence to local *minimax* points. In particular, Jin et al. (2020) showed that, under the assumption that $\nabla_{\boldsymbol{y}\boldsymbol{y}}^2 f$ is invertible, a *sufficient* timescale separation in GDA leads to a convergence to local minimax point, while an *insufficient* timescale separation (*e.g.*, a single-timescale GDA) fails to converge to local minimax points.  Our paper similarly presents that *insufficient* timescale separation (*e.g.*, a single-timescale EG) fails to converge to local minimax points, but without the invertibility condition on $\nabla_{\boldsymbol{y}\boldsymbol{y}}^2 f$. So, our result already discusses the local behavior of the single-timescale EG, which states that it does not have convergence to local minimax points.
>
> 4. **Extending to stochastic settings (Q2)**
>
>     Considering the attempts to explain stochastic gradient descent through the lens of stochastic differential equations (SDEs), it might be possible to extend our analyses to stochastic EG by blending it with SDEs; but we are not sure about the details of this idea at the moment. Still, it would certainly be an interesting direction for a future work.

---

### Comment · Area_Chair_1vx3 · 2023-11-20
**Discussion period closing soon**

Dear authors, dear reviewers,

As a reminder, the author-reviewer discussion period is coming to a close in two days.

To make sure that this phase is as constructive as possible, I would kindly ask the reviewers, if you haven’t already done so, to go through the authors’ posted rebuttals, follow up on the authors’ replies to your comments, and engage with the authors if you would like to ask any further questions.

Once the discussion period closes, it will be harder to get input from the authors, so it would be better to do this before the last day.

Regards,

The AC

---

### Meta-Review · Area_Chair_1vx3 · 2023-12-05

**Metareview:**

This paper examines the local convergence guarantees of a two-timescale extra-gradient algorithm for non-convex/non-concave min-max games. The authors focus on a hierarchical, Stackelberg-like minimax solution concept that builds on earlier work by Evtushenko (1974), Jin et al (2020) and Fiez et al (2020), and they provide several convergence results.

The reviewers appreciated the paper's results but, at the same time, several concerns were raised during the committee discussion phase, namely:
1. The role of the MVI condition. Under the current write-up, the title, abstract and introduction make it sound as the proposed algorithm converges globally and universally, without any further conditions, whereas this is not the case. The authors' comment on the Minty condition in Remark 6.8 is quite easy to miss, and a detailed discussion of relevant work in the literature was left out "due to space constraints". This, however, creates an unbalanced presentation, which the committee found problematic.
2. The paper's organization and overall readability is often uneven. There are many highly technical notions and details that the authors are not discussing in sufficient length, making certain parts of the paper inaccessible - such as the notion of "hemicurvature" which plays a central role in the paper, but is barely discussed.
3. One of the paper's main claims is the importance of dropping the invertibility assumption for $\nabla_{yy}^2 f$. However, the authors are assuming that the problem's Jacobian matrix is invertible (an assumption stated at the bottom of p. 7, again easy to miss), so it is not clear how "crucial" the elimination of the condition on $\nabla_{yy}^2 f$ really is.

Overall, there was some debate between the committee whether this paper belongs to a conference or a journal (like JMLR), where the authors would not be constrained by space, and would be able to explain things in more detail. In the end, a decision was reached to make a "conditional accept" recommendation subject to the following required changes:
- Stating clearly throughout the paper (from the title to the introduction) the local nature of the paper's main results, the assumptions concerning the invertibility of $DF$, etc.
- Clarifying the relation with previous work - in terms of assumptions, definitions, etc. - and avoiding subjective statements like "eliminate a crucial assumption" and the like, especially in the abstract and the introduction.

All in all, it is important to avoid creating false impressions regarding the paper's scope and results. The paper needs some work on this front but, in the end, the committee felt that the paper's merits outweighed its flaws, so an "accept" recommendation conditioned on the above seems the most appropriate choice.

**Justification For Why Not Higher Score:**

Several concerns remain.

**Justification For Why Not Lower Score:**

The paper has some interesting contributions.

---

### Decision · Program_Chairs · 2024-01-16

Accept (poster)